# Learning from Positive and Unlabeled Data with Arbitrary Positive Shift

**Zayd Hammoudeh**      **Daniel Lowd**
Department of Computer & Information Science
University of Oregon
Eugene, OR, USA
`{zayd, lowd}@cs.uoregon.edu`

## Abstract

*Positive-unlabeled* (PU) *learning* trains a binary classifier using only positive and unlabeled data. A common simplifying assumption is that the positive data is representative of the target positive class. This assumption rarely holds in practice due to temporal drift, domain shift, and/or adversarial manipulation. This paper shows that PU learning is possible even with arbitrarily non-representative positive data given unlabeled data from the source and target distributions. Our key insight is that only the negative class's distribution need be fixed. We integrate this into two statistically consistent methods to address arbitrary positive bias – one approach combines *negative-unlabeled learning* with *unlabeled-unlabeled learning* while the other uses a novel, recursive risk estimator. Experimental results demonstrate our methods' effectiveness across numerous real-world datasets and forms of positive bias, including disjoint positive class-conditional supports. Additionally, we propose a general, simplified approach to address PU risk estimation overfitting.

## 1 Introduction

*Positive-negative* (PN) *learning* (i.e., ordinary supervised classification) trains a binary classifier using positive and negative labeled datasets. In practice, good labeled data are often unavailable for one class. High negative-class diversity may make constructing a representative labeled set prohibitively difficult [1], or negative data may not be systematically recorded in some domains [2].

*Positive-unlabeled* (PU) *learning* addresses this problem by constructing classifiers using only labeled-positive and unlabeled data. PU learning has been applied to numerous real-world domains including: opinion spam detection [3], disease-gene identification [4], land-cover classification [5], and protein similarity prediction [6]. The related task of *negative-unlabeled* (NU) learning is functionally identical to PU learning but with labeled data drawn from the negative class.

Most PU learning methods assume the labeled set is *selected completely at random* (SCAR) from the target distribution [1, 6, 7, 8, 9, 10, 11]. External factors like temporal drift, domain shift, and adversarial concept drift often cause the labeled-positive and target distributions to diverge.

*Biased-positive, unlabeled* (bPU) *learning* algorithms relax SCAR by modeling *sample selection bias* for the labeled data [12, 13] or a *covariate shift* between the training and target distributions [14].

This paper generalizes bPU learning to the more challenging *arbitrary-positive, unlabeled* (aPU) learning setting, where the labeled (positive) data may be *arbitrarily different* from the target distribution's positive class. Solving this problem would eliminate the need to spend time and money labeling new data whenever the positive class drifts.

Devoid of some assumption, aPU learning is impossible [6]. As a first step to address aPU learning, our key insight is that given a labeled-positive set and two unlabeled sets as proposed by Sakai and Shimizu [14], aPU learning is possible when *all negative examples are generated from a single distribution*. The labeled and target-positive distributions' *supports* (sets of examples with non-zero probability) may even be disjoint. Many real-world PU learning tasks feature a shifting positive class but (largely) fixed negative class including:

1. **Land-Cover Classification**: Cross-border land-cover datasets often do not exist due to differing national technological standards or insufficient financial resources by one country [15]. This limits research into natural processes at broad geographic scales. However, cross-border geographic terrains often follow a similar distribution differing primarily in man-made objects (e.g., roads) due to local construction materials and regulations [5].
2. **Adversarial aPU Learning**: Malicious adversaries (email spammers, malware authors) rapidly adapt their attacks to bypass automated detection. The benign class changes much more slowly but may be too diverse to construct a representative labeled set [3, 16, 17, 18].

Our paper's four primary contributions are enumerated below. Note that most experiments and all proofs are in the supplemental materials.

1. We propose abs-PU – a simplified, statistically *consistent* approach to correct general PU risk estimation overfitting. Our aPU methods leverage abs-PU to streamline their optimization.
2. We address our aPU learning task via a two-step formulation; the first step applies standard PU learning and the second uses unlabeled-unlabeled (UU) learning.
3. We separately propose PURR – a novel, recursive, consistent aPU risk estimator.
4. We evaluate our methods on a wide range of benchmarks, demonstrating our algorithms' effectiveness over the state of the art in PU and bPU learning. Our empirical evaluation includes an adversarial aPU learning case study using public spam email datasets.

## 2   Ordinary Positive-Unlabeled Learning

We begin with an overview of PU learning without distributional shifts, including definitions and notation. Consider two random variables, covariate $X \in \mathbb{R}^d$ and label $Y \in \{\pm 1\}$, with joint distribution $p(x, y)$. Marginal distribution $p_\mathrm{u}(x)$ is composed from the positive prior $\pi \coloneqq p(Y = +1)$, positive class-conditional $p_\mathrm{p}(x) \coloneqq p(x|Y = +1)$, and negative class-conditional $p_\mathrm{n}(x) \coloneqq p(x|Y = -1)$.

**Risk**   Let $g : \mathbb{R}^d \to \mathbb{R}$ be an arbitrary *decision function* parameterized by $\theta$, and let $\ell : \mathbb{R} \to \mathbb{R}_{\geq 0}$ be the *loss function*. Risk $R(g) \coloneqq \mathbb{E}_{(X,Y) \sim p(x,y)}[\ell(Y g(X))]$ quantifies $g$'s expected loss over $p(x, y)$. It decomposes via the product rule to $R(g) = \pi R_\mathrm{p}^+(g) + (1 - \pi) R_\mathrm{n}^-(g)$, where the *labeled risk* is

$$R_\mathcal{D}^{\hat{y}}(g) \coloneqq \mathbb{E}_{X \sim p_\mathcal{D}(x)}[\ell(\hat{y} g(X))] \tag{1}$$

for predicted label $\hat{y} \in \{\pm 1\}$ and $\mathcal{D} \in \{\mathrm{p}, \mathrm{n}, \mathrm{u}\}$ denoting the positive class-conditional, negative class-conditional or marginal distribution respectively, as defined above.

Since $p(x, y)$ is unknown, *empirical risk* is used in practice. We consider the *case-control scenario* [19] where each dataset is i.i.d. sampled from its associated distribution. PN learning has two labeled datasets: positive set $\mathcal{X}_\mathrm{p} \coloneqq \{x_i^\mathrm{p}\}_{i=1}^{n_\mathrm{p}} \overset{\text{i.i.d.}}{\sim} p_\mathrm{p}(x)$ and negative set $\mathcal{X}_\mathrm{n} \coloneqq \{x_i^\mathrm{n}\}_{i=1}^{n_\mathrm{n}} \overset{\text{i.i.d.}}{\sim} p_\mathrm{n}(x)$. These are used to calculate empirical labeled risks $\widehat{R}_\mathrm{p}^+(g) = \frac{1}{n_\mathrm{p}} \sum_{i=1}^{n_\mathrm{p}} \ell(g(x_i^\mathrm{p}))$ and $\widehat{R}_\mathrm{n}^-(g) = \frac{1}{n_\mathrm{n}} \sum_{i=1}^{n_\mathrm{n}} \ell(-g(x_i^\mathrm{n}))$. We denote the empirical positive-negative risk

$$\widehat{R}_\mathrm{PN}(g) \coloneqq \pi \widehat{R}_\mathrm{p}^+(g) + (1 - \pi) \widehat{R}_\mathrm{n}^-(g). \tag{2}$$

PU learning cannot directly estimate $R_\mathrm{n}^{\hat{y}}(g)$ since there is no negative (labeled) data (i.e., $\mathcal{X}_\mathrm{n} = \emptyset$). Let $\mathcal{X}_\mathrm{u} \coloneqq \{x_i^\mathrm{u}\}_{i=1}^{n_\mathrm{u}} \overset{\text{i.i.d.}}{\sim} p_\mathrm{u}(x)$ be an unlabeled set with empirical labeled risk $\widehat{R}_\mathrm{u}^{\hat{y}}(g) = \frac{1}{n_\mathrm{u}} \sum_{i=1}^{n_\mathrm{u}} \ell(\hat{y} g(x_i^\mathrm{u}))$. du Plessis et al. [20] make a foundational contribution that,

$$(1 - \pi) R_\mathrm{n}^{\hat{y}}(g) = R_\mathrm{u}^{\hat{y}}(g) - \pi R_\mathrm{p}^{\hat{y}}(g). \tag{3}$$

Their unbiased PU (uPU) risk estimator is therefore $\widehat{R}_\mathrm{uPU}(g) \coloneqq \pi \widehat{R}_\mathrm{p}^+(g) + \widehat{R}_\mathrm{u}^-(g) - \pi \widehat{R}_\mathrm{p}^-(g)$. Kiryo et al. [8] observe that highly expressive models (e.g., neural networks) often overfit $\mathcal{X}_\mathrm{p}$ causing uPU to estimate that $\widehat{R}_\mathrm{u}^-(g) - \pi \widehat{R}_\mathrm{p}^-(g) < 0$.

Since negative-valued risk is impossible, Kiryo et al.'s non-negative PU (nnPU) risk estimator ignores negative estimates of risk via a $\max$ term:

$$\widehat{R}_{\text{nnPU}}(g) := \pi \widehat{R}_{\text{p}}^{+}(g) + \max\{0, \widehat{R}_{\text{u}}^{-}(g) - \pi \widehat{R}_{\text{p}}^{-}(g)\}. \tag{4}$$

When Kiryo et al.'s customized empirical risk minimization (ERM) framework detects overfitting (i.e., $\widehat{R}_{\text{u}}^{-}(g) - \pi \widehat{R}_{\text{p}}^{-}(g) < 0$), their framework "defits" $g$ using negated gradient $-\gamma \nabla_{\theta}(\widehat{R}_{\text{u}}^{-}(g) - \pi \widehat{R}_{\text{p}}^{-}(g))$, where hyperparameter $\gamma \in (0, 1]$ attenuates the learning rate to throttle "defitting." Observe that positive-labeled risk, $\widehat{R}_{\text{p}}^{+}(g)$, is excluded from nnPU's negated gradient.

## 3 Simplifying Non-Negativity Correction

Rather than enforcing the non-negative risk constraint with two combined techniques (a $\max$ term and "defitting") like Kiryo et al., we propose a simpler approach, inspired by Lagrange multipliers, that directly puts the non-negativity constraint into the risk estimator. Our *absolute-value correction*,

$$(1 - \pi)\ddot{R}_{\text{n}}^{\hat{y}}(g) := \left| \widehat{R}_{\text{u}}^{\hat{y}}(g) - \pi \widehat{R}_{\text{p}}^{\hat{y}}(g) \right|, \tag{5}$$

replaces nnPU's $\max$ with absolute value to prevent the optimizer overfitting an implausible risk estimate by explicitly penalizing those risk estimates for being negative. This penalty "defits" the learner automatically, eliminating the need for hyperparameter $\gamma$ and nnPU's custom ERM algorithm.

**Theorem 1.** *Let $g : \mathbb{R}^d \to \mathbb{R}$ be an arbitrary decision function and let $\ell : \mathbb{R} \to \mathbb{R}_{\geq 0}$ be a loss function bounded[1] w.r.t. $g$ then $\ddot{R}_{\text{n}}^{\hat{y}}(g)$ is a consistent estimator of $\widehat{R}_{\text{n}}^{\hat{y}}(g)$.*

We integrate absolute value correction into our *abs-PU risk estimator*,

$$\widehat{R}_{\text{abs-PU}}(g) := \pi \widehat{R}_{\text{p}}^{+}(g) + \left| \widehat{R}_{\text{u}}^{-}(g) - \pi \widehat{R}_{\text{p}}^{-}(g) \right|, \tag{6}$$

which by Theorem 1 is consistent like nnPU. When $\widehat{R}_{\text{u}}^{-}(g) - \pi \widehat{R}_{\text{p}}^{-}(g) < 0$, abs-PU's update gradient, $\nabla_{\theta}(\pi \widehat{R}_{\text{p}}^{+}(g) - \widehat{R}_{\text{u}}^{-}(g) + \pi \widehat{R}_{\text{p}}^{-}(g))$, includes $\widehat{R}_{\text{p}}^{+}(g)$. Hence, abs-PU spends comparatively more time optimizing the positive-labeled risk than nnPU. Also, by penalizing implausible risk, abs-PU estimates validation performance (i.e., risk) differently than nnPU.

Empirically we observed that abs-PU yields models of similar or slightly better accuracy than nnPU albeit with a simpler, more efficient optimization. The following builds on abs-PU with a full comparison to nnPU in supplemental Section E.6.

## 4 Arbitrary-Positive, Unlabeled Learning

*Arbitrary-positive unlabeled* (aPU) learning — the focus of this work — is one of three problem settings proposed by Sakai and Shimizu [14]. We generalize their original definition below.

Consider two joint distributions: train $p_{\text{tr}}(x, y)$ and test $p_{\text{te}}(x, y)$. Notation $p_{\text{tr-}\mathcal{D}}(x)$ where $\mathcal{D} \in \{\text{p}, \text{n}, \text{u}\}$ refers to the training positive class-conditional, negative class-conditional, and marginal distributions respectively. $p_{\text{te-}\mathcal{D}}(x)$ denotes the corresponding test distributions.

No assumption is made about the label's conditional probability, i.e., $p_{\text{tr}}(y|x)$ and $p_{\text{te}}(y|x)$, nor about positive class-conditionals $p_{\text{tr-p}}(x)$ and $p_{\text{te-p}}(x)$. We only assume a fixed negative class-conditional

$$p_{\text{n}}(x) = p_{\text{tr-n}}(x) = p_{\text{te-n}}(x). \tag{7}$$

Both the train and test positive-class priors, $\pi_{\text{tr}}$ and $\pi_{\text{te}}$ respectively, are treated as known throughout this work. In practice, they may be known *a priori* through domain-specific knowledge. Techniques also exist to estimate them from data [2, 21, 22, 23]. Theorem 4 in the supplemental materials provides an algorithm to estimate $\pi_{\text{te}}$ by training an additional classifier.

As shown in Figure 1a, the available datasets are: labeled (positive) set $\mathcal{X}_{\text{p}} \overset{\text{i.i.d.}}{\sim} p_{\text{tr-p}}(x)$ as well as unlabeled sets $\mathcal{X}_{\text{tr-u}} := \{x_i\}_{i=1}^{n_{\text{tr-u}}} \overset{\text{i.i.d.}}{\sim} p_{\text{tr-u}}(x)$ and $\mathcal{X}_{\text{te-u}} := \{x_i\}_{i=1}^{n_{\text{te-u}}} \overset{\text{i.i.d.}}{\sim} p_{\text{te-u}}(x)$ with their empirical risks defined as before. An optimal classifier minimizes the *test* risk/expected loss: $\mathbb{E}_{(X,Y) \sim p_{\text{te}}(x,y)}[\ell(Y g(X))]$.

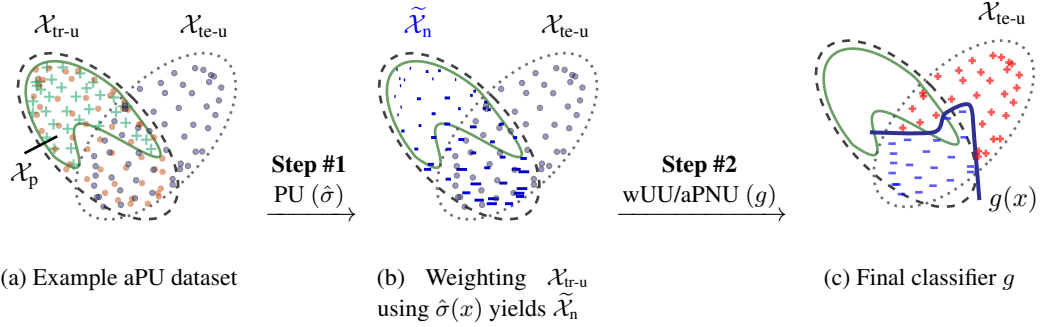

(a) Example aPU dataset

(b) Weighting $\mathcal{X}_{\text{tr-u}}$ using $\hat{\sigma}(x)$ yields $\widetilde{\mathcal{X}}_{\text{n}}$

(c) Final classifier $g$

Figure 1: Two-step aPU learning. Fig. 1a shows a toy aPU dataset with (+) representing a labeled positive example, (•) an unlabeled train sample, and (•) an unlabeled test sample. Borders surround each set for clarity. After learning probabilistic classifier $\hat{\sigma}$ in Step #1, Fig. 1b visualizes $\hat{\sigma}$'s predicted negative-posterior probability using marker (–) size. Fig. 1c shows the final decision boundary with (-) and (•) representing $\mathcal{X}_{\text{te-u}}$ examples classified negative and positive respectively.

## 4.1 Relating aPU Learning and Covariate Shift Adaptation Methods

Covariate shift [24] is a common technique to address differences between $p_{\text{tr}}(x, y)$ and $p_{\text{te}}(x, y)$. Unlike aPU learning, covariate shift restrictively assumes a *consistent input-output relation*, i.e., $p_{\text{tr}}(y|x) = p_{\text{te}}(y|x)$. Define the *importance function* as $w(x) := \frac{p_{\text{te-u}}(x)}{p_{\text{tr-u}}(x)}$. When $p(y|x)$ is fixed, it is easy to show that $w(x)p_{\text{tr}}(x, y) = p_{\text{te}}(x, y)$.

Sakai and Shimizu [14] exploit this relationship in their PUc risk estimator. $w(x)$ is approximated via direct density-ratio estimation [25] – specifically the RuLSIF algorithm [26] over $\mathcal{X}_{\text{tr-u}}$ and $\mathcal{X}_{\text{te-u}}$. Their PUc risk adds importance weighting to uPU, with the labeled risks still estimated from $\mathcal{X}_{\text{p}}$ and $\mathcal{X}_{\text{tr-u}}$. Sakai and Shimizu's formulation specifies linear-in-parameter models to enforce convexity. They improve tractability via a simplified version of du Plessis et al. [1]'s surrogate squared loss for $\ell$.

Selection bias bPU methods [12, 13] need the positive-labeled data to meet specific conditions that arbitrary-positive data will not satisfy making a comparison to those methods infeasible. PUc serves as the primary baseline here since as a covariate shift bPU method, it places no requirements on the positive data beyond that the training distribution's support be a superset of the target positive class.

## 4.2 Comparing Variations of the aPU Learning Problem

Sakai and Shimizu [14] show that PU learning with a fixed positive class and arbitrary negative shift is much simpler than aPU learning. In fact, provided a positive-labeled set and two unlabeled sets as above, they show that arbitrary negative shift is trivially equivalent to ordinary PU learning over $\mathcal{X}_{\text{p}}$ and $\mathcal{X}_{\text{te-u}}$ (since $\mathcal{X}_{\text{p}}$ being drawn from $p_{\text{te-p}}(x)$ renders $\mathcal{X}_{\text{tr-u}}$ unnecessary). When both the positive and negative classes shift arbitrarily, learning is impossible without additional data and/or assumptions. aPU learning's complexity sits between these two extremes.

# 5 aPU Learning via Unlabeled-Unlabeled Learning

To build an intuition for solving the aPU learning problem, consider the ideal case where a perfect classifier correctly labels $\mathcal{X}_{\text{tr-u}}$. Let $\mathcal{X}_{\text{tr-n}}$ be $\mathcal{X}_{\text{tr-u}}$'s negative examples. $\mathcal{X}_{\text{tr-n}}$ is SCAR w.r.t. $p_{\text{tr-n}}(x)$ and by Eq. (7)'s assumption also $p_{\text{te-n}}(x)$. Multiple options exist to then train the second classifier, $g$, e.g., NU learning with $\mathcal{X}_{\text{tr-n}}$ and $\mathcal{X}_{\text{te-u}}$.

A perfect classifier is unrealistic. Is there an alternative? Our key insight is that by weighting $\mathcal{X}_{\text{tr-u}}$ (similar to covariate shift's importance function) it can be transformed into a representative negative set. From there, we consider two methods to fit the second classifier $g$: one a variant of NU learning we call weighted-unlabeled, unlabeled (wUU) learning and the other a semi-supervised method we call arbitrary-positive, negative, unlabeled (aPNU) learning. We refer to the complete algorithms as PU2wUU and PU2aPNU, respectively.

---

**Algorithm 1** Two-step unlabeled-unlabeled aPU learning

---
**Input**: Labeled-positive set $\mathcal{X}_p$ and unlabeled sets $\mathcal{X}_{\text{tr-u}}, \mathcal{X}_{\text{te-u}}$
**Output**: $g$'s model parameters $\theta$
 1: Train probabilistic classifier $\hat{\sigma}$ using $\mathcal{X}_p$ and $\mathcal{X}_{\text{tr-u}}$
 2: Use $\hat{\sigma}$ to transform $\mathcal{X}_{\text{tr-u}}$ into surrogate negative set $\widetilde{\mathcal{X}}_n$
 3: Train final classifier, $g$, using ERM with $\widehat{R}_{\text{wUU}}(g)$ or $\widehat{R}_{\text{aPNU}}(g)$

---

Figure 1 visualizes our two-step approach, with a formal presentation in Algorithm 1. Below is a detailed description and theoretical analysis.

**Step #1: Create Surrogate Negative Set $\widetilde{\mathcal{X}}_n$ from $\mathcal{X}_{\text{tr-u}}$**

This step's goal is to learn the training distribution's negative class-posterior, $p_{\text{tr}}(Y = -1|x)$. We achieve this by training PU probabilistic classifier $\hat{\sigma} : \mathbb{R}^d \to [0, 1]$ using $\mathcal{X}_p$ and $\mathcal{X}_{\text{tr-u}}$. In principle, any probabilistic PU method can be used; we focused on ERM-based PU methods so the logistic loss served as surrogate, $\ell$. Sigmoid activation is applied to the model's output to bound its range to $(0, 1)$.

**Theorem 2.** *Let $g : \mathbb{R}^d \to \mathbb{R}$ be an arbitrary decision function and $\ell : \mathbb{R} \to \mathbb{R}_{\geq 0}$ be a loss function bounded w.r.t. $g$. Let $\hat{y} \in \{\pm 1\}$ be a predicted label. Define $\mathcal{X}_{\text{tr-u}} := \{x_i\}_{i=1}^{n_{\text{tr-u}}} \overset{\text{i.i.d.}}{\sim} p_{\text{tr-u}}(x)$, and restrict $\pi_{\text{tr}} \in [0, 1)$. Define $\widetilde{R}_{\text{n-u}}^{\hat{y}}(g) := \frac{1}{n_{\text{tr-u}}} \sum_{x_i \in \mathcal{X}_{\text{tr-u}}} \frac{\hat{\sigma}(x_i)\ell(\hat{y}g(x_i))}{1 - \pi_{\text{tr}}}$. Let $\hat{\sigma} : \mathbb{R}^d \to [0, 1]$ be in hypothesis set $\widehat{\Sigma}$. When $\hat{\sigma}(x) = p_{\text{tr}}(Y = -1|x)$, $\widetilde{R}_{\text{n-u}}^{\hat{y}}(g)$ is an unbiased estimator of $R_{\text{n}}^{\hat{y}}(g)$. When the concept class of functions that defines $p_{\text{tr}}(Y = -1|x)$ is probably approximately correct (PAC) learnable by some PAC-learning algorithm $\mathcal{A}$ that selects $\hat{\sigma} \in \widehat{\Sigma}$, then $\widetilde{R}_{\text{n-u}}^{\hat{y}}(g)$ is a consistent estimator of $R_{\text{n}}^{\hat{y}}(g)$.*

From Theorem 2, we see that *soft* weighting each unlabeled instance in $\mathcal{X}_{\text{tr-u}}$ by $\hat{\sigma}$ yields a *surrogate negative set* $\widetilde{\mathcal{X}}_n$ that can be used to estimate the train/test negative labeled risk. We form $\widetilde{\mathcal{X}}_n$ transductively, but inductive learning is an option. Since $\mathcal{X}_{\text{tr-u}}$ contains positive examples, $\hat{\sigma}$ may overfit and memorize random positive example variation. This is usually detectable via an implausible validation loss given $\pi_{\text{tr}}$, $n_p$, and $n_{\text{tr-u}}$. Care should be shown to tune $\hat{\sigma}$'s capacity and regularization.

Supplemental Section E.7 proposes and empirically evaluates two additional methods to construct $\widetilde{\mathcal{X}}_n$. While these other methods are not statistically consistent, they may outperform soft weighting.

**What if $\mathcal{X}_p$ is not SCAR?**  Our aPU learning setting, detailed in Section 4, specifies that $\mathcal{X}_p$ is representative of $\mathcal{X}_{\text{tr-u}}$'s positive examples. In scenarios where $\mathcal{X}_p$ is biased w.r.t. $\mathcal{X}_{\text{tr-u}}$, any bPU method (e.g., [12, 13]) can be used in step #1 to (hard) label $\mathcal{X}_{\text{tr-u}}$ thereby constructing $\widetilde{\mathcal{X}}_n$.

**Step #2: Train the Test Distribution Classifier $g$**

Negative-unlabeled (NU) learning is functionally the same as PU learning. Sakai et al. [27] formalize an unbiased NU risk estimator, $\widehat{R}_{\text{NU}}(g) := \left| \widehat{R}_u^+(g) - (1 - \pi)\widehat{R}_n^+(g) \right| + (1 - \pi)\widehat{R}_n^-(g)$ (defined here with our absolute-value correction). Our *weighted-unlabeled, unlabeled*[2] (wUU) estimator,

$$\widehat{R}_{\text{wUU}}(g) := \left| \widehat{R}_{\text{te-u}}^+(g) - (1 - \pi_{\text{te}})\widetilde{R}_{\text{n-u}}^+(g) \right| + (1 - \pi_{\text{te}})\widetilde{R}_{\text{n-u}}^-(g), \tag{8}$$

modifies Sakai et al.'s definition to use $\widetilde{\mathcal{X}}_n$ and $\mathcal{X}_{\text{te-u}}$. Observe that $\widehat{R}_{\text{wUU}}(g)$ uses only data that was originally unlabeled. When $\widetilde{R}_{\text{n-u}}^{\hat{y}}(g)$ is consistent, wUU is also consistent just like nnPU/abs-PU.

**Risk Estimation with Positive Data Reuse**  When $p_{\text{tr-p}}(x)$'s and $p_{\text{te-p}}(x)$'s supports intersect, $\mathcal{X}_p$ may contain useful information about the target distribution given limited data. In such settings, a semi-supervised approach leveraging $\mathcal{X}_p$, surrogate $\widetilde{\mathcal{X}}_n$, and $\mathcal{X}_{\text{te-u}}$ may perform better than wUU.

Sakai et al. [27] propose the PNU risk estimator, $\widehat{R}_{\text{PNU}}(g) := (1 - \rho)\widehat{R}_{\text{PN}}(g) + \rho\widehat{R}_{\text{NU}}(g)$, where hyperparameter $\rho \in (0, 1)$ weights the PN and NU estimators. Our *arbitrary-positive, negative, unlabeled* (aPNU) risk estimator in Eq. (9) modifies PNU to use $\widetilde{\mathcal{X}}_n$ and our absolute-value correction.

$$\widehat{R}_{\text{aPNU}}(g) = (1 - \rho)\pi_{\text{te}}\widehat{R}_p^+(g) + (1 - \pi_{\text{te}})\widetilde{R}_{\text{n-u}}^-(g) + \rho \left| \widehat{R}_{\text{te-u}}^+(g) - (1 - \pi_{\text{te}})\widetilde{R}_{\text{n-u}}^+(g) \right| \tag{9}$$

If $\rho = 0$, aPNU ignores the test distribution (i.e., $\mathcal{X}_{\text{te-u}}$) entirely. If $\rho = 1$, aPNU is simply wUU. When a large positive shift is expected (e.g., by domain-specific knowledge), $\mathcal{X}_p$ is of limited value so set $\rho$ closer to 1. For small expected positive shifts, set $\rho$ closer to 0. A midpoint value of $\rho = 0.5$ empirically performed well when no knowledge about the positive shift was assumed.

**ERM Framework** Both $\widehat{R}_{\text{wUU}}(g)$ and $\widehat{R}_{\text{aPNU}}(g)$ integrate into a standard ERM framework since they use our absolute-value correction. For completeness, supplemental materials Section C.1 details their custom ERM algorithm if Kiryo et al. [8]'s non-negativity correction is used instead.

**Heterogeneous Classifiers** Two-step learners enable different learner architectures in each step (e.g., random forest for step #1 and a neural network for step #2). Our experiments leverage this flexibility where $\hat{\sigma}$'s neural network may have fewer hidden layers or different hyperparameters than $g$ in step #2.

# 6  Positive-Unlabeled Recursive Risk Estimation

Two-step methods — both ours and PUc — solve a challenging problem by decomposing it into sequential (easier) subproblems. Serial decision making's disadvantage is that earlier errors propagate and can be amplified when subsequent decisions are made on top of those errors.

Can our aPU problem setting be learned in a single *joint* method? Sakai and Shimizu leave it as an open question. We show in this section the answer is yes. To understand why this is possible, it helps to simplify our perspective of unbiased PU and NU learning. When estimating a labeled risk, $\widehat{R}_{\mathcal{D}}^{\hat{y}}(g)$ (where $\mathcal{D} \in \{\text{p, n}\}$), the ideal case is to use SCAR data from class-conditional distribution $p_{\mathcal{D}}(x)$. When such labeled data is unavailable, the risk *decomposes* via the simple linear transformation,

$$(1 - \alpha)\widehat{R}_A^{\hat{y}}(g) = \widehat{R}_u^{\hat{y}}(g) - \alpha\widehat{R}_B^{\hat{y}}(g) \tag{10}$$

where $A = \text{n}$ and $B = \text{p}$ for PU learning or vice versa for NU learning. $\alpha$ is the positive (negative) prior for PU (NU) learning.

In standard PU and NU learning, either $\widehat{R}_A^{\hat{y}}(g)$ or $\widehat{R}_B^{\hat{y}}(g)$ can always be estimated from labeled data. If that were not true, can this decomposition be applied recursively (i.e., nested)? The answer is again yes. Below we apply recursive risk decomposition to our aPU learning task.

**Applying Recursive Risk to aPU learning**

Our positive-unlabeled recursive risk (PURR) estimator quantifies our aPU setting's empirical risk and integrates into a standard ERM framework. PURR's top-level definition is simply the test risk:

$$\widehat{R}_{\text{PURR}}(g) = \pi_{\text{te}}\widehat{R}_{\text{te-p}}^+(g) + (1 - \pi_{\text{te}})\widehat{R}_{\text{te-n}}^-(g). \tag{11}$$

Since only unlabeled data is drawn from the test distribution, both terms in Eq. (11) require risk decomposition. First, for $\widehat{R}_{\text{te-n}}^-(g)$, we consider its more general form $\widehat{R}_{\text{te-n}}^{\hat{y}}(g)$ below since $\widehat{R}_{\text{te-n}}^+(g)$ will be needed as well. Using Eq. (7)'s assumption, $\widehat{R}_{\text{te-n}}^{\hat{y}}(g)$ can be estimated directly from the training distribution. Combining Eq. (3) with absolute-value correction, we see that

$$\widehat{R}_{\text{te-n}}^{\hat{y}}(g) = \widehat{R}_{\text{tr-n}}^{\hat{y}}(g) = \frac{1}{1 - \pi_{\text{tr}}}\left|\widehat{R}_{\text{tr-u}}^{\hat{y}}(g) - \pi_{\text{tr}}\widehat{R}_{\text{tr-p}}^{\hat{y}}(g)\right|. \tag{12}$$

Next, $\widehat{R}_{\text{te-p}}^+(g)$, as a positive risk, undergoes NU decomposition so (with absolute-value correction):

$$\pi_{\text{te}}\widehat{R}_{\text{te-p}}^+(g) = \left|\widehat{R}_{\text{te-u}}^+(g) - (1 - \pi_{\text{te}})\widehat{R}_{\text{te-n}}^+(g)\right|. \tag{13}$$

Eq. (12) with $\hat{y} = +1$ substitutes for $\widehat{R}_{\text{te-n}}^+(g)$ in Eq. (13) yielding $\widehat{R}_{\text{PURR}}(g)$'s complete definition:

$$\widehat{R}_{\text{PURR}}(g) = \underbrace{\left|\widehat{R}_{\text{te-u}}^+(g) - (1 - \pi_{\text{te}})\underbrace{\left|\frac{\widehat{R}_{\text{tr-u}}^+(g) - \pi_{\text{tr}}\widehat{R}_{\text{tr-p}}^+(g)}{1 - \pi_{\text{tr}}}\right|}_{\widehat{R}_{\text{te-n}}^+(g)}\right|}_{\pi_{\text{te}}\widehat{R}_{\text{te-p}}^+(g)} + (1 - \pi_{\text{te}})\underbrace{\left|\frac{\widehat{R}_{\text{tr-u}}^-(g) - \pi_{\text{tr}}\widehat{R}_{\text{tr-p}}^-(g)}{1 - \pi_{\text{tr}}}\right|}_{\widehat{R}_{\text{te-n}}^-(g)}. \tag{14}$$

**Theorem 3.** *Fix decision function $g \in \mathcal{G}$. If $\ell$ is bounded over $g(x)$'s image and $\widehat{R}_{\text{te-n}}^{\hat{y}}(g)$, $\widehat{R}_{\text{te-p}}^{+}(g) > 0$ for $\hat{y} \in \{\pm 1\}$, then $\widehat{R}_{\text{PURR}}(g)$ is a consistent estimator. $\widehat{R}_{\text{PURR}}(g)$ is a biased estimator unless for all $\mathcal{X}_{\text{tr-u}} \overset{\text{i.i.d.}}{\sim} p_{\text{tr-u}}(x)$, $\mathcal{X}_{\text{te-u}} \overset{\text{i.i.d.}}{\sim} p_{\text{te-u}}(x)$, and $\mathcal{X}_{\text{p}} \overset{\text{i.i.d.}}{\sim} p_{\text{tr-p}}(x)$ it holds that $\Pr[\widehat{R}_{\text{tr-u}}^{\hat{y}}(g) - (1 - \pi_{\text{te}})\widehat{R}_{\text{tr-p}}^{\hat{y}}(g) < 0] = 0$ and $\Pr[\widehat{R}_{\text{te-u}}^{+}(g) - (1 - \pi_{\text{te}})\widehat{R}_{\text{te-n}}^{+}(g) < 0] = 0$.*

**Optimization**  PURR with absolute-value correction integrates into a standard ERM framework. If non-negativity is used instead, PURR's optimization scheme becomes significantly more complicated as it must consider four candidate gradients per update; see suppl. Section C.2 for more details.

# 7   Experimental Results

We empirically studied the effectiveness of our methods – PURR, PU2wUU, and PU2aPNU – using synthetic and real-world data.[3] Limited space allows us to discuss only two experiment sets here. Suppl. Section E details experiments on: synthetic data, 10 LIBSVM datasets [30] under a totally different positive-bias condition, and a study of our methods' robustness to negative-class shift.

## 7.1   Experimental Setup

Supplemental Section D enumerates our complete experimental setup with a brief summary below.

**Baselines**  PUc [14] with a linear-in-parameter model and Gaussian kernel basis is the primary baseline.[4] Ordinary nnPU is the performance floor. To ensure the strongest baseline, we separately trained nnPU with unlabeled set $\mathcal{X}_{\text{te-u}}$ as well as with the combined $\mathcal{X}_{\text{tr-u}} \cup \mathcal{X}_{\text{te-u}}$ (using the true, composite prior) and report each experiment's best performing configuration, denoted nnPU*. PN-test (trained on labeled $\mathcal{X}_{\text{te-u}}$) provides a reference for the performance ceiling. All methods saw identical training/test data splits and where applicable used the same initial weights.

**Datasets**  Section 7.2 considers the MNIST [31], CIFAR10 [32], and 20 Newsgroups [33] datasets with binary classes formed by partitioning each dataset's labels. Section 7.3 uses two different TREC [34] spam email datasets to demonstrate our methods' performance under real-world adversarial concept drift. Further details on all datasets are in the supplemental materials.

**Learner Architecture**  We focus on training neural networks (NNs) via stochastic optimization (i.e., AdamW [35] with AMSGrad [36]). Probabilistic classifier, $\hat{\sigma}$, used our abs-PU risk estimator with logistic loss. All other learners used sigmoid loss for $\ell$. Since PUc is limited to linear models with Gaussian kernels, we limited our NNs to at most three fully-connected layers of 300 neurons. For MNIST, our NNs were trained from scratch. Pretrained deep networks encoded the CIFAR10, 20 Newsgroups, and TREC spam datasets into static representations all learners used. Specifically, the 20 Newsgroups documents and TREC emails were encoded into 9,216 dimensional vectors using ELMo [37]. This encoding scheme was used by Hsieh et al. [11] and is based on [38]. DenseNet-121 [39] encoded each CIFAR10 image into a 1,024 dimensional vector.

**Hyperparameters**  Our only individually tuned hyperparameters are learning rate and weight decay. We assume the worst case of no *a priori* knowledge about the positive shift so midpoint value $\rho = 0.5$ was used. PUc's hyperparameters were tuned via importance-weighted cross validation [40]. For the complete hyperparameter details, see supplemental materials Section D.8.

## 7.2   Partially and Fully Disjoint Positive Class-Conditional Supports

Here we replicate scenarios where positive subclasses exist only in the test distribution (e.g., adversarial zero-day attacks). These experiments are modeled after Hsieh et al. [11]'s experiments for positive, unlabeled, biased-negative (PUbN) learning.

Table 1 lists the experiments' positive train/test and negative class definitions. Datasets are sampled u.a.r. from their constituent sublabels. Each dataset has four experimental conditions (ordered by row number): (1) $P_{\text{train}} = P_{\text{test}}$, i.e., no bias, (2 & 3 resp.) partially disjoint positive supports without and with prior shift, and (4) disjoint positive class definitions. $\pi_{\text{te}}$ equals $P_{\text{test}}$'s true prior w.r.t. $P_{\text{test}} \sqcup N$.

Table 1: Mean inductive misclassification rate (%) over 100 trials for MNIST, 20 Newsgroups, & CIFAR10 for different positive & negative class definitions. Bold denotes a *shifted task*'s best performing method. For *all* shifted tasks, our three methods – denoted with [†] – statistically outperformed PUc and nnPU* based on a paired t-test ($p < 0.01$). Each dataset's first three experiments have identical negative (N) & positive-test ($P_{test}$) class definitions. Positive train ($P_{train}$) specified as "$P_{test}$" denotes no bias. Additional shifted tasks (with result standard deviations) are in the supplemental materials.

| | N | $P_{test}$ | $P_{train}$ | $\pi_{tr}$ | $\pi_{te}$ | Two-Step (PU2) | | | Baselines | | Ref. |
| --- | --- | --- | --- | --- | --- | --- | --- | --- | --- | --- | --- |
| | | | | | | $PURR^{\dagger}$ | $aPNU^{\dagger}$ | $wUU^{\dagger}$ | PUc | nnPU* | $PN_{te}$ |
| **MNIST** | 0, 2, 4, 6, 8 | 1, 3, 5, 7, 9 | $P_{test}$ | 0.5 | 0.5 | 10.0 | 10.0 | 11.6 | 8.6 | 5.5 | ↑ |
| | | | 7, 9 | 0.5 | 0.5 | 9.4 | **7.1** | 8.3 | 26.8 | 35.1 | 2.8 |
| | | | | 0.29 | 0.5 | 6.8 | **5.3** | 6.0 | 29.2 | 36.7 | ↓ |
| | 0, 2 | 5, 7 | 1, 3 | 0.5 | 0.5 | 4.0 | 3.6 | **3.1** | 17.1 | 30.9 | 1.1 |
| **20 News.** | sci, soc, talk | alt, comp, misc, rec | $P_{test}$ | 0.56 | 0.56 | 15.4 | 14.9 | 16.7 | 14.9 | 14.1 | ↑ |
| | | | misc, rec | 0.56 | 0.56 | 17.5 | **13.5** | 15.1 | 23.9 | 28.8 | 10.5 |
| | | | | 0.37 | 0.56 | 13.9 | **12.8** | 14.3 | 28.9 | 28.8 | ↓ |
| | misc, rec | soc, talk | alt, comp | 0.55 | 0.46 | 5.9 | 7.1 | **5.6** | 18.5 | 35.3 | 2.1 |
| **CIFAR10** | Bird, Cat, Deer, Dog, Frog, Horse | Plane, Auto, Ship, Truck | $P_{test}$ | 0.4 | 0.4 | 14.1 | 14.2 | 15.5 | 13.8 | 12.3 | ↑ |
| | | | Plane | 0.4 | 0.4 | **13.8** | 14.5 | 15.1 | 20.6 | 27.4 | 9.8 |
| | | | | 0.14 | 0.4 | 12.1 | **11.9** | 12.4 | 26.7 | 26.7 | ↓ |
| | Deer, Horse | Plane, Auto | Cat, Dog | 0.5 | 0.5 | 14.1 | 14.9 | **11.2** | 33.1 | 47.5 | 7.7 |

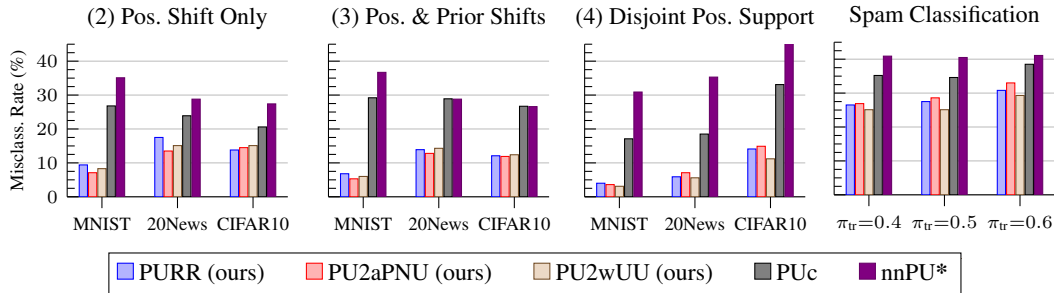

Figure 2: Mean inductive misclassification rate over 100 trials on the MNIST, 20 News., CIFAR10, & TREC spam datasets for our methods & baselines. Each numbered plot (i.e., 2–4) corresponds to one experimental shift task in Table 1. Spam classification experiments are detailed in Section 7.3.

By default $\pi_{tr} = \pi_{te}$; in the prior shift and disjoint support experiments (rows 3 and 4), $\pi_{tr}$ equals $P_{train}$'s true prior w.r.t. $P_{train} \sqcup N$.

**Analysis** Results are shown in Table 1 and Figure 2. On *unshifted* data (row 1 for each dataset), baselines PUc and nnPU* slightly outperformed our methods, which shows that PUc's architecture is sufficiently expressive. In contrast, on *shifted* data (rows 2–4 for each dataset), our methods' performance generally improved while both PUc's and nnPU*'s performance always degraded. This performance divergence demonstrates our methods' algorithmic advantage. In fact for all shifted tasks, our methods always outperformed PUc and nnPU* according to a paired t-test ($p < 0.01$). For partially disjoint positive supports (rows 2 and 3 for each dataset), PU2aPNU was the top performer for five of six setups (PURR was top on the other). This pattern reversed for fully disjoint supports (row 4) where PU2aPNU always lagged PU2wUU; this is expected as explained in Section 5.

Reducing $\pi_{tr}$ always improved our algorithms' performance and degraded PUc's. A smaller prior enables easier identification of $\mathcal{X}_{tr-u}$'s negative examples and in turn a more accurate estimation of $\mathcal{X}_{te-u}$'s negative risk. In contrast, importance weighting is most accurate in the absence of bias (see row 1 for each dataset). Any shift increases density estimation's (and by extension PUc's) inaccuracy.

Table 2: Mean inductive misclassification rate (%) over 100 trials for spam adversarial drift. Our methods – PURR, PU2wUU, and PU2aPNU – outperformed PUc & nnPU* based on a 1% paired t-test. Each result's standard deviation appears in supplemental Table 14.

| Train Set | | Test Set | | $\pi_{tr}$ | $\pi_{te}$ | | Two-Step (PU2) | | Baselines | | Ref. |
|---|---|---|---|---|---|---|---|---|---|---|---|
| Pos. | Neg. | Pos. | Neg. | | | PURR | aPNU | wUU | PUc | nnPU* | $PN_{te}$ |
| 2005 Spam | 2005 Ham | 2007 Spam | 2007 Ham | 0.4 | 0.5 | 26.5 | 26.9 | **25.1** | 35.2 | 40.9 | ↑ |
| | | | | 0.5 | 0.5 | 27.5 | 28.6 | **25.1** | 34.6 | 40.5 | 0.6 |
| | | | | 0.6 | 0.5 | 30.8 | 33.0 | **29.3** | 38.5 | 41.1 | ↓ |

nnPU* outperformed both PUc and our methods when there was no bias. This is expected. If an algorithm searches for non-existent phenomena, any additional patterns found will not generalize.

### 7.3 Case Study: Arbitrary Adversarial Concept Drift

PU learning has been applied to multiple adversarial domains including opinion spam [3, 16, 17, 18]. We use spam classification as a vehicle to test our methods in an adversarial setting by considering two different TREC email spam datasets – training on TREC05 and evaluating on TREC07. Spam – the positive class – evolves quickly over time, but the two datasets' ham emails are also quite different: TREC05 relies on Enron emails while TREC07 contains mostly emails from a university server. Thus, this represents a more challenging, realistic setting where Eq. (7)'s assumption does not hold.

Table 2 and Figure 2 show that our methods outperformed PUc and nnPU* according to a 1% paired t-test across three training priors ($\pi_{tr}$). PU2wUU was the top-performer as $\hat{\sigma}$ accurately labeled $\mathcal{X}_{tr\text{-}u}$, yielding a strong surrogate negative set. PU2aPNU performed slightly worse than PU2wUU as the significant adversarial concept drift greatly limited $\mathcal{X}_p$'s value. Overall, these experiments show that our aPU setting arises in real-world domains. All of our methods handled large positive shifts better than prior work, even in realistic cases where the negative class also shifts.

### 7.4 Discussion

Our two-step methods assume asymptotic consistency for $\widetilde{\mathcal{X}}_n$ in step #1, but finite training data ensures a non-consistent evaluation setting. Nonetheless, either PU2aPNU or PU2wUU was the top performer in all but one experiment in this section.[5] Supplemental Section E.7 includes additional experiments where we further stress our two-step methods by forcing $\hat{\sigma}$ away from our posterior estimate. Even under those deleterious step #1 conditions, our two-step learners are robust.

Conventional wisdom suggests that joint method PURR should outperform pipeline approaches. This intuition breaks down in our case because PURR, with its three risk decompositions, is strictly harder to optimize than wUU, aPNU, abs-PU, and nnPU – all of which have a single decomposition. This harder optimization can lead to worse accuracy compared to the two-step methods, especially on easier problems (e.g., MNIST), where each step can be solved accurately on its own.

For completeness, suppl. Section E.5 compares our methods to bPU selection bias method PUSB [13]. Our algorithms generally outperformed PUSB on data specifically tuned for their method even after accounting for the differing unlabeled sets. Those experiments indicate that PUSB's underlying assumption entails only a small data shift and further point to potential PUSB learning brittleness.

## 8 Conclusions

We examined arbitrary-positive, unlabeled (aPU) learning, where the labeled-positive and target-positive distributions may be arbitrarily different. A (nearly) fixed negative class-distribution allows us to train accurate classifiers without any labeled data from the target distribution (i.e., disjoint positive supports). Empirical results on real-world data above and in the supplementals show that our methods are still robust in the realistic case of some negative shift. Future work seeks a less restrictive yet statistically-sound replacement assumption of a fixed negative class-conditional distribution.

# 9 Broader Impact

The algorithms proposed in this work are general and could be applied to many different applications. Forecasting the broader impact of work like this is challenging and generally inaccurate. With that caveat, we discuss potential impacts based on possible applications.

The case study on email spam suggests that our methods may be useful in adversarial domains, such as the detection of fraud, malware, network intrusion, distributed denial of service (DDoS) attacks, and many types of spam. In these settings, one class (e.g., spam) evolves quickly as attackers try to evade detection. For many of these domains, improved classifiers would benefit society by reducing spam and fraud. However, for domains such as facial recognition, improved robustness could lead to reduced privacy and other societal harms. See Albert et al. [41] for an extensive discussion of the politics of adversarial machine learning.

In other domains, such as epidemiological analysis and land-cover classification, our work may lead to new or better models by reducing the need for labeled data and relaxing the SCAR assumption. As detailed in Section 1, only recently has the PU SCAR barrier been broken [12, 13, 14]. aPU learning pushes PU learning's positive-shift boundary to a new extreme. We hope this paper will enable PU learning to be applied in domains where existing bPU\PU methods are impractical. This could also benefit society if used responsibly, with experts performing proper model validation and vetting risks. Careful model validation is especially important when labeled data is limited and biased.

## Acknowledgments and Disclosure of Funding

This work was supported by a grant from the Air Force Research Laboratory and the Defense Advanced Research Projects Agency (DARPA) – agreement number FA8750-16-C-0166, subcontract K001892-00-S05.

This work benefited from access to the University of Oregon high performance computer, Talapas.

## Footnotes

[1]Each theorem's definition of "bounded" loss appears in the associated proof. See the supplemental materials.

[2]"Unlabeled-unlabeled learning" denotes the two unlabeled sets and is different from UU learning in [28, 29].

[3]Our implementation is publicly available at: https://github.com/ZaydH/arbitrary_pu.

[4]The PUc implementation was provided by Sakai and Shimizu [14] via personal correspondence.

[5]Supplemental Sections E.2 and E.4 enumerate multiple empirical setups where PURR is the top performer.

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
