[Supplementary Material]

# Learning from Positive and Unlabeled Data with Arbitrary Positive Shift

## Supplemental Materials

## A Nomenclature

Table 3: aPU nomenclature reference

| | |
|---|---|
| PN | Positive-negative learning, i.e., ordinary supervised classification |
| PU | Positive-unlabeled learning |
| NU | Negative-unlabeled learning |
| uPU | Unbiased Positive-Unlabeled risk estimator from [20]. See Section 2 |
| nnPU | Non-negative Positive-Unlabeled risk estimator from [8]. See Section 2 |
| abs-PU | Our Absolute-value Positive-Unlabeled risk estimator. See Section 3 |
| bPU | Biased-positive, unlabeled learning where the labeled-positive set is not representative of the target positive class. bPU algorithm categories include sample selection bias [12, 13] and covariate shift methods [14] |
| aPU | Proposed in this work, arbitrary-positive, unlabeled learning generalizes bPU learning where the positive training data may be arbitrarily different from the target application's positive-class distribution |
| PUc | Positive-Unlabeled Covariate shift algorithm from [14]. See Section 4.1 |
| PU2wUU | Our Positive-Unlabeled to Weighted Unlabeled-Unlabeled (two-step) aPU learner. See Section 5 |
| PU2aPNU | Our Positive-Unlabeled to Arbitrary-Positive, Negative, Unlabeled (two-step) aPU learner. See Section 5 |
| PURR | Our Positive-Unlabeled Recursive Risk (one-step) aPU estimator. See Section 6 |
| abs-PU | Our Absolute-value Positive-Unlabeled risk estimator. See Section 3 |
| nnPU* | Version of nnPU used as an empirical baseline. nnPU* considers two classifiers – one trained with $\mathcal{X}_{\text{te-u}}$ as the unlabeled set and the other trained with $\mathcal{X}_{\text{tr-u}} \cup \mathcal{X}_{\text{te-u}}$ as the unlabeled set – and reports whichever configuration performed better. See Section 7 |
| abs-PU* | Baseline equivalent of nnPU* except risk estimator $\widehat{R}_{\text{abs-PU}}(g)$ is used instead of $\widehat{R}_{\text{nnPU}}(g)$. See Section E.6.2 |
| $X$ | Covariate where $X \in \mathbb{R}^d$ |
| $Y$ | Dependent random variable, i.e., label, where $Y \in \{\pm 1\}$ |
| $\hat{y}$ | Predicted label $\hat{y} \in \{\pm 1\}$ |
| $g$ | Decision function, $g : \mathbb{R}^d \to \mathbb{R}$ |
| $\theta$ | Parameter(s) of decision function $g$ |
| $\mathcal{G}$ | Real-valued decision function hypothesis class, i.e., $g \in \mathcal{G}$ |
| $\ell$ | Loss function, $\ell : \mathbb{R} \to \mathbb{R}_{\geq 0}$ |
| $p_{\mathcal{T}}(x, y)$ | Joint distribution, where $\mathcal{T} \in \{\text{tr}, \text{te}\}$ for train and test resp. |
| $\pi_{\mathcal{T}}$ | Positive-class prior probability, $\pi_{\mathcal{T}} := p_{\mathcal{T}}(Y = +1)$ where $\mathcal{T} \in \{\text{tr}, \text{te}\}$ for train & test resp. |
| $p_{\mathcal{T}\text{-p}}(x)$ | Positive class-conditional $p_{\mathcal{T}\text{-p}}(x) := p_{\mathcal{T}}(x|Y = +1)$ where $\mathcal{T} \in \{\text{tr}, \text{te}\}$ for train & test resp. |
| $p_{\mathcal{T}\text{-n}}(x)$ | Negative class-conditional $p_{\mathcal{T}\text{-n}}(x) := p_{\mathcal{T}}(x|Y = -1)$ where $\mathcal{T} \in \{\text{tr}, \text{te}\}$ for train & test resp. |
| $p_{\mathcal{T}\text{-u}}(x)$ | Marginal distribution where $p_{\mathcal{T}\text{-u}}(x) := p_{\mathcal{T}}(x)$ where $\mathcal{T} \in \{\text{tr}, \text{te}\}$ for train and test resp. |
| $\mathcal{X}_{\text{p}}$ | Labeled (positive) dataset, i.e., $\mathcal{X}_{\text{p}} \overset{\text{i.i.d.}}{\sim} p_{\text{tr-p}}(x)$ |
| $\mathcal{X}_{\text{tr-u}}$ | Unlabeled dataset sampled from the *training* marginal distribution, i.e., $\mathcal{X}_{\text{tr-u}} \overset{\text{i.i.d.}}{\sim} p_{\text{tr-u}}(x)$ |
| $\mathcal{X}_{\text{te-u}}$ | Unlabeled dataset sampled from the *test* marginal distribution, i.e., $\mathcal{X}_{\text{te-u}} \overset{\text{i.i.d.}}{\sim} p_{\text{te-u}}(x)$ |
| $\hat{\sigma}$ | Probabilistic classifier, $\hat{\sigma} : \mathbb{R}^d \to [0, 1]$ that approximates $p_{\text{tr}}(Y = -1|x)$ |
| $\widehat{\Sigma}$ | Function class containing $\hat{\sigma}$ |
| $\mathcal{X}_{\text{n}}$ | Labeled negative dataset. In PU learning, $\mathcal{X}_{\text{n}} = \emptyset$ |
| $\widetilde{\mathcal{X}}_{\text{n}}$ | Surrogate negative set formed by reweighting $\mathcal{X}_{\text{tr-u}}$ by $\hat{\sigma}$ |
| $R(g)$ | Risk, i.e., expected loss, for decision function $g$ and loss $\ell$, i.e., $R(g) := \mathbb{E}_{(X,Y) \sim p(x,y)}[\ell(Y g(X))]$ |
| $\widehat{R}(g)$ | Empirical estimate of risk $R(g)$ |

Table 3: aPU nomenclature reference (continued)

| | |
|---|---|
| $\widehat{R}_{\mathcal{D}}^{\hat{y}}(g)$ | Empirical risk when predicting label $\hat{y} \in \{\pm 1\}$ on data sampled from some distribution, $p_{\mathcal{D}}(x)$. See Section 2 |
| $\ddot{R}_{\mathrm{n}}^{\hat{y}}(g)$ | Labeled negative risk with absolute-value correction. See Eq. (5) in Section 3 |
| $\widetilde{R}_{\mathrm{n\text{-}u}}^{\hat{y}}(g)$ | Surrogate negative risk formed by weighting unlabeled set $\mathcal{X}_{\mathrm{tr\text{-}u}}$ by probabilistic classifier $\hat{\sigma}$ where $\widetilde{R}_{\mathrm{n\text{-}u}}^{\hat{y}}(g) := \frac{1}{n_{\mathrm{tr\text{-}u}}} \sum_{x_i \in \mathcal{X}_{\mathrm{tr\text{-}u}}} \frac{\hat{\sigma}(x_i)\ell(\hat{y}g(x_i))}{1-\pi_{\mathrm{tr}}}$ |
| $w(x)$ | Covariate shift importance function based on density-ratio estimation where $w(x) := \frac{p_{\mathrm{te\text{-}u}}(x)}{p_{\mathrm{tr\text{-}u}}(x)}$ |
| $n_{\mathrm{p}}$ | Size of the labeled (positive) dataset, i.e., $n_{\mathrm{p}} := |\mathcal{X}_{\mathrm{p}}|$ |
| $n_{\mathrm{tr\text{-}u}}$ | Size of the unlabeled *training* dataset, i.e., $n_{\mathrm{tr\text{-}u}} := |\mathcal{X}_{\mathrm{tr\text{-}u}}|$ |
| $n_{\mathrm{te\text{-}u}}$ | Size of the unlabeled *test* dataset, i.e., $n_{\mathrm{te\text{-}u}} := |\mathcal{X}_{\mathrm{te\text{-}u}}|$ |
| $n_{\mathrm{Test}}$ | Size of the inductive test set |
| $\mathcal{A}$ | Learning or optimization algorithm |
| $\eta$ | Learning rate hyperparameter, $\eta > 0$ |
| $\lambda$ | Weight decay hyperparameter, $\lambda \geq 0$ |
| $\gamma$ | Non-negative gradient attenuator hyperparameter $\gamma \in (0, 1]$. This hyperparameter is ignored when absolute-value correction is used. |
| $\mathcal{N}(\boldsymbol{\mu}, \mathbf{I}_m)$ | Multivariate Gaussian (normal) distribution with mean $\boldsymbol{\mu}$ and $m$-dimensional identity covariance. See Section E.1 |
| $[a]_+$ | $:= \max\{0, a\}$. See Section C.2 |
| $\lfloor a \rceil$ | Rounds $a \in \mathbb{R}$ to the nearest integer. See Section E.7 |

# B  Proofs

## B.1  Proof of Theorem 1

*Proof.* Mild assumptions are made about the behavior of the loss and decision functions; the following conditions match those assumed by Kiryo et al. [8]. Define loss function $\ell$ as *bounded* over some class of real-valued functions $\mathcal{G}$ (where $g \in \mathcal{G}$) when the following conditions both hold:

1. $\exists C_g > 0$ such that $\sup_{g \in \mathcal{G}} \|g\|_\infty \leq C_g$

2. $\exists C_\ell > 0$ such that $\sup_{|t| \leq C_g} \max_{\hat{y} \in \{\pm 1\}} \ell(\hat{y}t) \leq C_\ell$ .

du Plessis et al. [20] show that

$$(1 - \pi)R_{\mathrm{n}}^{\hat{y}}(g) = R_{\mathrm{u}}^{\hat{y}}(g) - \pi R_{\mathrm{p}}^{\hat{y}}(g). \tag{15}$$

Consider the labeled negative-valued risk estimator with absolute-value correction

$$\ddot{R}_{\mathrm{n}}^{\hat{y}}(g) = \left| \widehat{R}_{\mathrm{n}}^{\hat{y}}(g) \right|. \tag{16}$$

An estimator, $\hat{\theta}_n$, over $n$ samples is consistent w.r.t. parameter $\theta$ if for all $\epsilon > 0$ it holds that

$$\lim_{n \to \infty} \mathrm{Pr}\left[\left|\hat{\theta}_n - \theta\right| \geq \epsilon\right] = 0.$$

Let estimator $\hat{Y} = \sum_{i=1}^k \beta_i \hat{\theta}_{(i)}$ be the weighted sum of $k$ consistent estimators with each constant $\beta_i \neq 0$. Let $\epsilon > 0$ be an arbitrary positive constant. If each $\hat{\theta}_{(i)}$ converges to within $\frac{\epsilon}{k|\beta_i|} > 0$ of $\theta_{(i)} \geq 0$, then $\hat{Y}$ converges to within $\epsilon$ of $\sum_{i=1}^k \beta_i \theta_{(i)}$. Therefore, to prove the consistency of $\ddot{R}_{\mathrm{n}}^{\hat{y}}(g)$ in Eq. (16), it is sufficient to show that each of its individual terms is consistent.

Both $\widehat{R}_{\mathrm{p}}^{\hat{y}}(g)$ and $\widehat{R}_{\mathrm{u}}^{\hat{y}}(g)$ are empirically estimated directly from a training data set. Let $\mathcal{D} \in \{\mathrm{p}, \mathrm{u}\}$ and $\mathcal{X}_{\mathcal{D}} \overset{\mathrm{i.i.d.}}{\sim} p_{\mathcal{D}}(x)$. For each (independent) $X \sim p_{\mathcal{D}}(x)$, $\ell(\hat{y}g(X))$ is an unbiased estimate of $R_{\mathcal{D}}^{\hat{y}}(g)$. In addition, $\ell(\hat{y}g(X)) < C_\ell < \infty$ implies that $\mathrm{Var}(\ell(\hat{y}g(X))) < \infty$. By Chebyshev's Inequality, $\widehat{R}_{\mathcal{D}}^{\hat{y}}(g)$ is consistent as

$$\lim_{|\mathcal{X}| \to \infty} \mathrm{Pr}\left[\left|\frac{1}{|\mathcal{X}|} \sum_{x_i \in \mathcal{X}} \left(\ell(\hat{y}g(x_i))\right) - R_{\mathcal{D}}^{\hat{y}}(g)\right| \geq \epsilon\right] < \frac{\mathrm{Var}(\ell(\hat{y}g(X)))}{|\mathcal{X}|\epsilon^2} = 0.$$

Since $\widehat{R}_{\mathrm{n}}^{\hat{y}}(g)$ is the weighted sum of consistent estimators, it is consistent as $n = \min\{n_{\mathrm{p}}, n_{\mathrm{u}}\} \to \infty$.

To show $\ddot{R}_{\mathrm{n}}^{\hat{y}}(g)$ is consistent, it suffices to show that

$$\lim_{n \to \infty} \mathrm{Pr}\left[\left|\ddot{R}_{\mathrm{n}}^{\hat{y}}(g) - R_{\mathrm{n}}^{\hat{y}}(g)\right| \geq \epsilon\right] = 0.$$

Because $\widehat{R}_{\mathrm{n}}^{\hat{y}}(g)$ is consistent, then as $n \to \infty$ it holds that $\widehat{R}_{\mathrm{n}}^{\hat{y}}(g) - \epsilon \leq R_{\mathrm{n}}^{\hat{y}}(g) \leq \widehat{R}_{\mathrm{n}}^{\hat{y}}(g) + \epsilon$. When $\widehat{R}_{\mathrm{n}}^{\hat{y}}(g) \geq R_{\mathrm{n}}^{\hat{y}}(g) \geq 0$, then $\ddot{R}_{\mathrm{n}}^{\hat{y}}(g) = \widehat{R}_{\mathrm{n}}^{\hat{y}}(g)$ (i.e., absolute value has no effect) so

$$0 \leq \ddot{R}_{\mathrm{n}}^{\hat{y}}(g) - R_{\mathrm{n}}^{\hat{y}}(g) \leq \epsilon.$$

Consider the alternate possibility where $\widehat{R}_{\mathrm{n}}^{\hat{y}}(g) < R_{\mathrm{n}}^{\hat{y}}(g)$. If $\widehat{R}_{\mathrm{n}}^{\hat{y}}(g) \geq 0$ or $R_{\mathrm{n}}^{\hat{y}}(g) = 0$, then absolute-value correction again has no effect on the estimation error (i.e., remains $\leq \epsilon$). Lastly, when $\widehat{R}_{\mathrm{n}}^{\hat{y}}(g) < 0$ and $R_{\mathrm{n}}^{\hat{y}}(g) > 0$, the estimation error strictly decreases as

$$\begin{aligned}
\mathrm{err}_{\hat{R}} &= \left|\widehat{R}_{\mathrm{n}}^{\hat{y}}(g) - R_{\mathrm{n}}^{\hat{y}}(g)\right| \\
&= -\widehat{R}_{\mathrm{n}}^{\hat{y}}(g) + R_{\mathrm{n}}^{\hat{y}}(g) && \text{Since } \widehat{R}_{\mathrm{n}}^{\hat{y}}(g) < 0 \text{ and } R_{\mathrm{n}}^{\hat{y}}(g) > 0 \\
&= \left|\widehat{R}_{\mathrm{n}}^{\hat{y}}(g)\right| + R_{\mathrm{n}}^{\hat{y}}(g) && \text{Again since } \widehat{R}_{\mathrm{n}}^{\hat{y}}(g) < 0 \\
&= \ddot{R}_{\mathrm{n}}^{\hat{y}}(g) + R_{\mathrm{n}}^{\hat{y}}(g) < \epsilon
\end{aligned}$$

so

$$\begin{aligned}
\mathrm{err}_{\ddot{R}} &= \left|\left|\widehat{R}_{\mathrm{n}}^{\hat{y}}(g)\right| - R_{\mathrm{n}}^{\hat{y}}(g)\right| \\
&=: \left|\ddot{R}_{\mathrm{n}}^{\hat{y}}(g) - R_{\mathrm{n}}^{\hat{y}}(g)\right| \\
&< \ddot{R}_{\mathrm{n}}^{\hat{y}}(g) + R_{\mathrm{n}}^{\hat{y}}(g) < \epsilon && \text{Since } \widehat{R}_{\mathrm{n}}^{\hat{y}}(g) < 0 \text{ and } R_{\mathrm{n}}^{\hat{y}}(g) > 0. \quad (17)
\end{aligned}$$

The above shows that as $n \to \infty$, it always holds that $\left|\ddot{R}_{\mathrm{n}}^{\hat{y}}(g) - R_{\mathrm{n}}^{\hat{y}}(g)\right| \leq \epsilon$ for arbitrary $\epsilon > 0$ making $\ddot{R}_{\mathrm{n}}^{\hat{y}}(g)$ consistent. □

## B.2 Proof of Theorem 2

*Proof.* Consider first the case that $\hat{\sigma}(x) = p_{\mathrm{tr}}(Y = -1|x)$:

$$\begin{aligned}
\mathbb{E}_{\mathcal{X}_{\mathrm{tr\text{-}u}} \overset{\text{i.i.d.}}{\sim} p_{\mathrm{tr\text{-}u}}(x)}\left[\widetilde{R}_{\mathrm{n\text{-}u}}^{\hat{y}}(g)\right] &= \mathbb{E}_{\mathcal{X}_{\mathrm{tr\text{-}u}} \overset{\text{i.i.d.}}{\sim} p_{\mathrm{tr\text{-}u}}(x)}\left[\frac{1}{n_{\mathrm{tr\text{-}u}}} \sum_{X_i \in \mathcal{X}_{\mathrm{tr\text{-}u}}} \frac{\ell(\hat{y}g(X_i))\hat{\sigma}(X_i)}{1 - \pi_{\mathrm{tr}}}\right] \\
&= \frac{1}{n_{\mathrm{tr\text{-}u}}} \sum_{i=1}^{n_{\mathrm{tr\text{-}u}}} \mathbb{E}_{X \sim p_{\mathrm{tr\text{-}u}}(x)}\left[\frac{\ell(\hat{y}g(X))\hat{\sigma}(X)}{1 - \pi_{\mathrm{tr}}}\right] && \text{Linearity of expectation} \\
&= \mathbb{E}_{X \sim p_{\mathrm{tr\text{-}u}}(x)}\left[\frac{\ell(\hat{y}g(X))\hat{\sigma}(X)}{1 - \pi_{\mathrm{tr}}}\right] \\
&= \mathbb{E}_{X \sim p_{\mathrm{tr\text{-}u}}(x)}\left[\frac{\ell(\hat{y}g(X))p_{\mathrm{tr}}(Y = -1|X)}{p_{\mathrm{tr}}(Y = -1)}\right] \\
&= \int_x \ell(\hat{y}g(x)) \frac{p_{\mathrm{tr}}(Y = -1|x)p_{\mathrm{tr\text{-}u}}(x)}{p_{\mathrm{tr}}(Y = -1)} \\
&= \mathbb{E}_{X \sim p_{\mathrm{tr\text{-}n}}(x)}[\ell(\hat{y}g(X))] && \text{Bayes' Rule} \\
&=: R_{\mathrm{tr\text{-}n}}^{\hat{y}}(g),
\end{aligned}$$

satisfying the definition of unbiased.

Next we consider whether $\widetilde{R}_{\mathrm{n\text{-}u}}^{\hat{y}}(g)$ is a consistent estimator of $R_{\mathrm{n}}^{\hat{y}}(g)$. For the complete definition of PAC learnability that we use here, see [42]. We provide a brief sketch of the definition below.

We assume that true posterior distribution, $p_{\mathrm{tr}}(Y = -1|x)$ is in some concept class $\mathcal{C}$ of functions — i.e., *concepts* — mapping $\mathbb{R}^d$ to $[0, 1]$. Let $\hat{\sigma}_{\mathcal{S}} \in \widehat{\Sigma}$ be the hypothesis selected by learning algorithm $\mathcal{A}$ after being provided a training sample $\mathcal{S}$ of size $n = \min\{n_{\mathrm{p}}, n_{\mathrm{tr\text{-}u}}\}$.[6] Consider the *realizable* setting so $\mathcal{C}$'s PAC learnability entails that for all

$\epsilon, \delta > 0$, there exists an $n'$ such that for all $n > n'$,

$$\Pr\left[\mathbb{E}_{X \sim p_{\text{tr-u}}(x)}[|\hat{\sigma}_{\mathcal{S}}(X) - p_{\text{tr}}(Y = -1|X)|] > \epsilon\right] < \delta. \tag{18}$$

Therefore, as $n \to \infty$, $\hat{\sigma}$'s expected (absolute) error w.r.t. $p_{\text{tr}}(Y\!=\!-1|x)$ decreases to 0 making $\widetilde{R}^{\hat{y}}_{\text{n-u}}(g)$ asymptotically unbiased. To demonstrate consistency, it is necessary to show that for all $\epsilon > 0$:

$$\lim_{n \to \infty} \Pr\left[\left|\widetilde{R}^{\hat{y}}_{\text{n-u}}(g) - R^{\hat{y}}_{\text{tr-n}}(g)\right| > \epsilon\right] = 0.$$

Let $\sup_{|t| \leq \|g\|_{\infty}} \ell(\hat{y}t) \leq C_{\ell}$, where $\|g\|_{\infty}$ is the Chebyshev norm of $g$ for $x \in \mathbb{R}^d$. Bounding the loss's magnitude bounds the variance when estimating the surrogate negative risk of $X \sim p_{\text{tr-u}}(x)$ such that $\frac{1}{(1 - \pi_{\text{tr}})^2}\text{Var}(\hat{\sigma}(X)\ell(\hat{y}g(X))) \leq C_{\text{var}}$ where $C_{\text{var}} \in \mathbb{R}_{\geq 0}$ and $\pi_{\text{tr}} \in [0, 1)$.

Since $\widetilde{R}^{\hat{y}}_{\text{n-u}}(g)$ is asymptotically unbiased, then from Chebyshev's inequality for $\epsilon > 0$:

$$\lim_{n \to \infty} \Pr\left[\left|\widetilde{R}^{\hat{y}}_{\text{n-u}}(g) - R^{\hat{y}}_{\text{tr-n}}(g)\right| \geq \epsilon\right] \leq \frac{\text{Var}(\widetilde{R}^{\hat{y}}_{\text{n-u}}(g))}{\epsilon^2}$$

$$= \frac{1}{(1 - \pi_{\text{tr}})^2\epsilon^2} \sum_{i=1}^{n_{\text{tr-u}}} \text{Var}\left(\frac{\hat{\sigma}(X)\ell(\hat{y}g(X))}{n_{\text{tr-u}}}\right) \quad \text{Linearity of independent r.v. var.}$$

$$\leq \frac{n_{\text{tr-u}}C_{\text{var}}}{n_{\text{tr-u}}^2\epsilon^2}$$

$$= 0 \qquad\qquad\qquad\qquad\qquad\qquad\qquad\qquad\qquad\quad \text{L'Hôpital's Rule.}$$

$\square$

## B.3   Proof Regarding Estimating $\pi_{\text{te}}$

We are not aware of an existing technique to directly estimate the test distribution's positive prior $\pi_{\text{te}}$ given only $\mathcal{X}_{\text{p}}$, $\mathcal{X}_{\text{tr-u}}$, and $\mathcal{X}_{\text{te-u}}$. We propose the following that uses an additional classifier.

**Theorem 4.** *Define* $\mathcal{X}_{\text{u}} := \{x_i\}_{i=1}^{n_{\text{u}}} \overset{\text{i.i.d.}}{\sim} p_{\text{u}}(x)$. *Let* $\mathcal{X}_{\text{n}} = \{x_i \in \mathcal{X}_{\text{u}} : Q_i = 1\}$ *be a set where* $Q_i$ *is a Bernoulli random variable with probability of success* $q_i = p(Y = -1|x_i)$. *Then* $\mathcal{X}_{\text{n}}$ *is a SCAR sample w.r.t. negative class-conditional distribution* $p_{\text{n}}(x) = p(x|Y\!=\!-1)$.

*Proof.* By Bayes' Rule

$$p_{\text{n}}(x) \propto p(Y\!=\!-1|x)p_{\text{u}}(x)$$

Each $x_i \in \mathcal{X}_{\text{u}}$ is sampled from $p_{\text{u}}(x)$. By including $x_i$ in $\mathcal{X}_{\text{n}}$ only if $Q_i = 1$, then $x_i$'s effective sampling probability is $p(Y\!=\!-1|x_i)p(x)$. Bayes' Rule includes prior inverse $\frac{1}{1-\pi}$, where $\pi = p(Y\!=\!+1)$; this constant scalar can be ignored since it does not change whether $\mathcal{X}_{\text{n}}$ is unbiased, i.e., it does not affect relative probability. $\square$

**Commentary**   Theorem 4 states the property generally, but consider it over aPU's training distribution. Probabilistic classifier $\hat{\sigma}$ is used as a surrogate for $p_{\text{tr}}(Y\!=\!-1|x)$. Rather than *soft* weighting the samples like in Theorem 2's proof, sample inclusion in the negative set is a *hard* "in-or-out" decision. This does not change the sample's statistical properties, but it allows us to create an unweighted negative set, we denote $\mathcal{X}_{\text{tr-n}}$.

By Eq. (7)'s assumption, $\mathcal{X}_{\text{tr-n}}$ is representative of samples from the negative class-conditional distribution $p_{\text{n}}(x) = p_{\text{tr-n}}(x) = p_{\text{te-n}}(x)$. Given a representative labeled set from the *test distribution*, well-known positive-unlabeled prior estimation techniques [21, 22] can be used without modification using $\mathcal{X}_{\text{tr-n}}$ and $\mathcal{X}_{\text{te-u}}$. Be aware that these PU prior estimation methods would return the negative-class's prior, $p_{\text{te}}(Y\!=\!-1)$, while our risk estimators use the positive class's prior, $\pi_{\text{te}} = 1 - p_{\text{te}}(Y\!=\!-1)$.

We provide empirical results regarding the effect of inaccurate prior estimation's in Section E.9.

### B.4 Proof of Theorem 3

The definition of "bounded loss" is identical to the proof of Theorem 1.

*Proof.* Consider first whether PURR is unbiased. du Plessis et al. [20] observe that the negative labeled risk can be found via decomposition where

$$(1 - \pi)R_n^{\hat{y}}(g) = R_u^{\hat{y}}(g) - \pi R_p^{\hat{y}}(g). \tag{19}$$

The positive labeled risk similarly decomposes as

$$\pi R_p^{\hat{y}}(g) = R_u^{\hat{y}}(g) - (1 - \pi)R_n^{\hat{y}}(g). \tag{20}$$

Applying these decompositions along with Eq. (7)'s assumption yields an unbiased version of PURR:

$$\widehat{R}_{\text{uPURR}}(g) = \widehat{R}_{\text{te-u}}^+(g) - (1 - \pi_{\text{te}}) \underbrace{\frac{\widehat{R}_{\text{tr-u}}^+(g) - \pi_{\text{tr}}\widehat{R}_{\text{tr-p}}^+(g)}{1 - \pi_{\text{tr}}}}_{\widehat{R}_{\text{te-n}}^+(g)} + (1 - \pi_{\text{te}}) \underbrace{\frac{\widehat{R}_{\text{tr-u}}^-(g) - \pi_{\text{tr}}\widehat{R}_{\text{tr-p}}^-(g)}{1 - \pi_{\text{tr}}}}_{\widehat{R}_{\text{te-n}}^-(g)}. \tag{21}$$

Since $\forall_t \ell(t) \geq 0$, it always holds that labeled risk $R_{\mathcal{D}}^{\hat{y}}(g) \geq 0$. When using risk decomposition (i.e., Eqs. (19) and (20)) to empirically estimate a labeled risk, it can occur that $\widehat{R}_{\mathcal{D}}^{\hat{y}}(g) < 0$. Absolute-value correction addresses these obviously implausible risk estimates. The unrolled definition of the PURR risk estimator with absolute-value correction is:

$$\widehat{R}_{\text{PURR}}(g) = \left| \widehat{R}_{\text{te-u}}^+(g) - (1 - \pi_{\text{te}}) \left| \underbrace{\frac{\widehat{R}_{\text{tr-u}}^+(g) - \pi_{\text{tr}}\widehat{R}_{\text{tr-p}}^+(g)}{1 - \pi_{\text{tr}}}}_{\widehat{R}_{\text{te-n}}^+(g)} \right| \right| + (1 - \pi_{\text{te}}) \left| \underbrace{\frac{\widehat{R}_{\text{tr-u}}^-(g) - \pi_{\text{tr}}\widehat{R}_{\text{tr-p}}^-(g)}{1 - \pi_{\text{tr}}}}_{\widehat{R}_{\text{te-n}}^-(g)} \right|. \tag{22}$$

Clearly, $\widehat{R}_{\text{PURR}}(g) \geq \widehat{R}_{\text{uPURR}}(g)$. For $\widehat{R}_{\text{PURR}}(g)$ to be unbiased, equality must strictly hold. This only occurs if the absolute-value is never needed, i.e., has probability 0 of occurring.

Next consider whether PURR is consistent. Theorem 1 showed that $\ddot{R}_n^{\hat{y}}(g)$ is consistent. Following the same logic in Theorem 1's proof, it is straightforward to see that when performing decomposition on $R_p^{\hat{y}}(g)$, $\ddot{R}_p^{\hat{y}}(g)$ is also consistent.

It follows by induction that PURR (and any similarly-defined recursive risk estimator) is consistent. Theorem 1 shows the consistency of the base case where both composite terms (e.g., $R_u^{\hat{y}}(g)$ and $R_B^{\hat{y}}(g)$ in Eq. (10)) were estimated directly from training data. By induction, it is again straightforward from Theorem 1 that any decomposed term (e.g., $R_A^{\hat{y}}(g)$ in Eq. (10)) formed from the sum of consistent estimators must be itself consistent.

Theorem 1 further demonstrated that applying absolute-value correction does not affect the consistency of a risk estimator. Therefore, any recursive risk estimator with absolute-value correction is consistent. PURR's consistency is just a single, specific example of this general property. $\square$

## C  Non-Negativity Correction Empirical Risk Minimization Algorithms

Kiryo et al. [8]'s non-negativity correction algorithm uses the $\max\{0, \cdot\}$ term to ensure a plausible risk estimate. Unlike our simpler absolute-value correction described in Section 3, Kiryo et al.'s non-negativity correction requires a custom empirical risk minimization (ERM) procedure. This section presents the custom ERM algorithms required if non-negativity correction is used for our two-step methods and PURR.

### C.1  Two-Step, Non-Negativity ERM Algorithm

The weighted-unlabeled, unlabeled (wUU) risk estimator with non-negativity correction is defined as:

$$\widehat{R}_{\text{nn-wUU}}(g) := \max\left\{0, \widehat{R}_{\text{te-u}}^+(g) - (1 - \pi_{\text{te}})\widetilde{R}_{\text{n-u}}^+(g)\right\} + (1 - \pi_{\text{te}})\widetilde{R}_{\text{n-u}}^-(g). \tag{23}$$

The arbitrary-positive, negative, unlabeled (aPNU) risk estimator with non-negativity correction is similarly defined as:

$$\widehat{R}_{\text{nn-aPNU}}(g) := (1 - \rho)\pi_{\text{te}}R_p^+(g) + (1 - \pi_{\text{te}})\widetilde{R}_{\text{n-u}}^-(g) + \rho \max\left\{0, R_{\text{te-u}}^+(g) - (1 - \pi_{\text{te}})\widetilde{R}_{\text{n-u}}^+(g)\right\}. \tag{24}$$

Like their counterparts with absolute-value correction, both $\widehat{R}_{\text{nn-wUU}}(g)$ and $\widehat{R}_{\text{nn-aPNU}}(g)$ are consistent estimators.

Algorithm 2 shows the custom ERM framework for $\widehat{R}_{\text{nn-wUU}}(g)$ and $\widehat{R}_{\text{nn-aPNU}}(g)$ with integrated "defitting." The algorithm learns parameters $\theta$ for decision function $g$. The non-negativity correction occurs whenever $\widehat{R}^+_{\text{te-u}}(g) - (1 - \pi_{\text{te}})\widehat{R}^+_{\text{n-u}}(g) < 0$ (see line 7). The basic algorithm is heavily influenced by the stochastic optimization algorithm proposed by Kiryo et al. [8].

---

**Algorithm 2** wUU and aPNU with non-negativity correction custom ERM procedure

---

**Input**: Datasets $(\mathcal{X}_{\text{p}}, \widetilde{\mathcal{X}}_{\text{n}}, \mathcal{X}_{\text{te-u}})$, hyperparameters $(\gamma, \eta)$ and risk estimator $\widehat{R}_{\text{TS}}(g) \in \{\widehat{R}_{\text{nn-wUU}}(g), \widehat{R}_{\text{nn-aPNU}}(g)\}$
**Output**: Decision function $g$'s parameters $\theta$
 1: Select SGD-like optimization algorithm $\mathcal{A}$
 2: **while** Stopping criteria not met **do**
 3:     Shuffle $(\mathcal{X}_{\text{p}}, \widetilde{\mathcal{X}}_{\text{n}}, \mathcal{X}_{\text{te-u}})$ into $N$ batches
 4:     **for each** minibatch $(\mathcal{X}_{\text{p}}^{(i)}, \widetilde{\mathcal{X}}_{\text{n}}^{(i)}, \mathcal{X}_{\text{te-u}}^{(i)})$ **do**
 5:         **if** $\widehat{R}^+_{\text{te-u}}(g) - (1 - \pi_{\text{te}})\widetilde{R}^+_{\text{n-u}}(g) < 0$ **then**
 6:             Set gradient $-\nabla_\theta \left( \widehat{R}^+_{\text{te-u}}(g) - (1 - \pi_{\text{te}})\widetilde{R}^+_{\text{n-u}}(g) \right)$
 7:             Update $\theta$ by $\mathcal{A}$ with attenuated learning rate $\gamma\eta$
 8:         **else**
 9:             Set gradient $\nabla_\theta \widehat{R}_{\text{TS}}(g)$
10:             Update $\theta$ by $\mathcal{A}$ with default learning rate $\eta$
11: **return** $\theta$ minimizing validation loss

---

Algorithm 2 terminates after a fixed epoch count (see Table 9 for the number of epochs used for each dataset). Although not shown in Algorithm 2, the validation loss is measured at the end of each epoch. The algorithm returns the model parameters with the lowest validation loss.

## C.2 PURR Non-Negativity ERM Algorithm

For readability and compactness, let $[a]_+ := \max\{0, a\}$. PURR with non-negativity correction is defined as

$$\widehat{R}_{\text{nn-PURR}}(g) := \underbrace{\left[ \widehat{R}^+_{\text{te-u}}(g) - (1 - \pi_{\text{te}}) \underbrace{\left[ \frac{\widehat{R}^+_{\text{tr-u}}(g) - \pi_{\text{tr}}\widehat{R}^+_{\text{tr-p}}(g)}{1 - \pi_{\text{tr}}} \right]_+}_{\widehat{R}^+_{\text{te-n}}(g)} \right]_+}_{\pi_{\text{te}}\widehat{R}^+_{\text{te-p}}(g)} + (1 - \pi_{\text{te}}) \left[ \underbrace{\frac{\widehat{R}^-_{\text{tr-u}}(g) - \pi_{\text{tr}}\widehat{R}^-_{\text{tr-p}}(g)}{1 - \pi_{\text{tr}}}}_{\widehat{R}^-_{\text{te-n}}(g)} \right]_+ . \quad (25)$$

Like $\widehat{R}_{\text{PURR}}(g)$ from Section 6, $\widehat{R}_{\text{nn-PURR}}(g)$ is a consistent estimator.

When a risk estimator only has a single term that can be negative (like nnPU, $\widehat{R}_{\text{nn-wUU}}(g)$, and $\widehat{R}_{\text{nn-aPNU}}(g)$), the custom non-negativity ERM framework is relatively straightforward as shown in Algorithm 2. However, $\widehat{R}_{\text{nn-PURR}}(g)$ has three non-negativity corrections — one of which is nested inside another non-negativity correction.

Algorithm 3 details $\widehat{R}_{\text{nn-PURR}}(g)$'s custom ERM procedure with learning rate $\eta$. Each non-negativity correction is individually checked with the ordering critical. The optimizer minimizes risk on positive set $\mathcal{X}_{\text{p}}$ by both decreasing $\widehat{R}^+_{\text{p}}(g)$ and increasing $\widehat{R}^-_{\text{p}}(g)$. In contrast, each unlabeled example's minimizing risk is uncertain. This creates explicit tension and uncertainty for the optimizer. This enforced trade-off over the best unlabeled risk commonly delays or counteracts unlabeled set overfitting. As such, overfitting is most likely with labeled (positive) data. When that occurs, $\widehat{R}^-_{\text{tr-p}}(g)$ increases significantly making $\widehat{R}^-_{\text{te-n}}(g)$ most likely to be negative so its non-negativity is checked first (line 5). Nested term $\widehat{R}^+_{\text{te-n}}(g)$ receives second highest priority since whenever its value is implausible, any term depending on it, e.g., $\widehat{R}^+_{\text{te-p}}(g)$, is meaningless. By elimination, $\widehat{R}^+_{\text{te-p}}(g)$ has lowest priority.

Algorithm 3 applies non-negativity correction by negating risk $\widehat{R}^{\tilde{y}}_A(g)$'s gradient (see Eq. (10)). This addresses overfitting by "defitting" $g$. A large negative gradient can push $g$ into a poor parameter space so hyperparameter $\gamma \in (0, 1]$ limits the amount of correction by attenuating gradient magnitude.

---

**Algorithm 3** PURR with non-negativity correction custom ERM procedure

---

**Input**: Datasets $(\mathcal{X}_{\mathrm{p}}, \mathcal{X}_{\mathrm{tr\text{-}u}}, \mathcal{X}_{\mathrm{te\text{-}u}})$ & hyperparameters $(\gamma, \eta)$
**Output**: Decision function $g$'s parameters $\theta$

1: Select SGD-like optimization algorithm $\mathcal{A}$
2: **while** Stopping criteria not met **do**
3:     Shuffle $(\mathcal{X}_{\mathrm{p}}, \mathcal{X}_{\mathrm{tr\text{-}u}}, \mathcal{X}_{\mathrm{te\text{-}u}})$ into $N$ batches
4:     **for each** minibatch $(\mathcal{X}_{\mathrm{p}}^{(i)}, \mathcal{X}_{\mathrm{tr\text{-}u}}^{(i)}, \mathcal{X}_{\mathrm{te\text{-}u}}^{(i)})$ **do**
5:         **if** $\widehat{R}_{\mathrm{te\text{-}n}}^{-}(g) < 0$ **then**
6:             Use $\mathcal{A}$ to update $\theta$ with $-\gamma\eta\nabla_\theta \widehat{R}_{\mathrm{te\text{-}n}}^{-}(g)$
7:         **else if** $\widehat{R}_{\mathrm{te\text{-}n}}^{+}(g) < 0$ **then**
8:             Use $\mathcal{A}$ to update $\theta$ with $-\gamma\eta\nabla_\theta \widehat{R}_{\mathrm{te\text{-}n}}^{+}(g)$
9:         **else if** $\widehat{R}_{\mathrm{te\text{-}p}}^{+}(g) < 0$ **then**
10:             Use $\mathcal{A}$ to update $\theta$ with $-\gamma\eta\nabla_\theta \widehat{R}_{\mathrm{te\text{-}p}}^{+}(g)$
11:         **else**
12:             Use $\mathcal{A}$ to update $\theta$ with $\eta\nabla_\theta \widehat{R}_{\mathrm{nn\text{-}PURR}}(g)$
13: **return** $\theta$ minimizing validation loss

---

# D Detailed Experimental Setup

This section details the experimental setup used to collect the results in Sections 7 and E.

## D.1 Reproducing our Experiments

Our implementation is written and tested in Python 3.6.5 and 3.7.1 using the `PyTorch` [43] neural network framework versions 1.3.1 and 1.4. The source code is available at: https://github.com/ZaydH/arbitrary_pu. The repository includes file `requirements.txt` that details Python package dependency information.

To run the program, invoke:

```
python driver.py ConfigFile
```

where `ConfigFile` is a `yaml`-format text file specifying the experimental setup. Repository folder "`src/configs`" contains the configuration files for the experiments in Sections 7, E.1, and E.4. Prior probability shifts can be made by modifying the configuration files (see `yaml` fields `train_prior` and `test_prior`).

**Datasets**  Our program automatically retrieves all necessary data. Synthetic data is generated by the program itself. Otherwise the dataset is downloaded automatically from the web. If you have trouble downloading any datasets, please verify that your network/firewall ports are properly configured.

## D.2 Class Definitions

### D.2.1 Partially and Fully Disjoint Positive Distribution Supports

Section 7.2's experimental setups are very similar to Hsieh et al. [11]'s experiments for positive, unlabeled, biased-negative learning. We even follow Hsieh et al.'s label partitions. The basic rationale motivating the splits are:

- **MNIST**: Odd (positive class) vs. even (negative class) digits. Each digit's frequency in the original dataset is approximately $0.1$ making each class's target prior $5 * 0.1 = 0.5$.

- **20 Newsgroups**: As its name suggests, the 20 Newsgroups dataset consists of 20 disjoint labels. Categories are formed by partitioning those 20 labels into 7 groups based on the corresponding text documents' general theme. Our classes are formed by splitting the categories into two disjoint sets. Specifically, the positive-test class consists of documents with labels 0 to 10 in the original dataset. The negative class is comprised of documents whose labels in the original dataset are 11-19. This split's actual positive prior probability is approximately 0.56.[7]

- **CIFAR10**: Inanimate objects (positive class) vs. animals (negative class). CIFAR10 is a multiclass dataset with ten labels. Each label is equally common in the training and test set, i.e., has prior 0.1. Since CIFAR10's positive-test class has exactly four labels (e.g., plane, automobile, truck, and ship), the positive-test prior is $4 * 0.1 = 0.4$.

For this experiment set, the distribution shift between train and test is premised on new subclasses emerging in the test distribution (e.g., due to novel adversarial attacks or systematic failure to collect data on a positive subpopulation in the original dataset).

### D.2.2 TREC Spam Classification

As noted previously, PU learning has been applied to multiple adversarial domains including opinion spam [3, 16, 17, 18]. We use spam classification as a vehicle for testing our method in an adversarial domain.

Clearly, email spam classification is not a scenario where PU learning would generally be applied. Labeled data for both classes is generally plentiful (especially at the corporate level), and for most modern email systems, spam classification is a solved problem. For our purposes, spam email provides a good avenue for demonstrating our methods' performance in an adversarial setting for multiple reasons, including:

- The positive class (i.e., spam) evolves significantly faster than the negative class (i.e., not spam or "ham").

- Our fixed negative class-conditional distribution assumption (i.e., Eq. (7)) will not explicitly hold. This more closely represents what will be encountered "in-the-wild."

- Public spam/ham datasets exist eliminating the need to use our own proprietary adversarial learning dataset.

- Email dates provide a realistic criteria for partitioning the training and test datasets.

To be clear, what we propose here is not intended as a plausible, deployable spam classifier. Rather, we show that our methods apply to real-world adversarial domains.

**Dataset Construction**     The Text REtrieval Conference (TREC) is organized annually be the United States' National Institute of Standards and Technology (NIST) to support information retrieval research [34]. In 2005, 2006, and 2007, TREC arranged annual spam classifier competitions where they released corpuses of spam and ham (i.e., not spam) emails.

As detailed in Table 5, the training set consisted of the TREC 2005 (TREC05) email dataset[8] while the test set was the TREC 2007 (TREC07) email dataset[9]. Basic statistics for the two datasets appear in Table 4.

The two sets of emails come from different domains. TREC05's ham emails derive largely from the Enron dataset. In contrast, TREC07's emails were received by a particular server between April and July 2007. Many of the ham emails were received by the University of Waterloo where the datasets were curated.

Due to the extended time required to encode all emails using the ELMo embedder (see Section D.7), we consider the first 10,000 emails from each dataset as defined by the dataset's `full/index` file.

Table 4: TREC05 & TREC07 dataset statistics

|  | TREC05 | TREC07 |
| --- | --- | --- |
| Dataset Size | 92,189 | 75,419 |
| Approx. % Spam | ~57% | ~66% |

### D.2.3 Identical Positive Supports with Bias

Table 6 defines the positive and negative classes for the 10 LIBSVM datasets used in Section E.4. Label "+1" always corresponded to the positive class. In two-class (binary) datasets, the other label was the negative class. For multiclass datasets (e.g., connect4), whichever other class had the most examples was used as the negative class.

Table 5: TREC spam email classification datasets

| Class | Definition |
|---|---|
| Pos. Train | TREC05 Spam |
| Neg. Train | TREC05 Ham |
| Pos. Test | TREC07 Spam |
| Neg. Test | TREC07 Ham |

Table 6: Positive & negative class definitions for the LIBSVM datasets in Section E.4

| Dataset | $d$ | Pos. Class | Neg. Class |
|---|---|---|---|
| banana | 2 | +1 | 2 |
| cod-rna | 8 | +1 | −1 |
| susy | 18 | +1 | 0 |
| ijcnn1 | 22 | +1 | −1 |
| covtype.b | 54 | +1 | 2 |
| phishing | 68 | +1 | 0 |
| a9a | 123 | +1 | −1 |
| connect4 | 126 | +1 | −1 |
| w8a | 300 | +1 | −1 |
| epsilon | 2,000 | +1 | −1 |

### D.3 bPU Selection Bias Invariance of Order

Section E.5's experiments follow the invariance of order assumption as proposed and implemented by Kato et al. [13]. Their original experiments considered the MNIST dataset. For completeness, we expand our comparison to their method to also include the MNIST variants, FashionMNIST [44] and KMNIST [45]. Like MNIST, both FashionMNIST and KMNIST are multiclass datasets consisting of 10 disjoint labels. As described in Section D.2.1, binary classes are formed by partitioning the original set of labels.

As before, MNIST splits the labels between odds (positive class) and evens (negative class). For consistency, we used the same odd/even label partition for FashionMNIST and KMNIST. Note that such a partitioning lacks a corresponding semantic meaning for those two datasets.

### D.4 Training, Validation, and Test Set Sizes

Table 7 lists the default size of each dataset's positive ($\mathcal{X}_\text{p}$), unlabeled train ($\mathcal{X}_\text{tr-u}$), unlabeled test ($\mathcal{X}_\text{te-u}$), and inductive test sets. All LIBSVM datasets (e.g., susy, a9a, etc. in Section E.4) used the dataset sizes defined by Sakai and Shimizu [14]. The separate validation set – made up of only positive and unlabeled examples – was one-fifth Table 7's training set sizes. Each learner observed identical dataset splits in each trial.

Special inductive test set sizes were needed for two of Section 7.2's disjoint positive-support experiments. To understand why, consider the MNIST disjoint-support experiment (i.e., the fourth MNIST row in Table 1) where the negative class (N) is comprised of labels $\{0, 2\}$ and the positive-test class ($\text{P}_\text{test}$) is composed of labels $\{5, 7\}$. Each label has approximately 1,000 examples in the dedicated test set meaning there are approximately 4,000 total test examples between the negative and positive classes. However, MNIST's default inductive test set size ($n_\text{Test}$) is 5,000 (see Table 7). Rather than duplicating test set examples, we reduced MNIST's $n_\text{Test}$ to 1,500 *for the disjoint positive-support experiments only*. 20 Newsgroups has the same issue so its disjoint-positive support $n_\text{Test}$ was also reduced as specified in Table 8. To be clear, for all other datasets and experimental setups in Sections 7.2, 7.3, E.1, and E.4, Table 7 applies.

MNIST, 20 Newsgroups, and CIFAR10 have predefined test sets, which we exclusively used to collect the inductive results. They were not used for training or validation. Only some LIBSVM datasets have dedicated test sets, and for those that do, Sakai and Shimizu [14] do not specify whether the test set was held out in their experiments. When applicable, we merge the LIBSVM train and test datasets together as if there was only a single monolithic training set. $\mathcal{X}_\text{p}$, $\mathcal{X}_\text{tr-u}$, $\mathcal{X}_\text{te-u}$ and the inductive test set are independently sampled at random from this monolithic set without replacement.

Table 7: Each dataset's default training set sizes. LIBSVM denotes all datasets downloaded directly from [30] and used in Section E.4. All quantities in the table do *not* include the validation set.

| Dataset | $n_{\text{p}}$ | $n_{\text{tr-u}}$ | $n_{\text{te-u}}$ | $n_{\text{Test}}$ |
|---|---|---|---|---|
| MNIST | 1,000 | 5,000 | 5,000 | 5,000 |
| 20 Newsgroups | 500 | 2,500 | 2,500 | 5,000 |
| CIFAR10 | 1,000 | 5,000 | 5,000 | 3,000 |
| TREC Spam | 500 | 1,250 | 1,250 | 1,000 |
| Synthetic | 1,000 | 1,000 | 1,000 | N/A |
| LIBSVM | 250 | 583 | 583 | 2,000 |
| FashionMNIST | 833 | $\leftarrow$ See Sec. E.5 $\rightarrow$ | | 5,000 |
| KMNIST | 833 | $\leftarrow$ See Sec. E.5 $\rightarrow$ | | 5,000 |

Table 8: Smaller MNIST and 20 Newsgroups inductive test set sizes, i.e., $n_{\text{Test}}$, used in the disjoint-support experiments.

| Dataset | $n_{\text{Test}}$ |
|---|---|
| MNIST | 3,000 |
| 20 Newsgroups | 1,500 |

Since the PUc formulation is convex, Sakai and Shimizu train their final model on the combined training and validation set.

### D.5 CIFAR10 Image Representation

Each CIFAR10 [32] image is 32 pixels by 32 pixels with three (RGB) color channels (3,072 dimensions total). PUc specifies a convex model so it cannot be used to train (non-convex) deep convolutional networks directly. To ensure a meaningful comparison, we leveraged the DenseNet-121 deep convolutional network architecture pretrained on 1.2 million images from ImageNet [39]. The network's (linear) classification layer was removed, and the experiments used the 1,024-dimension feature vector output by DenseNet's convolutional backbone.

### D.6 20 Newsgroups Document Representation

The 20 Newsgroups dataset is a collection of internet discussion board posts. The original dataset consisted of 20,000 documents [33]; it was pruned to 18,828 documents in 2007 after removal of duplicates and cross-posts [46]. This latest dataset has a predefined split of 11,314 train and 7,532 test documents. Similar to CIFAR10, we use transfer learning to create a richer representation of each document.

Classic word embedding models like GloVe and Word2Vec yield token representations that are independent of context. Proposed by Peters et al. [37], ELMo (embeddings for language models) enhances classic word embeddings by making the token representations context dependent. We use ELMo to encode each 20 Newsgroup document as described below.

ELMo's embedder consists of three sequential layers — first a character convolutional neural network (CNN) provides subword information and improves unknown word robustness. The CNN's output is then fed into a two-layer, bidirectional LSTM. The output from each of ELMo's layers is a 1,024-dimension vector. For a token stream of length $m$, the output of ELMo's embedder would be a tensor of size $\langle \text{\#Layers} \times d_{\text{layer}} \times \text{\#Tokens} \rangle$ — in this case $\langle 3 \times 1024 \times m \rangle$.

Like Hsieh et al. [11] who used this encoding scheme for positive, unlabeled, biased-negative (PUbN) (PUbN) learning, we used Rücklé et al. [38]'s sentence representation encoding scheme, which takes the minimum, maximum, and average value along each ELMo layer's output dimension. The dimension of the resulting document encoding is:

$$|\{\max, \min, \text{avg}\}| \cdot \text{\#Layers} \cdot d_{\text{layer}} = 3 \cdot 3 \cdot 1024 = 9,216.$$

When documents are encoded serially, each document implicitly contains information about all preceding documents. Put simply, the order documents are processed affects each document's final encoding. For consistency, all 20 Newsgroups experiments used a single identical encoding for all learners.

The Allen Institute for Artificial Intelligence has published multiple pretrained ELMo models. We used the ELMo model trained on a 5.5 billion token corpus — 1.9 billion from Wikipedia and 3.6 billion from a news crawl. We chose this version because ELMo's developers report that it was the best performing.

## D.7  TREC Email Representation

The TREC05 and TREC07 emails are encoded using the ELMo embedder identical to 20 Newsgroups. See Section D.6 above for the details.

## D.8  Models and Hyperparameters

This section reviews the experiments' hyperparameter methodology.

As specified by its authors, PUc's hyperparameters were tuned via importance-weighted cross validation (IWCV) [40]. PUc's author-supplied implementation includes a built-in hyperparameter tuning architecture that we used without modification.

Our hyperparameters and best-epoch weights were selected using the validation loss (using the associated risk estimation) on a validation set. Our experiments' hyperparameters can be grouped into two categories. First, some hyperparameters (e.g., number of epochs) apply to most/all learners (excluding PUc). The second category's hyperparameters are individualized to each learner and were used for all of that learner's experiments on the corresponding dataset.

Table 9 enumerates the general hyperparameter settings that applied to most/all learners. Batch sizes were selected based on the dataset sizes (see Tables 7 and 8) while the epoch count was determined after monitoring the typical time required for the best validation loss to stop (meaningfully) changing. A grid search was used to select each dataset's layer count; we specifically searched set $\{1, 2, 3\}$ for $g$ and $\{0, 1, 2\}$ for $\hat{\sigma}$. With the exception of the output layer, each linear layer used ReLU activation and batch normalization [47]. The selected layer count minimized the median validation loss across all learners.

Tables 10, 11, and 12 enumerate the final hyperparameter settings for our models, nnPU, and the positive-negative (PN) learners respectively. The selected hyperparameter setting had the best average validation loss across 10 independent trials. We also used a grid search for these parameters. The search space was: learning rate $\eta \in \{10^{-5}, 10^{-4}, 10^{-3}\}$, weight decay $\lambda \in \{10^{-4}, 10^{-3}, 5 \cdot 10^{-3}, 10^{-2}, 10^{-1}\}$, and (where applicable) gradient attenuator $\gamma \in \{0.1, 0.5, 1.0\}^{10}$.

By monitoring the (implausible) validation loss during Step #1, we observed overfitting when using the rich ELMo representations for the 20 Newsgroups and TREC email datasets. To address this, we added a dropout layer (with probability $p = 0.5$) before the input to each linear (i.e., fully-connected) layer. It is uncommon to use dropout even on the input dimension. However, we deliberately made this choice to still allow dropout even if we use a strictly linear-in-parameter model. Dropout was not used for any other dataset.

Table 9: General hyperparameter settings

| Dataset | #Epoch | Layer Count | | Batch Size | | | Dropout? |
|---|---|---|---|---|---|---|---|
| | | $g(x)$ | $\hat{\sigma}(x)$ | $g(x)$ | $\hat{\sigma}(x)$ | $PN_{te}$ | |
| MNIST | 200 | 3 | 1 | 5,000 | 5,000 | 4,000 | |
| 20 Newsgroups | 200 | 1 | 1 | 5,000 | 2,500 | 2,000 | ✓ |
| CIFAR10 | 200 | 2 | 1 | 10,000 | 2,500 | 1,500 | |
| TREC Spam | 200 | 1 | 0 | 1,000 | 1,000 | 1,000 | ✓ |
| Synthetic | 100 | N/A | N/A | 2,000 | 750 | 500 | |
| banana | 500 | 3 | 2 | 500 | 750 | 500 | |
| cod-rna | 500 | 2 | 1 | 500 | 750 | 500 | |
| susy | 500 | 2 | 2 | 500 | 750 | 500 | |
| ijcnn1 | 500 | 2 | 2 | 500 | 750 | 500 | |
| covtype.b | 500 | 3 | 1 | 500 | 750 | 500 | |
| phishing | 500 | 2 | 2 | 500 | 750 | 500 | |
| a9a | 500 | 2 | 2 | 500 | 750 | 500 | |
| connect4 | 500 | 2 | 1 | 500 | 750 | 500 | |
| w8a | 500 | 2 | 1 | 500 | 750 | 500 | |
| epsilon | 500 | 1 | 0 | 500 | 750 | 500 | |
| FashionMNIST | 200 | 3 | 1 | ⟵ See Section E.5 ⟶ | | | |
| KMNIST | 200 | 3 | 1 | ⟵ See Section E.5 ⟶ | | | |

Table 10: Dataset-specific hyperparameter settings for our aPU learners. Hyperparameter $\gamma*$ only applies when using Kiryo et al. [8]'s non-negativity correction instead of our absolute-value correction.

| Dataset | PURR | | | $\hat{\sigma}$ | | | aPNU | | | wUU | | |
|---|---|---|---|---|---|---|---|---|---|---|---|---|
| | $\eta$ | $\lambda$ | $\gamma^*$ | $\eta$ | $\lambda$ | $\gamma^*$ | $\eta$ | $\lambda$ | $\gamma^*$ | $\eta$ | $\lambda$ | $\gamma^*$ |
| MNIST | 1E−3 | 1E−3 | 1 | 1E−3 | 5E−3 | 1 | 1E−3 | 1E−3 | 1 | 1E−4 | 5E−3 | 1 |
| 20 Newsgroups | 1E−4 | 1E−4 | 0.5 | 1E−3 | 5E−3 | 1 | 1E−4 | 1E−4 | 0.5 | 1E−4 | 1E−4 | 0.5 |
| CIFAR10 | 1E−3 | 1E−3 | 1 | 1E−3 | 5E−3 | 1 | 1E−3 | 1E−4 | 0.5 | 1E−3 | 1E−2 | 0.5 |
| TREC Spam | 1E−3 | 1E−2 | 1 | 1E−3 | 1E−1 | 1 | 1E−3 | 1E−3 | 0.5 | 1E−3 | 1E−2 | 0.5 |
| Synthetic | 1E−2 | 0 | 1 | 1E−2 | 0 | 1 | 1E−2 | 0 | 1 | 1E−2 | 0 | 1 |
| banana | 1E−4 | 1E−3 | 0.1 | 1E−4 | 5E−3 | 1 | 1E−5 | 1E−3 | 0.5 | 1E−3 | 1E−3 | 0.1 |
| cod_rna | 1E−4 | 1E−3 | 0.5 | 1E−3 | 1E−4 | 1 | 1E−3 | 1E−3 | 0.1 | 1E−4 | 1E−3 | 0.5 |
| susy | 1E−5 | 1E−2 | 0.5 | 1E−4 | 5E−3 | 1 | 1E−5 | 1E−3 | 0.1 | 1E−5 | 1E−4 | 0.5 |
| ijcnn1 | 1E−4 | 1E−3 | 0.5 | 1E−4 | 5E−3 | 1 | 1E−4 | 1E−2 | 0.5 | 1E−4 | 1E−2 | 0.5 |
| covtype.b | 1E−5 | 1E−3 | 1 | 1E−3 | 1E−4 | 1 | 1E−5 | 1E−3 | 0.1 | 1E−4 | 1E−3 | 1 |
| phishing | 1E−5 | 1E−3 | 0.5 | 1E−3 | 1E−4 | 1 | 1E−5 | 1E−3 | 0.5 | 1E−5 | 1E−3 | 0.5 |
| a9a | 1E−5 | 1E−4 | 1 | 1E−4 | 5E−3 | 1 | 1E−5 | 1E−4 | 0.5 | 1E−4 | 1E−3 | 0.5 |
| connect4 | 1E−4 | 1E−3 | 0.5 | 1E−3 | 1E−4 | 1 | 1E−4 | 1E−4 | 0.5 | 1E−3 | 1E−2 | 0.5 |
| w8a | 1E−5 | 1E−4 | 0.5 | 1E−3 | 1E−4 | 1 | 1E−5 | 1E−3 | 0.5 | 1E−5 | 1E−2 | 0.5 |
| epsilon | 1E−5 | 1E−2 | 0.1 | 1E−3 | 1E−4 | 1 | 1E−5 | 1E−2 | 0.1 | 1E−4 | 1E−2 | 0.1 |
| FashionMNIST | 1E−3 | 1E−3 | 1 | 1E−3 | 5E−3 | 1 | 1E−3 | 1E−3 | 1 | 1E−4 | 5E−3 | 1 |
| KMNIST | 1E−3 | 1E−3 | 1 | 1E−3 | 5E−3 | 1 | 1E−3 | 1E−3 | 1 | 1E−4 | 5E−3 | 1 |

Table 11: Dataset-specific hyperparameter settings for nnPU.

| Dataset | nnPU$_{te \cup tr}$ | | | nnPU$_{te}$ | | |
|---|---|---|---|---|---|---|
| | $\eta$ | $\lambda$ | $\gamma$ | $\eta$ | $\lambda$ | $\gamma$ |
| MNIST | 1E−3 | 1E−3 | 0.5 | 1E−3 | 1E−3 | 0.5 |
| 20 Newsgroups | 1E−3 | 1E−3 | 0.5 | 1E−3 | 1E−2 | 0.5 |
| CIFAR10 | 1E−4 | 1E−3 | 0.1 | 1E−4 | 1E−3 | 0.1 |
| TREC Spam | 1E−3 | 1E−2 | 0.1 | 1E−3 | 1E−2 | 0.1 |
| Synthetic | 1E−2 | 0 | 1 | 1E−2 | 0 | 1 |
| banana | 1E−3 | 1E−3 | 1 | 1E−4 | 1E−3 | 0.5 |
| cod_rna | 1E−3 | 1E−3 | 0.5 | 1E−3 | 1E−3 | 0.5 |
| susy | 1E−5 | 1E−2 | 0.1 | 1E−3 | 1E−3 | 0.5 |
| ijcnn1 | 1E−3 | 1E−2 | 0.5 | 1E−3 | 1E−3 | 0.5 |
| covtype.b | 1E−3 | 1E−2 | 0.5 | 1E−3 | 1E−2 | 0.5 |
| phishing | 1E−3 | 1E−2 | 0.5 | 1E−3 | 1E−2 | 0.5 |
| a9a | 1E−3 | 1E−2 | 1 | 1E−3 | 1E−3 | 0.5 |
| connect4 | 1E−3 | 1E−3 | 0.1 | 1E−3 | 1E−4 | 1 |
| w8a | 1E−3 | 1E−3 | 0.5 | 1E−3 | 1E−3 | 0.5 |
| epsilon | 1E−3 | 1E−3 | 0.5 | 1E−3 | 1E−3 | 0.5 |
| FashionMNIST | 1E−3 | 1E−3 | 0.5 | 1E−3 | 1E−3 | 0.5 |
| KMNIST | 1E−3 | 1E−3 | 0.5 | 1E−3 | 1E−3 | 0.5 |

Table 12: Dataset-specific hyperparameter settings for the positive-negative (PN) learners

| Dataset | PN$_{te}$ | | PN$_{tr}$ | |
|---|---|---|---|---|
| | $\eta$ | $\lambda$ | $\eta$ | $\lambda$ |
| MNIST | 1E−3 | 1E−3 | 1E−3 | 1E−3 |
| 20 Newsgroups | 1E−3 | 1E−3 | 1E−3 | 1E−2 |
| CIFAR10 | 1E−4 | 1E−3 | 1E−3 | 1E−2 |
| TREC Spam | 1E−3 | 1E−2 | 1E−3 | 1E−2 |
| Synthetic | 1E−2 | 0 | 1E−2 | 0 |
| banana | 1E−4 | 1E−2 | 1E−4 | 1E−3 |
| cod_rna | 1E−3 | 1E−4 | 1E−3 | 1E−4 |
| susy | 1E−4 | 1E−2 | 1E−5 | 1E−2 |
| ijcnn1 | 1E−3 | 1E−3 | 1E−3 | 1E−2 |
| covtype.b | 1E−3 | 1E−2 | 1E−3 | 1E−2 |
| phishing | 1E−3 | 1E−3 | 1E−3 | 1E−2 |
| a9a | 1E−5 | 1E−2 | 1E−3 | 1E−3 |
| connect4 | 1E−3 | 1E−2 | 1E−3 | 1E−3 |
| w8a | 1E−4 | 1E−4 | 1E−4 | 1E−3 |
| epsilon | 1E−4 | 1E−3 | 1E−3 | 1E−3 |
| FashionMNIST | 1E−3 | 1E−3 | 1E−3 | 1E−3 |
| KMNIST | 1E−3 | 1E−3 | 1E−3 | 1E−3 |

# E Additional Experimental Results

This section includes experiments we consider insightful but for which there was insufficient space to include in the paper's main body. With the exception of the synthetic data experiments (see Section E.1) which focus on visually illustrative examples to build intuitions, performance evaluation is based on the inductive misclassification rate since it approximates the expected zero-one loss for an unseen example.

## E.1 Illustration using Synthetic Data

This section uses synthetic data to visualize scenarios where our algorithms succeed in spite of challenging conditions.

For simplicity, $\hat{\sigma}$ and $g$ are linear-in-parameter models optimized by L-BFGS. PUc also trains a linear-in-parameter models without Gaussian kernels. Since all methods use the same classifier architecture, our methods' performance advantage comes solely from algorithmic design.

Synthetic data were generated from multivariate Gaussians $\mathcal{N}(\boldsymbol{\mu}, \mathbf{I}_2)$ with different means $\boldsymbol{\mu}$ and identity covariance $\mathbf{I}_2$. In all experiments, the positive-test and negative class-conditional distributions were

$$p_{\text{te-p}}(x) = \frac{1}{2}\mathcal{N}([-2 \quad -1], \mathbf{I}_2) + \frac{1}{2}\mathcal{N}([-2 \quad 1], \mathbf{I}_2)$$

$$p_{\text{n}}(x) = \frac{1}{2}\mathcal{N}([\;\; 2 \quad -1], \mathbf{I}_2) + \frac{1}{2}\mathcal{N}([\;\; 2 \quad 1], \mathbf{I}_2).$$

$\pi_{\text{te}} = \pi_{\text{tr}} = 0.5$ makes the ideal *test* decision boundary $x_1 = 0$. The datasets in Figure 3 vary only in the positive-train class-conditional distribution, denoted $p_{\text{tr-}(\cdot)\text{-p}}(x)$ where "·" is subfigure a to c.

Figure 3a's positive-train class-conditional distribution is

$$p_{\text{tr-(a)-p}}(x) = \frac{1}{2}\mathcal{N}([\;\; 6 \quad -1], \mathbf{I}_2) + \frac{1}{2}\mathcal{N}([\;\; 6 \quad 1], \mathbf{I}_2), \tag{26}$$

making the training distribution's optimal separator linear. PUc performed poorly on this setup for two reasons: covariate shift's assumption $p_{\text{tr}}(y|x) = p_{\text{te}}(y|x)$ does not hold, and the positive-train supports are functionally disjoint so importance function $w(x)$ is practically unbounded. Our methods all performed well, even PU2aPNU where inclusion of $\mathcal{X}_{\text{p}}$'s risk had minimal impact since for most good boundaries, $\mathcal{X}_{\text{p}}$'s risk was an inconsequential penalty.

Figure 3b adds to $p_{\text{tr-(a)-p}}(x)$ a third Gaussian where

$$p_{\text{tr-(b)-p}}(x) = \frac{2}{3}p_{\text{tr-(a)-p}}(x) + \frac{1}{3}\mathcal{N}([-6 \quad 0], \mathbf{I}_2), \tag{27}$$

so the training distribution's optimal separator is non-linear. PUc performs poorly for the same reasons described above. The new centroid does not meaningfully affect PURR. The most important takeaway is that linear $\hat{\sigma}$'s inability to partition $\mathcal{X}_{\text{tr-u}}$ has limited impact on PU2wUU and PU2aPNU; $\mathcal{X}_{\text{tr-u}}$'s misclassified examples act as a fixed penalty that only slightly offsets the two-step decision boundaries.

Figure 3c uses the worst-case positive-train class-conditional, i.e., $p_{\text{tr-(c)-p}}(x) = p_{\text{n}}(x)$, making positive (labeled) data statistically identical to the (train and test) negative class-conditional distribution. Its training marginal $p_{\text{tr-u}}(x)$ is not separable – linearly or otherwise. Unlike PUc, our methods learned correct boundaries, which shows their robustness.

(a) Approx. linearly separable $\mathcal{X}_{\text{tr-u}}$    (b) Non-linearly separable $\mathcal{X}_{\text{tr-u}}$    (c) $p_{\text{tr-(c)-p}}(x) = p_{\text{n}}(x)$

Figure 3: Predicted linear decision boundaries for three synthetic datasets ($n_{\text{p}} = n_{\text{tr-u}} = n_{\text{te-u}} = 1,000$). Our three methods – PURR, PU2aPNU, and PU2wUU – are robust to non-linear & non-existent training class boundaries, but PUc fails in all three cases. Ideal boundary: $x_1 = 0$.

### E.2 Expanded MNIST, 20 Newsgroups, and CIFAR10 Experiment Set

Table 13 is an expanded version of Section 7.2's Table 1. We provide these additional results to give the reader further evidence of our methods' superior performance.

In this section, each of the three datasets (i.e., MNIST, 20 Newsgroups, and CIFAR10) now has two positive-training ($P_{train}$) class configurations that are partially disjoint from the positive-test ($P_{test}$) class. For each such configuration, Table 13 contains three experiments (in order):

1. $\pi_{tr} < \pi_{te}$
2. $\pi_{tr} = \pi_{te}$
3. $\pi_{tr} > \pi_{te}$

It is easier to directly compare the effects of increasing/decreasing $\pi_{tr}$ when the magnitude of the training prior increase and decrease are equivalent (e.g., for MNIST $\pi_{te} = 0.5$ so we tested performance at $\pi_{tr} = \pi_{te} \pm 0.12$ and $\pi_{tr} = \pi_{te} \pm 0.21$ depending on the class partition). We maintained that rule of thumb when possible, but cases did arise where there were insufficient positive example with the labels in $P_{train}$ to support such a high positive prior. In those cases, we clamp that $P_{train}$ class definition's maximum $\pi_{tr}$.

The key takeaway from Table 13 is that across these additional, orthogonal definitions of $P_{train}$, our methods still outperform PUc and nnPU* — usually by a wide margin (statistical significance according to 1% paired t-test).

In all experiments, our methods' performance degraded as $\pi_{tr}$ increased since a larger prior makes it harder to identify the negative examples in $\mathcal{X}_{tr\text{-}u}$. To gain an intuition about why this is true, consider the extreme case where $\pi_{tr} = 1$; learning is impossible since the positive-train class-conditional distribution may be arbitrarily different, and there are no negative samples that can be used to relate the two distributions. In contrast when $\pi_{tr} = 0$, identifying the negative set is trivial (i.e., all of $\mathcal{X}_{tr\text{-}u}$ is negative), and NU learning can be applied directly to learn $g$.

PUc performs best when $\pi_{tr} = \pi_{te}$. When $\pi_{tr}$ diverges from that middle point, PUc's performance declines. To gain an intuition why that is, consider density-ratio estimation in terms of the component class conditionals. When $\pi_{tr} = \pi_{te}$, $w(x) = 1$ for all negative examples; from Table 13's results, we know that PUc performs best when there is no bias, i.e., $P_{train} = P_{test}$. A static positive prior eliminates one possible source of bias making density-ratio estimation easier and more accurate.

Table 13: Full MNIST, 20 Newsgroups, and CIFAR10 experimental class partition results. Each result is the inductive misclassification rate (%) mean and standard deviation over 100 trials for MNIST, 20 Newsgroups, and CIFAR10 with different positive & negative class definitions. For *all* experiments with positive bias (i.e., rows 2–8 for each dataset), all three of our methods had statistically significant better performance than PUc and nnPU* according to a 1% paired t-test. Boldface indicates a shifted task's best performing method. Negative (N) & positive-test ($P_{test}$) class definitions are identical for each dataset's first three experiments. Positive train ($P_{train}$) specified as $P_{test}$ denotes no bias. Our three methods – PURR, PU2aPNU, and PU2wUU – are denoted with [†].

| | N | $P_{test}$ | $P_{train}$ | $\pi_{tr}$ | $\pi_{te}$ | PURR[†] | aPNU[†] | wUU[†] | PUc | nnPU* | $PN_{te}$ |
|---|---|---|---|---|---|---|---|---|---|---|---|
| | | | | | | | Two-Step (PU2) | | Baselines | | Ref. |
| MNIST | 0, 2, 4, 6, 8 | 1, 3, 5, 7, 9 | $P_{test}$ | 0.5 | 0.5 | 10.0 (1.3) | 10.0 (1.2) | 11.6 (1.6) | 8.6 (0.8) | 5.5 (0.5) | ↑ |
| | | | | 0.29 | 0.5 | 6.8 (0.8) | **5.3 (0.6)** | 6.0 (0.7) | 29.2 (2.1) | 36.7 (2.7) | |
| | | | 7, 9 | 0.5 | 0.5 | 9.4 (1.5) | **7.1 (0.9)** | 8.3 (1.5) | 26.8 (2.4) | 35.1 (2.5) | |
| | | | | 0.71 | 0.5 | 14.0 (3.0) | **11.1 (1.4)** | 14.8 (3.1) | 26.9 (3.0) | 34.5 (2.9) | 2.8 (0.2) |
| | | | | 0.38 | 0.5 | 8.1 (1.0) | **6.5 (0.8)** | 7.6 (0.9) | 20.2 (2.5) | 25.9 (1.1) | |
| | | | 1, 3, 5 | 0.5 | 0.5 | 10.0 (1.6) | **8.4 (1.1)** | 10.2 (1.4) | 18.5 (2.9) | 26.9 (1.2) | |
| | | | | 0.63 | 0.5 | 12.5 (2.3) | **11.4 (1.3)** | 14.3 (2.3) | 18.6 (3.3) | 28.5 (1.2) | ↓ |
| | 0, 2 | 5, 7 | 1, 3 | 0.5 | 0.5 | 4.0 (0.8) | 3.6 (0.9) | **3.1 (0.7)** | 17.1 (4.6) | 30.9 (5.3) | 1.1 (0.2) |
| 20 Newsgroups | sci, soc, talk | alt, comp, misc, rec | $P_{test}$ | 0.56 | 0.56 | 15.4 (1.3) | 14.9 (1.0) | 16.7 (2.3) | 14.9 (1.0) | 14.1 (0.8) | ↑ |
| | | | | 0.37 | 0.56 | 13.9 (0.7) | **12.8 (0.6)** | 14.3 (0.9) | 28.9 (1.8) | 28.8 (1.3) | |
| | | | misc, rec | 0.56 | 0.56 | 17.5 (2.1) | **13.5 (0.8)** | 15.1 (1.3) | 23.9 (3.0) | 28.8 (1.7) | |
| | | | | 0.65 | 0.56 | 20.2 (2.8) | **14.0 (0.9)** | 15.9 (1.5) | 21.8 (3.3) | 29.0 (1.8) | 10.5 (0.5) |
| | | | | 0.37 | 0.56 | **13.3 (0.6)** | 13.7 (0.6) | 14.4 (0.7) | 30.3 (2.0) | 31.4 (0.7) | |
| | | | comp | 0.56 | 0.56 | 16.0 (1.5) | **14.9 (0.7)** | 15.7 (0.9) | 28.6 (2.6) | 31.2 (0.8) | |
| | | | | 0.65 | 0.56 | 19.2 (2.4) | **15.6 (0.9)** | 16.5 (1.2) | 27.8 (2.7) | 31.3 (0.7) | ↓ |
| | misc, rec | soc, talk | alt, comp | 0.55 | 0.46 | 5.9 (1.0) | 7.1 (1.1) | **5.6 (1.7)** | 18.5 (4.3) | 35.3 (5.2) | 2.1 (0.3) |
| CIFAR10 | Bird, Cat, Deer, Dog, Frog, Horse | Plane, Auto, Ship, Truck | $P_{test}$ | 0.4 | 0.4 | 14.1 (0.8) | 14.2 (1.3) | 15.4 (1.7) | 13.8 (0.7) | 12.3 (0.6) | ↑ |
| | | | | 0.14 | 0.4 | 12.1 (0.7) | **11.9 (0.7)** | 12.4 (0.9) | 26.7 (1.4) | 26.7 (1.0) | |
| | | | Plane | 0.4 | 0.4 | **13.8 (0.9)** | 14.5 (1.4) | 15.1 (1.6) | 20.6 (1.5) | 27.4 (1.0) | |
| | | | | 0.6 | 0.4 | **16.1 (1.1)** | 16.7 (1.5) | 20.0 (2.7) | 21.5 (1.6) | 28.4 (1.0) | 9.7 (0.5) |
| | | | | 0.25 | 0.4 | 12.7 (0.7) | **12.4 (0.7)** | 12.8 (0.8) | 19.2 (1.1) | 20.3 (0.8) | |
| | | | Auto, Truck | 0.4 | 0.4 | 14.1 (0.9) | **13.9 (1.1)** | 14.4 (1.2) | 17.7 (1.0) | 20.3 (0.8) | |
| | | | | 0.55 | 0.4 | **16.0 (1.1)** | 16.2 (1.6) | 17.1 (2.2) | 18.3 (1.1) | 20.5 (0.9) | ↓ |
| | Deer, Horse | Plane, Auto | Cat, Dog | 0.5 | 0.5 | 14.1 (0.9) | 14.9 (1.5) | **11.2 (0.8)** | 33.1 (2.7) | 47.5 (2.0) | 7.7 (0.4) |

A16

### E.3  Case Study: Arbitrary Adversarial Concept Drift

This section's experiments model adversarial settings where the positive class-conditional distribution shifts significantly faster than the negative class distribution. As explained in Section D.2.2, the training set was composed of spam and ham emails from the TREC0$\underline{5}$ dataset; the test set was composed of spam and ham emails from the TREC0$\underline{7}$ dataset. The two dataset's ham emails are quite different – TREC05 relies heavily on Enron emails while TREC07 contains many emails received on a university email server. We are therefore confident our fixed-negative-distribution assumption in Eq. (7) does not hold.

Table 14 and Figure 4 compare our methods to PUc and nnPU across three different training priors ($\pi_{tr}$). Under all three experimental conditions, our three methods outperformed both PUc and nnPU* according to a 1% paired t-test. PU2wUU was the top performer for all experiments. As evidenced by the PN misclassification rate, a highly accurate classifier can be constructed for this dataset. Similarly, $\hat{\sigma}$ accurately labels $\mathcal{X}_{tr\text{-}u}$. The resulting surrogate negative set is more useful than $\mathcal{X}_p$ to classify the spam emails from the test distribution. PU2aPNU performed slightly worse than PU2wUU because the spam emails in $\mathcal{X}_p$ are of very limited value due to the significant adversarial concept drift.[11]

Table 14: Inductive misclassification rate (%) mean and standard deviation over 100 trials for arbitrary adversarial concept drift on the TREC spam email datasets. In all experiments, our three methods – PURR, PU2aPNU, & PU2wUU – (which are denoted by [†]) statistically outperformed PUc and nnPU* according to a paired t-test ($p < 0.01$) with PU2wUU the top performer across all training priors ($\pi_{tr}$).

| Train | | Test | | $\pi_{tr}$ | $\pi_{te}$ | | Two-Step (PU2) | | Baselines | | Ref. |
|---|---|---|---|---|---|---|---|---|---|---|---|
| Pos. | Neg. | Pos. | Neg. | | | PURR[†] | aPNU[†] | wUU[†] | PUc | nnPU* | PN$_{te}$ |
| 2005 Spam | 2005 Ham | 2007 Spam | 2007 Ham | 0.4 | 0.5 | 26.5 (2.6) | 26.9 (3.1) | **25.1 (3.1)** | 35.2 (11.3) | 40.9 (3.1) | ↑ |
| | | | | 0.5 | 0.5 | 27.5 (3.4) | 28.6 (4.5) | **25.1 (3.3)** | 34.6 (10.2) | 40.5 (2.7) | 0.6 (0.3) |
| | | | | 0.6 | 0.5 | 30.8 (4.2) | 33.0 (5.7) | **29.3 (6.5)** | 38.5 (10.8) | 41.1 (2.9) | ↓ |

Figure 4: Mean inductive misclassification rate (%) over 100 trials for the TREC spam datasets across three training priors ($\pi_{tr}$). Our PU2wUU method was the top performer across all experiments.

### E.4 Identical Positive Supports with Bias

The positive bias applied in this section's experiments is totally different from that in Sections 7.2 and 7.3. Here we mimic situations where the labeled data are complete but non-representative resulting in identical marginal distribution supports but shifts in the marginal distribution's magnitude. We follow the experimental setup described in Sakai and Shimizu [14]'s PUc paper. LIBSVM [30] benchmarks are used exclusively to ensure suitability with the SVM-like PUc; benchmarks "banana," "susy," "ijcnn1," and "a9a" appear in Sakai and Shimizu [14]'s PUc paper.

Sakai and Shimizu's bias operation is based on the median feature vector. Formally, given dataset $\mathcal{X} \subset \mathbb{R}^d$, define $c_{\text{med}}$ as the median of set $\{\|x - \bar{x}\|_2 : x \in \mathcal{X}\}$ where $\|\cdot\|_2$ is the $L_2$ (Euclidean) norm and $\bar{x}$ is $\mathcal{X}$'s mean vector, i.e.,

$$\bar{x} = \frac{1}{|\mathcal{X}|} \sum_{x \in \mathcal{X}} x.$$

Partition $\mathcal{X}$ into subsets $\mathcal{X}_{\text{lo}} \coloneqq \{x \in \mathcal{X} : \|x - \bar{x}\|_2 < c_{\text{med}}\}$ and $\mathcal{X}_{\text{hi}} \coloneqq \mathcal{X} \setminus \mathcal{X}_{\text{lo}}$. Examples in $\mathcal{X}_{\text{p}}$ and $\mathcal{X}_{\text{tr-u}}$ are selected from $\mathcal{X}_{\text{lo}}$ with probability $p = 0.9$ and from $\mathcal{X}_{\text{hi}}$ with probability $1 - p$. $p = 0.1$ is used when constructing $\mathcal{X}_{\text{te-u}}$ and the test set. This bias operation simplifies density-ratio estimation since $\forall_{x \in \mathcal{X}} \ w(x) \in \{\frac{1}{9}, 9\}$. Their setting $\pi_{\text{tr}} = \pi_{\text{te}} = 0.5$ also simplifies density estimation as detailed in Section E.2.

We modified Sakai and Shimizu's setup such that $\mathcal{X}$ was exclusively the original dataset's positive-valued examples. Negative examples were sampled uniformly at random.

**Analysis**  The experiments enumerated in Table 15 and shown visually in Figure 5 used the bias procedure described above on 10 LIBSVM datasets. According to a 1% paired t-test, PURR and PU2aPNU outperformed the baselines, PUc and nnPU*, on all ten benchmarks; PU2wUU outperformed the baselines on nine of ten benchmarks.

PURR was the top performer on three benchmarks; PU2aPNU was the top performer on five benchmarks while PU2wUU was the top performer on two benchmarks. Each estimator is best suited to a different feature dimension range. PURR performed best when the dataset had fewer features (e.g., $<50$) while PU2aPNU performed well when the dimension was moderate. PU2wUU was the top performer when the dimension was large (e.g., $\geq 300$).

Accurate risk estimation is more challenging when the training sets are comparatively small but the feature count is high. We expect that is causing PURR to struggle to reconcile/relate the different labeled losses (e.g., positive-labeled, unlabeled train, unlabeled test) in these higher dimension datasets.

Table 15: Inductive misclassification rate (%) mean and standard deviation over 100 trials with Sakai and Shimizu [14]'s median feature vector-based bias for 10 LIBSVM datasets. Underlining denotes a statistically significant performance improvement versus PUc and nnPU* according to a 1% paired t-test. Boldface indicates each dataset's best performing method. $n_p = 300$ and $n_{\text{tr-u}} = n_{\text{te-u}} = 700$. Datasets are ordered by increasing dimension. Our three methods – PURR, PU2aPNU, and PU2wUU – are denoted with [†].

| Dataset | $d$ | | Two-Step (PU2) | | Baselines | | Ref. |
|---|---|---|---|---|---|---|---|
| | | PURR[†] | aPNU[†] | wUU[†] | PUc | nnPU* | $PN_{\text{te}}$ |
| banana | 2 | 12.9 (2.1) | **11.8 (1.6)** | 13.3 (2.3) | 17.4 (3.4) | 28.8 (3.8) | 8.6 (0.6) |
| cod-rna | 8 | **14.7 (2.6)** | 15.1 (3.2) | 15.5 (2.9) | 25.2 (5.0) | 24.9 (2.3) | 6.5 (0.9) |
| susy | 18 | **24.2 (2.1)** | 25.6 (2.2) | 25.8 (2.2) | 27.3 (4.3) | 45.9 (3.9) | 20.5 (1.3) |
| ijcnn1 | 22 | 22.7 (2.8) | **17.7 (2.8)** | 24.6 (3.1) | 23.9 (3.6) | 34.7 (3.6) | 6.8 (0.8) |
| covtype.b | 54 | **29.5 (2.9)** | 32.5 (3.2) | 29.9 (2.4) | 39.4 (4.2) | 55.5 (2.8) | 22.3 (1.4) |
| phishing | 68 | 11.3 (1.4) | **9.6 (1.0)** | 11.1 (1.8) | 13.8 (4.1) | 22.5 (4.1) | 6.2 (0.6) |
| a9a | 123 | 27.1 (2.1) | **26.6 (1.8)** | 27.1 (2.1) | 32.8 (2.6) | 32.5 (2.3) | 20.6 (1.0) |
| connect4 | 126 | 34.9 (3.1) | **32.9 (2.7)** | 35.0 (2.9) | 37.0 (2.8) | 45.1 (2.6) | 21.6 (1.3) |
| w8a | 300 | 17.2 (2.6) | 21.0 (2.9) | **16.8 (2.9)** | 29.3 (6.2) | 41.1 (4.3) | 6.6 (0.7) |
| epsilon | 2,000 | 33.5 (4.8) | 36.5 (5.0) | **31.5 (1.7)** | 62.8 (6.7) | 64.6 (1.5) | 23.7 (1.1) |

Figure 5: Mean inductive misclassification rate (%) over 100 trials with Sakai and Shimizu [14]'s median feature vector-based bias for the 10 LIBSVM datasets in Section E.4.

### E.5 Comparison to bPU Selection Bias Method PUSB

Recall that baseline PUc is a covariate-shift bPU method. A NeurIPS reviewer requested an experiment comparing our proposed approaches to a selection bias bPU learning baseline. This section compares our algorithms to Kato et al. [13]'s Positive-Unlabeled Selection Bias (PUSB) method.

Let random variable $S \in \{\pm 1\}$ denote whether some training example $(X, Y) \sim p(x, y)$ is labeled. For all types of PU learning (e.g., unbiased, bPU, aPU), it is straightforward that $S = +1$ implies $Y = +1$, i.e.,

$$p(Y = +1 | S = +1) = 1 \tag{28}$$

and

$$p(S = +1 | Y = -1) = 0. \tag{29}$$

PUSB makes what Kato et al. term the *invariance-of-order assumption*. Formally, for any pair of training examples $x_i, x_j \in \mathbb{R}^d$, it holds that

$$p(Y = +1 | x_i) \geq p(Y = +1 | x_j) \iff p(S = +1 | x_i) \geq p(S = +1 | x_j). \tag{30}$$

In words, a training example is at least as likely to be positive-valued as another example if and only if it is at least as likely to be labeled as that other example. As mentioned in Section 4, it is not possible to directly compare our approaches to existing selection bias bPU methods like PUSB. Such bPU learning methods assume access to only a single unlabeled set ($\mathcal{X}_{\text{te-u}}$) drawn from the test distribution while aPU learning provides two unlabeled sets ($\mathcal{X}_{\text{tr-u}}$ and $\mathcal{X}_{\text{te-u}}$).

To ensure a fair comparison, we sought to replicate Kato et al. [13]'s experimental setup as closely as possible provided the constraints of our method – even using their source code[12] verbatim where possible (e.g., PUSB used the `Chainer` [48] neural network framework as specified by Kato et al.). Like in the PUSB paper, we analyzed the performance of all methods on the MNIST [31] dataset. To enrich the comparison, we also consider the drop-in MNIST variants FashionMNIST [44] and KMNIST [45].

**Dataset Construction**    Our experiments exactly duplicate Kato et al.'s procedure for constructing biased-positive set $\mathcal{X}_{\text{p}}$. Specifically, a multilayer perceptron (MLP) with four hidden layers of 300 neurons each and ReLU activation is trained using the PN logistic loss on the dataset's complete training and test sets. $\mathcal{X}_{\text{p}}$ is then selected u.a.r. without replacement from those positive-valued training examples the aforementioned MLP identifies as having the highest positive posterior. Unlabeled test set $\mathcal{X}_{\text{te-u}}$ and the inductive test set are drawn u.a.r. without replacement from the complete training and test sets respectively.

Kato et al. uses the complete MNIST training set as the unlabeled set. Since both PUc and our methods require two unlabeled sets, we cannot follow the same methodology here. Instead, we limit the size of the test unlabeled set and create unlabeled training set $\mathcal{X}_{\text{tr-u}}$ by selecting its positive examples according to the procedure described above for $\mathcal{X}_{\text{p}}$ and selecting its negative-valued examples u.a.r. without replacement from the training set's negative elements. Table 16 details our experiments' positive priors as well as the dataset and mini-batch sizes.

**Hyperparameters**    Identical hyperparameters were used for the MNIST, FashionMNIST, and KMNIST datasets. Our methods, nnPU*, and PN$_{\text{te}}$ used identical hyperparameter settings as those tuned for MNIST in Section 7's experiments.

PUSB's hyperparameters match those specified by Kato et al. for MNIST, e.g., learning rate $\eta = 10^{-5}$ and weight decay $\lambda = 5 \cdot 10^{-3}$. PUSB learners were trained for 250 epochs using the Adam [49] optimizer. As in the original paper, PUSB's neural network had four hidden layers of 300 neurons each and batch normalization before each ReLU activation.

**Results Analysis**    Table 17 compares the performance of our methods – PURR, PU2aPNU, and PU2wUU – to the extended baseline set – PUc, PUSB, and nnPU* – for the experimental setup described above. To mitigate the effects of different unlabeled set configurations, our experiments tested two unlabeled set sizes, with one size half the other (Table 16). PUc's and our methods' results in Table 17a used 6,000 total unlabeled samples, i.e., the same quantity used by PUSB in Table 17b. Figure 6 visualizes these cross-table, matching-unlabeled-set-size results graphically. For nnPU* in Figure 6, three unlabeled set configurations are considered namely, nnPU$_{\text{te}}$ and nnPU$_{\text{te} \cup \text{tr}}$ with $|\mathcal{X}_{\text{tr-u}}| = |\mathcal{X}_{\text{te-u}}| = 3{,}000$ as well as nnPU$_{\text{te}}$ with $|\mathcal{X}_{\text{te-u}}| = 6{,}000$. Observe that these are the only nnPU* configurations using at most 6,000 unlabeled examples.

As mentioned in Section 7.2, when there is little to no dataset shift, shift-unaware methods (e.g., nnPU) are expected to be the top performer. As an intuition why – when a method searches for a non-existent phenomenon, any patterns found will not generalize. Since nnPU* is the top performer for MNIST and KMNIST despite not accounting for shift at all, it then stands to reason that Kato et al. [13]'s invariance-of-order bias induces only a small shift here.

We saw in Section 7.2 that for such mild shifts (e.g., no bias), PUc often outperforms our methods. We generally see the same trend in Table 17 for MNIST and KMNIST (primary exception being PU2aPNU for MNIST). This is again expected. Under mild shifts, covariate shift's consistent input-output relation assumption generally holds. In addition, importance function $w(x) \approx 1$ for all $x$ under limited bias, in which case PUc simplifies to essentially standard nnPU.

All of our methods outperformed all baselines for FashionMNIST. What is more, our methods outperformed PUSB in all but one case (PU2wUU for KMNIST) even after accounting for unlabeled set size (Figure 6). In fact, PUSB always lagged nnPU*. This hints at a level of brittleness for Kato et al.'s method since PUSB struggled on a bias condition it specifically targets.

Table 16: Positive priors, dataset sizes (including the validation set), and mini-batch sizes for Section E.5's invariance of order selection bias experiments. The first column lists the table where each setup's corresponding results are enumerated.

| | Prior | | Dataset Size | | | | Batch Size | | | |
|---|---|---|---|---|---|---|---|---|---|---|
| | $\pi_{tr}$ | $\pi_{te}$ | $n_p$ | $n_{tr\text{-}u}$ | $n_{te\text{-}u}$ | $n_{Test}$ | $g(x)$ | $\hat{\sigma}(x)$ | PUSB | $PN_{te}$ |
| Table 17a | 0.5 | 0.5 | 1,000 | 3,000 | 3,000 | 5,000 | 2,500 | 2,500 | 1,000 | 2,000 |
| Table 17b | 0.5 | 0.5 | 1,000 | 6,000 | 6,000 | 5,000 | 5,000 | 5,000 | 1,000 | 4,000 |

Table 17: Inductive misclassification rate (%) mean and standard deviation over 100 trials for the experiments using Kato et al. [13]'s invariance-of-order setup on the MNIST, FashionMNIST, and KMNIST datasets. Bold face denotes each dataset's best performing method according to mean misclassification rate. Our methods – PURR, PU2aPNU, and PU2wUU – are denoted with [†].

(a) $|\mathcal{X}_{tr\text{-}u}| = |\mathcal{X}_{te\text{-}u}| = 3{,}000$

| Dataset | | Two-Step (PU2) | | Baselines | | | Ref. |
|---|---|---|---|---|---|---|---|
| | PURR[†] | aPNU[†] | wUU[†] | PUc | PUSB | nnPU* | $PN_{te}$ |
| MNIST | 13.0 (2.3) | 9.7 (1.3) | 11.6 (1.4) | 10.6 (1.1) | 15.9 (1.0) | **8.8 (0.9)** | 3.6 (0.3) |
| FashionMNIST | 6.4 (1.4) | **5.3 (0.7)** | 5.9 (1.0) | 9.0 (1.1) | 10.5 (1.2) | 8.5 (1.3) | 3.5 (0.3) |
| KMNIST | 31.6 (2.4) | 29.7 (2.2) | 33.7 (2.3) | 27.3 (1.4) | 33.4 (1.2) | **24.6 (1.4)** | 16.4 (0.8) |

(b) $|\mathcal{X}_{tr\text{-}u}| = |\mathcal{X}_{te\text{-}u}| = 6{,}000$

| Dataset | | Two-Step (PU2) | | Baselines | | | Ref. |
|---|---|---|---|---|---|---|---|
| | PURR[†] | aPNU[†] | wUU[†] | PUc | PUSB | nnPU* | $PN_{te}$ |
| MNIST | 10.5 (1.8) | 8.5 (1.2) | 9.3 (1.0) | 10.2 (1.1) | 14.2 (1.0) | **8.0 (0.9)** | 2.8 (0.2) |
| FashionMNIST | 5.6 (1.3) | **4.8 (0.6)** | 5.0 (0.8) | 9.1 (1.2) | 10.0 (1.2) | 8.2 (1.2) | 3.1 (0.2) |
| KMNIST | 29.6 (2.2) | 29.3 (2.1) | 32.0 (2.2) | 27.0 (1.4) | 32.1 (1.2) | **24.1 (1.4)** | 13.7 (0.7) |

Figure 6: Mean inductive misclassification rate (%) over 100 trials for the experiments using Kato et al. [13]'s invariance-of-order setup on the MNIST, FashionMNIST, and KMNIST datasets. All learners saw up to 6,000 total unlabeled examples with results cross-compiled between Table 17a (for our methods and PUc) and Table 17b (for PUSB). Here nnPU* considers three different unlabeled set configurations as described in Section E.5.

### E.6 Empirical Comparison of Absolute-Value and Non-Negativity Corrections

Section 3 describes our streamlined absolute-value correction to address PU learning overfitting. This section compares our simpler absolute-value correction to Kiryo et al. [8]'s non-negativity correction using max and "defitting."

#### E.6.1 Ordinary Positive-Unlabeled Learning Performance Without Distributional Shift

We first consider a direct comparison of nnPU and abs-PU on *unshifted* data. $\mathcal{X}_\text{p}$ and $\mathcal{X}_\text{u}$ are constructed identically to the procedure used to construct the positive-labeled and unlabeled-train datasets in our aPU learning experiments. Unlike before, the inductive test set is now drawn from the *training* distribution. We then trained classifiers using nnPU and abs-PU with the sigmoid loss. In all experiments, the classifiers had identical initial weights and were trained on identical dataset splits.

Hyperparameters (including $\gamma$) were tuned using nnPU; these identical hyperparameters were then used for abs-PU (i.e., not in any way tuned for absolute-value correction). Therefore, the results represent the *performance floor* when transitioning from nnPU to abs-PU. This was done due to time constraints.

Table 18 compares abs-PU and nnPU for the datasets in Sections 7.2[13], 7.3, and E.4. We also report the difference between abs-PU and nnPU with a positive number indicating that abs-PU performed better that nnPU.

abs-PU was the top performer on eight of fourteen benchmarks and tied with nnPU on two others; the results are generally too close to be statistically significant. Both methods had comparable variances. In summary, abs-PU is both simpler and saw similar or slightly better performance than nnPU on unbiased data, even under conditions (i.e., hyperparameters) that favor nnPU.

Table 18: Comparison of inductive misclassification rate (%) mean and standard deviation over 100 trials for abs-PU and nnPU on unshifted data. Boldface denotes the best performing algorithm according to mean misclassification rate. For the difference (Diff.) column, a positive value denotes that abs-PU outperformed nnPU.

| Dataset | abs-PU | nnPU | nnPU – abs-PU (Diff.) |
|---|---|---|---|
| MNIST | 6.6 (0.7) | **6.5 (0.7)** | –0.1 ( 0 ) |
| 20 Newsgroups | **13.3 (1.3)** | 13.5 (1.2) | 0.2 (–0.1) |
| CIFAR10 | **12.4 (0.7)** | **12.4 (0.7)** | 0 ( 0 ) |
| TREC Spam | **2.0 (1.0)** | 2.1 (0.9) | 0.1 (–0.1) |
| banana | **10.5 (1.0)** | **10.5 (1.1)** | 0 ( 0.1) |
| cod-rna | **10.3 (1.8)** | 10.4 (2.0) | 0.1 ( 0.2) |
| susy | 28.8 (1.7) | **28.7 (1.8)** | –0.1 ( 0.1) |
| ijcnn1 | **10.1 (1.4)** | 10.2 (1.5) | 0.1 ( 0.1) |
| covtype.b | **32.8 (2.2)** | 33.3 (2.1) | 0.5 (–0.1) |
| phishing | 8.6 (1.3) | **8.5 (1.2)** | –0.1 (–0.1) |
| a9a | **15.9 (1.1)** | 16.0 (1.2) | 0.1 ( 0.1) |
| connect4 | 24.6 (2.2) | **24.4 (2.0)** | –0.2 (–0.2) |
| w8a | **17.8 (1.6)** | 17.9 (1.6) | 0.1 ( 0 ) |
| epsilon | **31.1 (1.4)** | 31.2 (1.7) | 0.1 ( 0.3) |

#### E.6.2 Ordinary Positive-Unlabeled Learning Performance Under Distribution Shift

The previous section compared the performance of nnPU and abs-PU under ideal conditions, i.e., no positive shift. This section compares nnPU and abs-PU *with positive shift*, specifically under the aPU learning conditions we use in our experimental evaluation.

Like in the previous section, all classifiers in each experimental trial had identical initial weights and saw identical dataset splits. Hyperparameters (including $\gamma$) were tuned using nnPU; these identical hyperparameters were then used for abs-PU (i.e., not in any way tuned for absolute-value correction). Therefore, the results again represent the *performance floor* if transitioning from nnPU to abs-PU. This choice was made due to limited time.

Recall from Section 7 that evaluation baseline nnPU* considers two nnPU-based classifiers – one trained with unlabeled set $\mathcal{X}_\text{te-u}$ and the other trained with unlabeled set $\mathcal{X}_\text{tr-u} \cup \mathcal{X}_\text{te-u}$ (using the true composite prior), and we report whichever of those two classifiers performed best on average. In this section, we introduce abs-PU*, which like nnPU*, considers

two classifiers separately trained with the different unlabeled set configurations: $\mathcal{X}_{\text{te-u}}$ and $\mathcal{X}_{\text{tr-u}} \cup \mathcal{X}_{\text{te-u}}$. The only difference is that abs-PU*, as its name would suggest, uses our abs-PU risk estimator. We specifically separated this section to delineate the baseline performance of our contribution (abs-PU) versus existing methods (nnPU).

Table 19 compares abs-PU* and nnPU* for the extended set of experiments in Table 13 (see Section E.2). Recall that those experiments tested cases where some positive subclasses exist only in the test distribution. Similar to Table 18, a positive value in the column labeled "Diff." denotes that abs-PU* performed better than nnPU*.

For multiple positive-train ($P_{\text{train}}$) class configurations (e.g., MNIST $P_{\text{train}} = \{1, 3, 5\}$), abs-PU* and nnPU* exhibited similar performance. When there was a large difference between the two methods (e.g., 20 Newsgroups $P_{\text{train}} = \{\text{misc}, \text{rec}\}$), abs-PU* had significantly better mean accuracy – reducing the misclassification rate by multiple percentage points. The difference between the methods was most pronounced when $P_{\text{train}}$ and $P_{\text{test}}$ are disjoint.

These results indicate that in some cases, abs-PU* is learning decision boundaries that better generalize to *unseen types of data*. To be clear, this does not apply to all datasets (CIFAR10 exhibited little difference between the methods except when the positive supports were disjoint) nor even to all class partitions within a dataset (see MNIST positive-train classes $\{7, 9\}$ versus $\{1, 3, 5\}$). It should also be noted that missing positive subclasses is a more extreme form of positive shift. The next set of results considers the more mild case of marginal-distribution magnitude shifts.

Table 19: Comparison of inductive misclassification rate (%) mean and standard deviation over 100 trials for abs-PU* and nnPU* for the experimental shift tasks (eight per dataset) in Table 13 with partially/fully disjoint positive class supports. Boldface denotes the best performing task according to mean misclassification rate. For the difference column, a positive value indicates abs-PU* outperformed nnPU*.

| | N | $P_{\text{test}}$ | $P_{\text{train}}$ | $\pi_{\text{tr}}$ | $\pi_{\text{te}}$ | abs-PU* | nnPU* | Diff. |
|---|---|---|---|---|---|---|---|---|
| **MNIST** | 0, 2, 4, 6, 8 | 1, 3, 5, 7, 9 | 7, 9 | 0.29 | 0.5 | **34.4 (2.6)** | 36.7 (2.7) | 2.3 ( 0.1) |
| | | | | 0.5 | 0.5 | **33.1 (2.3)** | 35.1 (2.5) | 2.0 ( 0.2) |
| | | | | 0.71 | 0.5 | **32.7 (2.2)** | 34.5 (2.9) | 1.8 ( 0.7) |
| | | | 1, 3, 5 | 0.38 | 0.5 | **25.9 (1.2)** | **25.9 (1.1)** | 0  (–0.1) |
| | | | | 0.5 | 0.5 | 27.1 (1.3) | **26.9 (1.2)** | –0.2 (–0.1) |
| | | | | 0.63 | 0.5 | 28.7 (1.1) | **28.5 (1.2)** | –0.2 ( 0.1) |
| | 0, 2 | 5, 7 | 1, 3 | 0.5 | 0.5 | **25.7 (6.9)** | 30.9 (5.3) | 5.2 (–1.6) |
| **20 Newsgroups** | sci, soc, talk | alt, comp, misc, rec | misc, rec | 0.37 | 0.56 | **27.0 (1.9)** | 28.8 (1.3) | 1.8 (–0.6) |
| | | | | 0.56 | 0.56 | **26.0 (1.7)** | 28.8 (1.7) | 2.8 ( 0  ) |
| | | | | 0.65 | 0.56 | **25.9 (1.7)** | 29.0 (1.8) | 3.1 ( 0.1) |
| | | | comp | 0.37 | 0.56 | **31.2 (0.7)** | 31.4 (0.7) | 0.2 ( 0  ) |
| | | | | 0.56 | 0.56 | **31.0 (0.9)** | 31.2 (0.8) | 0.2 (–0.1) |
| | | | | 0.65 | 0.56 | **31.0 (0.8)** | 31.3 (0.7) | 0.3 (–0.1) |
| | misc, rec | soc, talk | alt, comp | 0.55 | 0.46 | **34.6 (5.0)** | 35.3 (5.2) | 0.7 ( 0.2) |
| **CIFAR10** | Bird, Cat, Deer, Dog, Frog, Horse | Plane, Auto, Ship, Truck | Plane | 0.14 | 0.4 | **26.5 (1.0)** | 26.7 (1.0) | 0.2 ( 0  ) |
| | | | | 0.4 | 0.4 | **27.4 (1.0)** | **27.4 (1.0)** | 0  ( 0  ) |
| | | | | 0.6 | 0.4 | **28.3 (1.1)** | 28.4 (1.0) | 0.1 (–0.1) |
| | | | Auto, Truck | 0.25 | 0.4 | **20.3 (0.8)** | **20.3 (0.8)** | 0  ( 0  ) |
| | | | | 0.4 | 0.4 | 20.4 (0.9) | **20.3 (0.8)** | –0.1 (–0.1) |
| | | | | 0.55 | 0.4 | 20.9 (0.9) | **20.5 (0.9)** | –0.4 ( 0  ) |
| | Deer, Horse | Plane, Auto | Cat, Dog | 0.5 | 0.5 | **44.6 (1.8)** | 47.5 (2.0) | 2.9 ( 0.2) |

Table 20 compares abs-PU* and nnPU* for the 10 LIBSVM datasets in Table 15 (see Section E.4). Recall that in these experiments, the positive-train and positive-test class-conditionals have identical supports. For seven of ten benchmarks, abs-PU* had better mean performance than nnPU* and had equivalent performance on one other benchmark. abs-PU* did have generally higher result variance. For some benchmarks (e.g., `ijcnn1`, `covtype.b`, `epsilon`, etc.), the change in variance was more than offset by the improvement in mean accuracy. Had the abs-PU* learning rates been tuned directly instead of using nnPU*'s hyperparameter settings, we expect this variance difference would have been mitigated. Again however, limited time prevented that experiment.

In summary, abs-PU*'s performance is comparable or slightly/significantly better than that of nnPU* under aPU learning conditions that are deleterious to ordinary PU risk estimators but that may be more realistic to real-world data.

Table 20: Comparison of inductive misclassification rate (%) mean and standard deviation over 100 trials for abs-PU\*
and nnPU\* for the 10 LIBSVM datasets in Table 15 under Sakai and Shimizu [14]'s mean feature vector bias. Boldface
denotes the best performing task according to mean misclassification rate. For the difference column, a positive value
indicates abs-PU\* outperformed nnPU\*.

| Dataset | $d$ | abs-PU* | nnPU* | Diff. |
|---|---|---|---|---|
| banana | 2 | **28.5 (4.1)** | 28.8 (3.8) | 0.3 (–0.3) |
| cod-rna | 8 | 25.1 (2.5) | **24.9 (2.3)** | –0.2 (–0.2) |
| susy | 18 | **45.9 (3.9)** | **45.9 (3.9)** | 0 ( 0 ) |
| ijcnn1 | 22 | **33.3 (3.9)** | 34.7 (3.6) | 1.4 (–0.3) |
| covtype.b | 54 | **54.6 (3.1)** | 55.5 (2.8) | 0.9 (–0.3) |
| phishing | 68 | 22.9 (4.2) | **22.5 (4.1)** | –0.4 (–0.1) |
| a9a | 123 | **32.0 (2.5)** | 32.5 (2.3) | 0.5 (–0.2) |
| connect4 | 126 | **44.9 (3.1)** | 45.1 (2.6) | 0.2 (–0.5) |
| w8a | 300 | **40.0 (4.0)** | 41.1 (4.3) | 1.1 ( 0.3) |
| epsilon | 2,000 | **64.1 (1.4)** | 64.6 (1.5) | 0.5 ( 0.1) |

### E.6.3 Effect of Absolute-Value Correction on Our aPU Learning Methods

This section examines the effect of using absolute-value correction over non-negativity correction for our three
aPU learning methods – PURR, PU2aPNU, and PU2wUU. Recall that non-negativity correction requires custom
ERM algorithms to support "defitting." Section C describes our methods' custom ERM frameworks when using
non-negativity.

Due to time constraints, hyperparameter tuning was performed using non-negativity correction with the same hy-
perparameters used for the absolute-value based methods. Therefore, these results maximally favor the baseline of
non-negativity correction.

Table 21's experiments are identical to Table 13 in Section E.2. "abs" denotes our standard aPU learning methods
(see Sections 5 and 6) while "nn" denotes our methods modified to use Kiryo et al. [8]'s non-negativity correction.
For MNIST, neither absolute-value correction nor non-negativity clearly outperformed the other. For the more
challenging 20 Newsgroups and CIFAR10 datasets, absolute-value correction had consistently better performance than
non-negativity. The only exception were the disjoint support experiments and one experimental setup for PU2wUU
on 20 Newsgroups. Although not shown in Table 13 due to limited space, both correction strategies had comparable
variance.

Table 22's experiments match the experimental conditions for the 10 LIBSVM datasets in Table 15 from Section E.4. Bi-
asing follows Sakai and Shimizu [14]'s median feature vector-based approach. Neither absolute-value nor non-negativity
correction consistently outperformed the other in these LIBSVM experiments. Note though that since absolute-value
correction is a simpler method with one less hyperparameter, $\gamma$, to tune, comparable performance implicitly favors
absolute-value correction over non-negativity.

Table 21: Comparison of mean inductive misclassification rate (%) over 100 trials for the non-overlapping support experiments in Table 13 when using absolute-value (abs) and non-negativity (nn) corrections for our aPU learning methods. The best performing method (according to mean misclassification rate) is shown in bold. A positive difference (Diff.) denotes that our absolute-value correction had better performance. Result standard deviations are comparable for both correction methods but are not shown here to improve table clarity.

| | $P_{test}$ | $P_{train}$ | $\pi_{tr}$ | $\pi_{te}$ | PURR | | | PU2aPNU | | | PU2wUU | | |
|---|---|---|---|---|---|---|---|---|---|---|---|---|---|
| | | | | | abs | nn | Diff. | abs | nn | Diff. | abs | nn | Diff. |
| **MNIST** | 1, 3, 5, 7, 9 | $P_{test}$ | 0.5 | 0.5 | **10.0** | 10.2 | 0.2 | 10.0 | **9.8** | −0.2 | **11.6** | 11.7 | 0.1 |
| | | 7, 9 | 0.29 | 0.5 | 6.8 | **6.6** | −0.2 | **5.3** | **5.3** | 0 | **6.0** | **6.0** | 0 |
| | | | 0.5 | 0.5 | **9.4** | 9.4 | 0 | **7.1** | **7.1** | 0 | **8.3** | **8.3** | 0 |
| | | | 0.71 | 0.5 | **14.0** | 14.6 | 0.6 | **11.1** | 11.3 | 0.2 | **14.8** | 15.2 | 0.4 |
| | | 1, 3, 5 | 0.38 | 0.5 | 8.1 | **8.0** | −0.1 | **6.5** | **6.5** | 0 | **7.6** | 7.7 | 0.1 |
| | | | 0.5 | 0.5 | 10.0 | **9.9** | −0.1 | **8.4** | **8.4** | 0 | **10.2** | **10.2** | 0 |
| | | | 0.63 | 0.5 | **12.5** | 12.9 | 0.4 | **11.4** | **11.4** | 0 | **14.3** | 14.5 | 0.2 |
| | 5, 7 | 1, 3 | 0.5 | 0.5 | 4.0 | **3.9** | −0.1 | **3.6** | **3.6** | 0 | **3.1** | 3.2 | 0.1 |
| **20 Newsgroups** | alt, comp, misc, rec | $P_{test}$ | 0.56 | 0.56 | **15.4** | 15.5 | 0.1 | **14.9** | 15.0 | 0.1 | **16.7** | **16.7** | 0 |
| | | misc, rec | 0.37 | 0.56 | **13.9** | 13.9 | 0 | **12.8** | **12.8** | 0 | **14.3** | **14.3** | 0 |
| | | | 0.56 | 0.56 | **17.5** | 17.7 | 0.2 | **13.5** | **13.5** | 0 | **15.1** | **15.1** | 0 |
| | | | 0.65 | 0.56 | **20.2** | 20.8 | 0.6 | **14.0** | **14.0** | 0 | **15.9** | **15.9** | 0 |
| | | comp | 0.37 | 0.56 | **13.3** | 13.3 | 0 | **13.7** | **13.7** | 0 | 14.5 | **14.4** | −0.1 |
| | | | 0.56 | 0.56 | **16.0** | 16.5 | 0.5 | **14.9** | **14.9** | 0 | **15.7** | **15.7** | 0 |
| | | | 0.65 | 0.56 | **19.2** | 19.6 | 0.4 | **15.6** | **15.6** | 0 | **16.5** | **16.5** | 0 |
| | soc, talk | alt, comp | 0.55 | 0.46 | 5.9 | **5.8** | −0.1 | **7.1** | **7.1** | 0 | **5.6** | 5.7 | 0.1 |
| **CIFAR10** | Plane, Auto, Ship, Truck | $P_{test}$ | 0.4 | 0.4 | **14.1** | 14.3 | 0.2 | **14.2** | 14.4 | 0.2 | **15.4** | 15.8 | 0.4 |
| | | Plane | 0.14 | 0.4 | **11.9** | 12.0 | 0.1 | **11.9** | 12.0 | 0.1 | **12.4** | **12.4** | 0 |
| | | | 0.4 | 0.4 | **13.8** | 14.0 | 0.2 | **14.5** | 14.6 | 0.1 | **15.1** | 15.5 | 0.4 |
| | | | 0.6 | 0.4 | **16.1** | 16.6 | 0.5 | **16.7** | 17.1 | 0.4 | **20.0** | 20.2 | 0.2 |
| | | Auto, Truck | 0.25 | 0.4 | **12.7** | 12.8 | 0.1 | **12.4** | 12.5 | 0.1 | **12.8** | 13.0 | 0.2 |
| | | | 0.4 | 0.4 | **14.1** | 14.3 | 0.2 | **13.9** | 14.0 | 0.1 | **14.4** | 14.6 | 0.2 |
| | | | 0.55 | 0.4 | **16.0** | 16.4 | 0.4 | **16.2** | 16.3 | 0.1 | **17.1** | 17.4 | 0.3 |
| | Plane, Auto | Cat, Dog | 0.5 | 0.5 | 14.1 | **14.0** | −0.1 | 14.9 | **14.8** | −0.1 | **11.2** | 11.3 | 0.1 |

Table 22: Comparison of inductive misclassification rate (%) mean and standard deviation over 100 trials for Table 15's LIBSVM dataset experiments using Sakai and Shimizu's mean feature vector biasing with absolute-value (abs) and non-negativity (nn) corrections for our aPU learning methods. The best performing method (according to mean misclassification rate) is shown in bold. A positive difference (Diff.) denotes that our absolute-value correction had better performance than non-negativity correction.

| Dataset | PURR | | | PU2aPNU | | | PU2wUU | | |
|---|---|---|---|---|---|---|---|---|---|
| | abs | nn | Diff. | abs | nn | Diff. | abs | nn | Diff. |
| banana | **12.9 (2.1)** | **12.9 (2.2)** | 0 ( 0.1) | 11.8 (1.6) | **11.7 (1.6)** | −0.1 ( 0 ) | **13.3 (2.3)** | 14.0 (2.3) | 0.7 ( 0 ) |
| cod-rna | 14.7 (2.6) | **14.6 (2.9)** | −0.1 ( 0.3) | **15.1 (3.2)** | **15.1 (3.2)** | 0 ( 0 ) | **15.5 (2.9)** | **15.5 (3.3)** | 0 ( 0.4) |
| susy | **24.2 (2.1)** | 24.6 (2.1) | 0.4 ( 0 ) | **25.6 (2.2)** | **25.6 (2.2)** | 0 ( 0 ) | **25.8 (2.2)** | 26.0 (2.1) | 0.2 (−0.1) |
| ijcnn1 | **22.7 (2.8)** | 23.0 (2.8) | 0.3 ( 0 ) | **17.7 (2.8)** | 19.0 (2.9) | 1.3 ( 0.1) | **24.6 (3.1)** | 24.9 (2.9) | 0.3 (−0.2) |
| covtype.b | **29.5 (2.9)** | 29.6 (2.9) | 0.1 ( 0 ) | **32.5 (3.2)** | 32.6 (3.1) | 0.1 (−0.1) | **29.9 (2.4)** | 30.1 (2.7) | 0.2 ( 0.3) |
| phishing | **11.3 (1.4)** | 11.9 (1.4) | 0.6 ( 0 ) | **9.6 (1.0)** | **9.6 (1.0)** | 0 ( 0 ) | **11.1 (1.8)** | 11.7 (1.9) | 0.6 ( 0.1) |
| a9a | 27.1 (2.1) | **27.0 (2.1)** | −0.1 ( 0 ) | 26.6 (1.8) | **26.5 (1.8)** | −0.1 ( 0 ) | 27.1 (2.1) | **27.0 (2.0)** | −0.1 (−0.1) |
| connect4 | 34.9 (3.1) | **34.2 (2.6)** | −0.7 (−0.5) | **32.9 (2.7)** | 33.0 (2.7) | 0.1 ( 0 ) | 35.0 (2.9) | **34.9 (2.6)** | −0.1 (−0.3) |
| w8a | 17.2 (2.6) | **17.1 (2.4)** | −0.1 (−0.2) | 21.0 (2.9) | **20.3 (2.9)** | −0.7 ( 0 ) | **16.8 (2.9)** | 18.4 (2.7) | 1.6 (−0.2) |
| epsilon | 33.5 (4.8) | **32.7 (3.1)** | −0.8 (−1.7) | **36.5 (5.0)** | 37.8 (6.9) | 1.3 ( 1.9) | 31.5 (1.7) | **31.3 (1.7)** | −0.2 ( 0 ) |

## E.7 Alternate Methods for Step #1 of Our Two-Step Methods

Recall from Section 5 that our two-step methods' first step transform unlabeled training set $\mathcal{X}_{\text{tr-u}}$ into surrogate negative set $\widetilde{\mathcal{X}}_{\text{n}}$ by *soft weighting* each $x \in \mathcal{X}_{\text{tr-u}}$ using classifier

$$\hat{\sigma}_{\text{soft}}(x) := \hat{\sigma}(x) \approx p_{\text{tr}}(Y = -1 | x). \tag{31}$$

In this section, we propose and empirically evaluate two alternative step #1 methods – hard and top-k weighting. Regardless of which step #1 method is used to create $\widetilde{\mathcal{X}}_{\text{n}}$, no changes are required to our step #2 risk estimators – wUU and aPNU.

### E.7.1 Overview of the Alternate Step #1 Methods

**Hard Weighting** Guo et al. [50] show that modern neural networks are generally poorly calibrated and tend to report "peaky" confidence estimates. $\hat{\sigma}$ is vulnerable to similar "peaky" behavior. *Hard weighting* assigns each unlabeled training example, $x \in \mathcal{X}_{\text{tr-u}}$, weight

$$\hat{\sigma}_{\text{hard}}(x) := \lfloor \hat{\sigma}(x) \rceil \tag{32}$$

where for $a \in \mathbb{R}$, $\lfloor a \rceil$ rounds $a$ to the nearest integer (i.e., 0 or 1 for probabilistic classifier $\hat{\sigma}$).

Hard weighting simulates worst-case "peaked" behavior. Although not statistically consistent for non-separable data, hard weighting may sometimes outperform soft-weighting due to its thresholding effect.

**Top-k Weighting** To broadly summarize Guo et al.'s primary contribution, neural network probability estimates may be inaccurate. Our *top-k weighting* method attempts to overcome that inaccuracy by focusing, not on the specific probability values predicted by $\hat{\sigma}$, but instead on the *ordering* of those posterior estimates.

By definition, the expected number of positive-labeled examples in $\mathcal{X}_{\text{tr-u}}$ is $\pi_{\text{tr}} \cdot n_{\text{tr-u}}$, where $n_{\text{tr-u}} := |\mathcal{X}_{\text{tr-u}}|$ is the unlabeled training set size and $\pi_{\text{tr}}$ is the positive training prior. Define $k := \lfloor \pi_{\text{tr}} \cdot n_{\text{tr-u}} \rfloor \in \mathbb{Z}_+$. After training $\hat{\sigma}$ (same as before), let set $\mathcal{X}_{\text{tr-u-k}}$ be the k examples in $\mathcal{X}_{\text{tr-u}}$ with the highest predicted posteriors according to $\hat{\sigma}$. Top-k weighting assigns weight 1 to any $x \in \mathcal{X}_{\text{tr-u-k}}$ and weight 0 to any $x \in (\mathcal{X}_{\text{tr-u}} \setminus \mathcal{X}_{\text{tr-u-k}})$. Formally, for any $x \in \mathcal{X}_{\text{tr-u}}$,

$$\hat{\sigma}_{\text{top-k}}(x) := \begin{cases} 1 & x \in \mathcal{X}_{\text{tr-u-k}} \\ 0 & \text{Otherwise} \end{cases}. \tag{33}$$

Observe that top-k weighting uses strictly more information than both soft and hard weighting. However, by relying on $\pi_{\text{tr}}$ to estimate k, top-k weighting is generally more deleteriously affected by misestimation of $\pi_{\text{tr}}$.

### E.7.2 Step #1 Labeling Accuracy

These experiments examine how accurately our three proposed step #1 methods label $\mathcal{X}_{\text{tr-u}}$. The labeling error rate is defined as

$$\text{Error Rate}_{\mathcal{M}} := \frac{100\%}{n_{\text{tr-u}}} \sum_{x \in \mathcal{X}_{\text{tr-u}}} \frac{|2\hat{\sigma}_{\mathcal{M}}(x) - 1 - y_x|}{2}, \tag{34}$$

where $y_x \in \{\pm 1\}$ is unlabeled training example $x$'s true (unknown) label and $\mathcal{M} \in \{\text{soft}, \text{hard}, \text{top-k}\}$ denotes the step #1 method. For hard and top-k weighting, Eq. (34) corresponds to their (scaled) transductive misclassification rate on $\mathcal{X}_{\text{tr-u}}$. Note that the difference between the soft and hard weightings' labeling error rates is indicative of the "peakiness" of $\hat{\sigma}$, with a smaller gap indicating that $\hat{\sigma}$'s estimates are more peaked.

For all experiments in this section, the three step #1 methods saw identical dataset splits and used the same initial model parameters.

**Analysis** Table 23 compares the three weighting methods' step #1 labeling error rate for the 10 LIBSVM datasets in Table 15 (see Section E.4). Recall that in these experiments, the positive-train and positive-test class conditionals have identical supports. The step #1 methods' labeling error rates varied widely from around 10% on the `phishing` dataset to 30–40% for the `covtype.b` and `epsilon` datasets.

Recall from Table 15 that PURR was the top performer for the `cod-rna`, `susy`, and `covtype.b` datasets. The step #1 labeling error on those three datasets ranged from moderate to poor. However, `epsilon` had soft weighting's worst step #1 labeling error rate yet PU2wUU still outperformed PURR (see Table 15). This demonstrates that step #1 labeling accuracy alone does not determine which algorithm class, i.e., two-step or joint, is best.

Table 23: Comparison of the soft, hard, and top-k weighting schemes' step #1 labeling error rate mean and standard deviation across 100 trials for the 10 LIBSVM datasets in Table 15.

| Dataset | $d$ | Soft | Hard | Top-k |
|---|---|---|---|---|
| banana | 2 | 20.1 (3.8) | 13.2 (1.8) | 12.4 (1.8) |
| cod-rna | 8 | 20.8 (4.0) | 13.2 (1.9) | 12.7 (1.7) |
| susy | 18 | 39.6 (2.7) | 30.7 (2.4) | 30.6 (2.4) |
| ijcnn1 | 22 | 27.0 (4.5) | 19.3 (2.5) | 15.8 (2.7) |
| covtype.b | 54 | 44.1 (4.2) | 37.1 (3.3) | 34.9 (2.7) |
| phishing | 68 | 13.5 (3.6) | 10.4 (1.5) | 9.6 (1.1) |
| a9a | 123 | 24.4 (3.8) | 17.4 (1.4) | 18.1 (1.8) |
| connect4 | 128 | 34.2 (5.5) | 27.8 (4.4) | 24.7 (2.4) |
| w8a | 300 | 27.9 (5.1) | 19.6 (1.6) | 18.5 (1.7) |
| epsilon | 2,000 | 44.2 (1.5) | 32.5 (2.6) | 33.7 (2.0) |

Table 24 compares the three weighting methods' step #1 labeling error rate for the extended set of experiments in Table 13 (see Section E.2). Recall that those experiments, on datasets MNIST, 20 Newsgroups, and CIFAR10, replicated scenarios where some positive subclasses exist only in the test distribution. As expected, the easier MNIST dataset had better step #1 labeling accuracy than the more challenging 20 Newsgroups and CIFAR10 datasets. On the whole, these three datasets had better average step #1 labeling error rate than the 10 LIBSVM datasets discussed above.

### E.7.3  Step #1 Method's Effect on Overall Two-Step Performance

These experiments study how each step #1 method affects our two step methods' – PU2aPNU and PU2wUU – inductive, test (i.e., end-to-end) misclassification rate. As in the previous section, all methods saw identical dataset splits and initial model parameters in each experimental trial.

**Analysis**    Table 25 compares the two-step inductive misclassification rate when using the three step #1 methods for the 10 LIBSVM datasets in Table 15 (see Section E.4). Soft weighting was the best performing method for all ten datasets for PU2aPNU and for seven of ten datasets for PU2wUU. It is also noteworthy that only soft weighting learned a meaningful classifier for the epsilon dataset. In fact, hard and top-k weighting performed worse than random chance for epsilon.

Table 26 compares the two-step, inductive misclassification rate when using the three step #1 methods for the experiments in Table 13 on MNIST, 20 Newsgroups, and CIFAR10 (see Section E.2). For the vast majority of setups, top-k weighting was the best performing method for both PU2aPNU and PU2wUU. Top-k often improved performance over soft weighting by 10–20% or more – in particular for PU2wUU. The one experimental setup where soft weighting consistently performed as well or better than top-k was when the positive train and test supports were disjoint. Observe that in those experiments, the positive and negative classes are composed of fewer constituent labels. As such, we believe that top-k weighting is exacerbating overfitting in those models resulting in the worse performance.

### E.7.4  Discussion

The experiments in the previous subsection demonstrate that the best performing step #1 method is benchmark/setup dependent. If a user is highly confident that their data is readily and easily separable (like MNIST), top-k weighting may perform particularly well. Although not shown here, we empirically observed that misestimation of training prior $\pi_{tr}$ negatively affects top-k weighting's accuracy – many times severely.

If the training datasets (e.g., $\mathcal{X}_p$, $\mathcal{X}_{tr-u}$, and $\mathcal{X}_{te-u}$) are large enough that asymptotic consistency guarantees generally apply, soft weighting may perform best. We made soft-weighting the focus of Section 5 due to its stronger statistical guarantees. Had top-k weighting been used in Section 7.2's experiments instead of soft weighting, our performance advantage over the baselines, PUc and nnPU*, would have widened.

Table 24: Comparison of the soft, hard, and top-k weighting schemes' step #1 labeling error rate mean and standard deviation across 100 trials for Table 13's experiments on partially/fully disjoint positive-class support for MNIST, 20 Newsgroups, and CIFAR10.

| | N | $P_{\text{test}}$ | $P_{\text{train}}$ | $\pi_{\text{tr}}$ | $\pi_{\text{te}}$ | Soft | Hard | Top-k |
|---|---|---|---|---|---|---|---|---|
| MNIST | 0, 2, 4, 6, 8 | 1, 3, 5, 7, 9 | $P_{\text{test}}$ | 0.5 | 0.5 | 16.2 (2.3) | 12.6 (1.0) | 10.6 (1.0) |
| | | | | 0.29 | 0.5 | 11.0 (2.0) | 6.8 (0.8) | 5.4 (0.5) |
| | | | 7, 9 | 0.5 | 0.5 | 10.8 (1.9) | 8.3 (0.9) | 6.5 (0.6) |
| | | | | 0.71 | 0.5 | 11.1 (2.4) | 8.3 (0.5) | 7.7 (0.6) |
| | | | | 0.38 | 0.5 | 12.6 (1.6) | 9.0 (0.9) | 7.3 (0.7) |
| | | | 1, 3, 5 | 0.5 | 0.5 | 14.0 (2.4) | 10.7 (0.9) | 9.0 (1.0) |
| | | | | 0.63 | 0.5 | 15.0 (3.1) | 11.4 (0.7) | 10.3 (1.0) |
| | 0, 2 | 5, 7 | 1, 3 | 0.5 | 0.5 | 8.9 (1.7) | 6.5 (0.7) | 5.4 (0.4) |
| 20 Newsgroups | sci, soc, talk | alt, comp, misc, rec | $P_{\text{test}}$ | 0.56 | 0.56 | 23.1 (4.3) | 16.9 (1.3) | 16.5 (1.3) |
| | | | | 0.37 | 0.56 | 15.5 (1.3) | 12.0 (1.4) | 9.3 (1.3) |
| | | | misc, rec | 0.56 | 0.56 | 14.1 (1.5) | 11.6 (1.0) | 10.2 (1.2) |
| | | | | 0.65 | 0.56 | 12.8 (1.5) | 10.3 (0.9) | 9.8 (1.1) |
| | | | | 0.37 | 0.56 | 15.7 (0.9) | 12.1 (0.8) | 11.1 (1.0) |
| | | | comp | 0.56 | 0.56 | 14.3 (1.2) | 12.0 (1.1) | 11.6 (1.2) |
| | | | | 0.65 | 0.56 | 12.8 (1.3) | 10.7 (1.1) | 11.1 (1.3) |
| | misc, rec | soc, talk | alt, comp | 0.55 | 0.46 | 12.4 (1.0) | 10.6 (1.1) | 10.1 (1.2) |
| CIFAR10 | Bird, Cat, Deer, Dog, Frog, Horse | Plane, Auto, Ship, Truck | $P_{\text{test}}$ | 0.4 | 0.4 | 21.3 (3.1) | 16.6 (1.3) | 14.6 (0.7) |
| | | | | 0.14 | 0.4 | 16.6 (3.1) | 9.2 (0.6) | 8.4 (0.5) |
| | | | Plane | 0.4 | 0.4 | 21.3 (4.0) | 14.7 (1.1) | 13.1 (0.7) |
| | | | | 0.6 | 0.4 | 21.8 (3.0) | 15.0 (1.0) | 13.8 (0.6) |
| | | | Auto, Truck | 0.25 | 0.4 | 14.7 (1.4) | 10.4 (0.8) | 9.1 (0.5) |
| | | | | 0.4 | 0.4 | 15.7 (2.6) | 12.5 (1.1) | 10.9 (0.7) |
| | | | | 0.55 | 0.4 | 17.0 (3.3) | 13.3 (1.2) | 11.8 (0.6) |
| | Deer, Horse | Plane, Auto | Cat, Dog | 0.5 | 0.5 | 30.9 (2.6) | 21.4 (1.1) | 21.0 (1.0) |

Table 25: Effect of step #1 method on our two-step methods' overall inductive misclassification rate (%) for the 10 LIBSVM datasets in Table 15. The table's upper half reports each method's misclassification rate mean and standard deviation over 100 trials. Boldface denotes each experimental setup's best performing method according to mean misclassification rate. The table's lower half is an alternate visualization showing the difference (Diff.) in misclassification rate mean and standard deviation w.r.t. to our soft method. Red denotes that the associated alternate step #1 method had worse (i.e., higher) mean misclassification rate than soft weighting while green denotes that the alternate method had a better (i.e., lower) mean misclassification rate.

| Dataset | $d$ | PU2aPNU | | | PU2wUU | | |
|---|---|---|---|---|---|---|---|
| | | Soft | Hard | Top-k | Soft | Hard | Top-k |
| banana | 2 | **11.7 (1.6)** | 13.4 (2.0) | 13.1 (1.9) | 13.4 (2.4) | 14.0 (2.6) | **13.3 (2.5)** |
| cod-rna | 8 | **14.6 (3.8)** | 18.6 (3.7) | 18.3 (3.9) | **15.5 (3.2)** | 17.4 (3.2) | 16.7 (3.2) |
| susy | 18 | **25.8 (2.6)** | 27.8 (3.2) | 27.6 (2.5) | **25.8 (2.4)** | 26.3 (3.5) | 26.1 (2.6) |
| ijcnn1 | 22 | **18.0 (2.7)** | 22.1 (3.6) | 18.4 (2.9) | 25.1 (3.4) | 24.3 (3.3) | **21.4 (2.9)** |
| covtype.b | 54 | **32.1 (3.2)** | 40.6 (3.5) | 37.0 (3.5) | **29.7 (2.5)** | 40.1 (3.7) | 34.8 (4.2) |
| phishing | 68 | **9.6 (0.9)** | 9.8 (1.0) | 10.0 (1.1) | 11.6 (2.1) | 10.9 (1.4) | **10.1 (1.2)** |
| a9a | 123 | **26.8 (1.6)** | 28.6 (1.7) | 27.9 (1.6) | **27.4 (2.1)** | 28.5 (1.8) | 28.3 (1.9) |
| connect4 | 126 | **32.9 (2.1)** | 37.2 (2.8) | 35.6 (2.3) | **34.8 (2.7)** | 38.0 (3.1) | 35.3 (2.8) |
| w8a | 300 | **21.6 (2.4)** | 23.7 (2.0) | 24.6 (2.3) | **16.9 (2.7)** | 22.4 (2.4) | 22.0 (2.7) |
| epsilon | 2,000 | **35.0 (4.4)** | 58.6 (3.2) | 54.6 (3.4) | **31.2 (1.1)** | 52.6 (3.9) | 52.9 (5.8) |

| Dataset | $d$ | PU2aPNU | | | PU2wUU | | |
|---|---|---|---|---|---|---|---|
| | | Soft | Diff. Hard | Diff. Top-k | Soft | Diff. Hard | Diff. Top-k |
| banana | 2 | 11.7 (1.6) | 1.7 ( 0.4) | 1.4 ( 0.4) | 13.4 (2.4) | 0.5 ( 0.2) | –0.1 ( 0.1) |
| cod-rna | 8 | 14.6 (3.8) | 4.1 (–0.1) | 3.7 ( 0.1) | 15.5 (3.2) | 1.9 ( 0 ) | 1.2 ( 0 ) |
| susy | 18 | 25.8 (2.6) | 2.0 ( 0.6) | 1.9 (–0.1) | 25.8 (2.4) | 0.6 ( 1.1) | 0.4 ( 0.2) |
| ijcnn1 | 22 | 18.0 (2.7) | 4.1 ( 0.8) | 0.3 ( 0.2) | 25.1 (3.4) | –0.8 ( 0.2) | –3.7 (–0.5) |
| covtype.b | 54 | 32.1 (3.2) | 8.5 ( 0.2) | 4.9 ( 0.3) | 29.7 (2.5) | 10.3 ( 1.2) | 5.0 ( 1.7) |
| phishing | 68 | 9.6 (0.9) | 0.2 ( 0 ) | 0.4 ( 0.2) | 11.6 (2.1) | –0.6 (–0.7) | –1.5 (–0.9) |
| a9a | 123 | 26.8 (1.6) | 1.9 ( 0.2) | 1.2 ( 0 ) | 27.4 (2.1) | 1.2 (–0.4) | 0.9 (–0.2) |
| connect4 | 126 | 32.9 (2.1) | 4.3 ( 0.7) | 2.7 ( 0.2) | 34.8 (2.7) | 3.3 ( 0.4) | 0.6 ( 0.1) |
| w8a | 300 | 21.6 (2.4) | 2.1 (–0.4) | 3.0 (–0.1) | 16.9 (2.7) | 5.6 (–0.3) | 5.1 ( 0 ) |
| epsilon | 2,000 | 35.0 (4.4) | 23.7 (–1.2) | 19.6 (–0.9) | 31.2 (1.1) | 21.4 ( 2.8) | 21.7 ( 4.6) |

Table 26: Effect of step #1 method on our two-step methods' overall inductive misclassification rate (%) for the MNIST, 20 Newsgroups, and CIFAR10 datasets. The table's upper half reports each method's misclassification rate mean and standard deviation over 100 trials. Boldface denotes each experimental setup's best performing method according to mean misclassification rate. The table's lower half is an alternate visualization showing the difference (Diff.) in misclassification rate mean and standard deviation w.r.t. to our soft method. Red denotes that the associated alternate step #1 method had worse (i.e., higher) mean misclassification rate than soft weighting while green denotes that the alternate method had a better (i.e., lower) mean misclassification rate.

| | N | $P_{test}$ | $P_{train}$ | $\pi_{tr}$ | $\pi_{te}$ | PU2aPNU | | | PU2wUU | | |
|---|---|---|---|---|---|---|---|---|---|---|---|
| | | | | | | Soft | Hard | Top-k | Soft | Hard | Top-k |
| MNIST | 0, 2, 4, 6, 8 | 1, 3, 5, 7, 9 | $P_{test}$ | 0.5 | 0.5 | 10.2 (1.5) | 9.8 (1.3) | **7.8 (1.0)** | 11.8 (1.5) | 10.6 (1.1) | **9.3 (1.0)** |
| | | | | 0.29 | 0.5 | 5.4 (0.5) | 5.3 (0.5) | **4.9 (0.4)** | 6.1 (0.7) | **5.5 (0.4)** | 5.6 (0.3) |
| | | | 7, 9 | 0.5 | 0.5 | 6.9 (0.9) | 7.7 (1.2) | **5.9 (0.6)** | 8.0 (1.3) | 7.5 (0.9) | **6.4 (0.5)** |
| | | | | 0.71 | 0.5 | 11.0 (1.4) | 12.9 (1.2) | **9.9 (1.3)** | 14.9 (3.7) | 12.9 (1.1) | **9.9 (1.0)** |
| | | | | 0.38 | 0.5 | 6.4 (0.8) | 6.6 (0.8) | **5.7 (0.6)** | 7.6 (0.9) | 7.0 (0.7) | **6.6 (0.6)** |
| | | | 1, 3, 5 | 0.5 | 0.5 | 8.4 (1.1) | 9.0 (1.0) | **7.2 (0.9)** | 10.0 (1.4) | 9.3 (0.9) | **8.0 (0.8)** |
| | | | | 0.63 | 0.5 | 11.3 (1.4) | 12.8 (1.4) | **10.2 (1.4)** | 14.1 (2.5) | 12.9 (1.2) | **10.5 (1.1)** |
| | 0, 2 | 5, 7 | 1, 3 | 0.5 | 0.5 | 3.5 (1.0) | 4.1 (1.1) | **2.8 (0.6)** | 3.1 (0.7) | 3.7 (0.9) | **2.8 (0.4)** |
| 20 Newsgroups | sci, soc, talk | alt, comp, misc, rec | $P_{test}$ | 0.56 | 0.56 | 14.9 (1.3) | 14.9 (1.4) | **14.4 (1.5)** | 16.6 (2.5) | 15.9 (1.8) | **15.5 (1.9)** |
| | | | | 0.37 | 0.56 | 12.8 (0.6) | 12.9 (0.8) | **12.4 (0.6)** | 14.2 (0.9) | 13.7 (0.9) | **13.1 (0.7)** |
| | | | misc, rec | 0.56 | 0.56 | 13.6 (0.9) | 14.0 (0.9) | **13.4 (0.9)** | 15.1 (1.3) | 14.8 (1.1) | **14.1 (1.1)** |
| | | | | 0.65 | 0.56 | 14.0 (0.9) | 14.4 (0.9) | **13.8 (0.9)** | 15.8 (1.3) | 15.3 (1.2) | **14.6 (0.9)** |
| | | | | 0.37 | 0.56 | 13.7 (0.6) | 13.8 (0.6) | **13.0 (0.7)** | 14.5 (0.8) | 14.1 (0.7) | **13.3 (0.7)** |
| | | | comp | 0.56 | 0.56 | 14.9 (0.7) | 15.7 (0.7) | **14.3 (0.8)** | 15.7 (0.9) | 15.9 (0.8) | **14.6 (0.9)** |
| | | | | 0.65 | 0.56 | 15.5 (1.1) | 16.5 (1.0) | **15.2 (1.2)** | 16.3 (1.4) | 16.7 (1.3) | **15.4 (1.3)** |
| | misc, rec | soc, talk | alt, comp | 0.55 | 0.46 | **7.2 (1.2)** | 8.1 (1.2) | 7.5 (1.2) | **5.8 (1.6)** | 7.1 (1.5) | **5.8 (1.4)** |
| CIFAR10 | Bird, Cat, Deer, Dog, Frog, Horse | Plane, Auto, Ship, Truck | $P_{test}$ | 0.4 | 0.4 | 13.9 (1.2) | 13.6 (0.9) | **12.0 (0.7)** | 15.0 (1.2) | 14.7 (0.9) | **13.2 (0.8)** |
| | | | | 0.14 | 0.4 | **12.0 (0.8)** | 12.1 (0.6) | 12.2 (0.7) | 12.5 (0.9) | 11.8 (0.6) | **11.7 (0.7)** |
| | | | Plane | 0.4 | 0.4 | 14.4 (1.3) | 15.4 (1.1) | **14.1 (0.8)** | 14.9 (1.4) | 15.0 (1.2) | **13.2 (0.8)** |
| | | | | 0.6 | 0.4 | **16.7 (1.5)** | 20.0 (1.5) | 16.9 (1.1) | 20.1 (2.3) | 20.0 (1.8) | **15.5 (1.1)** |
| | | | Auto, Truck | 0.25 | 0.4 | **12.4 (0.7)** | 12.6 (0.7) | **12.4 (0.7)** | 12.8 (0.7) | 12.4 (0.7) | **12.2 (0.7)** |
| | | | | 0.4 | 0.4 | 14.0 (1.2) | 14.7 (1.1) | **13.4 (0.8)** | 14.4 (1.2) | 14.6 (1.3) | **13.1 (0.8)** |
| | | | | 0.55 | 0.4 | 16.2 (1.6) | 17.7 (1.8) | **15.3 (1.0)** | 17.0 (2.1) | 17.7 (2.1) | **14.8 (1.0)** |
| | Deer, Horse | Plane, Auto | Cat, Dog | 0.5 | 0.5 | **15.1 (1.7)** | 20.2 (1.2) | 19.2 (1.1) | **11.2 (0.8)** | 16.3 (1.3) | 14.2 (1.0) |

| | N | $P_{test}$ | $P_{train}$ | $\pi_{tr}$ | $\pi_{te}$ | PU2aPNU | | | PU2wUU | | |
|---|---|---|---|---|---|---|---|---|---|---|---|
| | | | | | | Soft | Diff. Hard | Diff. Top-k | Soft | Diff. Hard | Diff. Top-k |
| MNIST | 0, 2, 4, 6, 8 | 1, 3, 5, 7, 9 | $P_{test}$ | 0.5 | 0.5 | 10.2 (1.5) | –0.4 (–0.2) | –2.5 (–0.5) | 11.8 (1.5) | –1.3 (–0.4) | –2.6 (–0.5) |
| | | | | 0.29 | 0.5 | 5.4 (0.5) | –0.1 ( 0 ) | –0.5 (–0.1) | 6.1 (0.7) | –0.5 (–0.3) | –0.5 (–0.3) |
| | | | 7, 9 | 0.5 | 0.5 | 6.9 (0.9) | 0.8 ( 0.3) | –1.0 (–0.3) | 8.0 (1.3) | –0.5 (–0.4) | –1.6 (–0.8) |
| | | | | 0.71 | 0.5 | 11.0 (1.4) | 1.9 (–0.2) | –1.1 (–0.1) | 14.9 (3.7) | –2.0 (–2.7) | –5.0 (–2.7) |
| | | | | 0.38 | 0.5 | 6.4 (0.8) | 0.2 ( 0 ) | –0.7 (–0.2) | 7.6 (0.9) | –0.6 (–0.2) | –1.0 (–0.3) |
| | | | 1, 3, 5 | 0.5 | 0.5 | 8.4 (1.1) | 0.6 ( 0 ) | –1.2 (–0.2) | 10.0 (1.4) | –0.8 (–0.5) | –2.1 (–0.7) |
| | | | | 0.63 | 0.5 | 11.3 (1.4) | 1.5 ( 0 ) | –1.1 ( 0 ) | 14.1 (2.5) | –1.2 (–1.3) | –2.1 (–0.7) |
| | 0, 2 | 5, 7 | 1, 3 | 0.5 | 0.5 | 3.5 (1.0) | 0.6 ( 0.1) | –0.7 (–0.4) | 3.1 (0.7) | 0.6 ( 0.3) | –0.3 (–0.3) |
| 20 Newsgroups | sci, soc, talk | alt, comp, misc, rec | $P_{test}$ | 0.56 | 0.56 | 14.9 (1.3) | 0 ( 0.2) | –0.5 ( 0.3) | 16.6 (2.5) | –0.7 (–0.8) | –1.1 (–0.6) |
| | | | | 0.37 | 0.56 | 12.8 (0.6) | 0.1 ( 0.1) | –0.4 ( 0 ) | 14.2 (0.9) | –0.5 ( 0 ) | –1.1 (–0.2) |
| | | | misc, rec | 0.56 | 0.56 | 13.6 (0.9) | 0.3 ( 0 ) | –0.2 ( 0 ) | 15.1 (1.3) | –0.3 (–0.2) | –1.0 (–0.2) |
| | | | | 0.65 | 0.56 | 14.0 (0.9) | 0.4 ( 0 ) | –0.2 (–0.1) | 15.8 (1.3) | –0.5 (–0.1) | –1.2 (–0.4) |
| | | | | 0.37 | 0.56 | 13.7 (0.6) | 0.1 ( 0 ) | –0.6 ( 0 ) | 14.5 (0.8) | –0.4 ( 0 ) | –1.2 ( 0 ) |
| | | | comp | 0.56 | 0.56 | 14.9 (0.7) | 0.8 ( 0 ) | –0.6 ( 0 ) | 15.7 (0.9) | 0.2 (–0.1) | –0.1 (–0 ) |
| | | | | 0.65 | 0.56 | 15.5 (1.1) | 1.0 ( 0 ) | –0.4 ( 0.1) | 16.3 (1.4) | 0.4 (–0.1) | –0.9 (–0.1) |
| | misc, rec | soc, talk | alt, comp | 0.55 | 0.46 | 7.2 (1.2) | 1.0 ( 0.1) | 0.3 ( 0 ) | 5.8 (1.6) | 1.4 (–0.1) | 0 (–0.3) |
| CIFAR10 | Bird, Cat, Deer, Dog, Frog, Horse | Plane, Auto, Ship, Truck | $P_{test}$ | 0.4 | 0.4 | 13.9 (1.2) | –0.3 (–0.3) | –1.9 (–0.5) | 15.0 (1.2) | –0.4 (–0.3) | –1.9 (–0.4) |
| | | | | 0.14 | 0.4 | 12.0 (0.8) | 0.1 (–0.1) | 0.1 (–0.1) | 12.5 (0.9) | –0.8 (–0.3) | –0.8 (–0.2) |
| | | | Plane | 0.4 | 0.4 | 14.4 (1.3) | 1.0 (–0.2) | –0.3 (–0.5) | 14.9 (1.4) | 0.1 (–0.2) | –1.8 (–0.6) |
| | | | | 0.6 | 0.4 | 16.7 (1.5) | 3.3 ( 0 ) | 0.2 (–0.4) | 20.1 (2.3) | –0.1 (–0.5) | –4.6 (–1.2) |
| | | | Auto, Truck | 0.25 | 0.4 | 12.4 (0.7) | 0.2 ( 0 ) | 0 ( 0 ) | 12.8 (0.7) | –0.4 (–0.1) | –0.6 ( 0 ) |
| | | | | 0.4 | 0.4 | 14.0 (1.2) | 0.6 (–0.1) | –0.6 (–0.4) | 14.4 (1.2) | 0.3 ( 0.1) | –1.3 (–0.4) |
| | | | | 0.55 | 0.4 | 16.2 (1.6) | 1.5 ( 0.2) | –0.9 (–0.7) | 17.0 (2.1) | 0.7 ( 0 ) | –2.2 (–1.1) |
| | Deer, Horse | Plane, Auto | Cat, Dog | 0.5 | 0.5 | 15.1 (1.7) | 5.1 (–0.5) | 4.1 (–0.6) | 11.2 (0.8) | 5.1 ( 0.6) | 3.0 ( 0.3) |

## E.8 Analyzing the Effect of Positive and Negative Class-Conditional Distribution Shift

The goal of these experiments is to:

1. Demonstrate the effectiveness of our approaches across the entire spectrum of positive-train class-conditional distribution shift.
2. Study how our methods perform when the assumption of a fixed negative class-conditional distribution is violated.

We look at these trends across three datasets (as in Section 7.2): MNIST, 20 Newsgroups, and CIFAR10. The positive and negatives classes are formed by combining two labels from the original dataset (the use of two labels per class is necessary for this experimental setup). Table 27 enumerates each dataset's positive and negative class definitions; these definitions apply for both train and test. The dataset sizes are listed in Table 28; note that $n_{\text{Test}}$ is the size of the inductive test set used to measure performance. The validation set was one-fifth the training set size. The priors were also fixed such that $\pi_{\text{tr}} = \pi_{\text{te}} = 0.5$.

Table 27: Positive and negative class definitions for the class-conditional bias experiments

| Dataset | Positive | | Negative | |
|---|---|---|---|---|
| | $C_1$ | $C_2$ | $C_1$ | $C_2$ |
| MNIST | 8 | 9 | 3 | 4 |
| 20 Newsgroups | sci | rec | comp | talk |
| CIFAR10 | Auto | Plane | Ship | Truck |

Table 28: Dataset sizes for the class-conditional bias experiments

| Dataset | $n_{\text{p}}$ | $n_{\text{tr-u}}$ | $n_{\text{te-u}}$ | $n_{\text{Test}}$ |
|---|---|---|---|---|
| MNIST | 250 | 5,000 | 5,000 | 1,500 |
| 20 Newsgroups | 500 | 2,000 | 2,000 | 1,000 |
| CIFAR10 | 500 | 5,000 | 5,000 | 1,500 |

The default rule in this section is that the positive/negative train/test classes are selected uniformly at random without replacement from their respective subclasses. In each experiment, either the positive-train or negative-train class-conditional distribution is shifted (never both). The test distribution is never biased and is identical for all experiments.

**Positive-Train Shift**   In these experiments, the positive-train class-conditional distribution (i.e., $p_{\text{tr-p}}(x)$) is shifted. Recall that each positive class is composed of two labels; denote them $C_1$ and $C_2$ (e.g., $C_1 = $ Auto and $C_2 = $ Plane for CIFAR10). $\Pr[\text{Label}_{\text{tr}}=C_1|Y = +1]$ is the probability that any positive-valued *training* example has original label $C_1$. Since there are two labels per class,

$$\Pr[\text{Label}_{\text{tr}}=C_2|Y = +1] = 1 - \Pr[\text{Label}_{\text{tr}}=C_1|Y = +1]. \tag{35}$$

The positive-train class-conditional distribution shift entails sweeping $\Pr[\text{Label}_{\text{tr}}=C_1|Y = +1]$ from 0.5 to 1 (i.e., from unbiased on the left to maximally biased on the right). This setup is more challenging than shifting the positive-test distribution since it entails the learner seeing fewer *labeled* examples from positive subclass $C_2$.

Figures 7a, 7c, and 7e show the positive-train shift's effect on the MNIST, 20 Newsgroups, and CIFAR10 misclassification rate respectively (where $C_1$ corresponds to digit 8, document category "rec", and image type "automobile"). PURR's performance was consistent across the entire bias range while the two step methods' (PU2wUU and PU2aPNU) performance improved as bias increased (due to easier identification of negative examples as explained in Section 7.2). In contrast, PUc's performance degrades as bias increases; this degradation is largely due to poor density estimation and demonstrates why covariate shift methods can be non-ideal.

$\text{PN}_{\text{tr}}$ and $\text{PN}_{\text{te}}$ are trained using (labeled) $\mathcal{X}_{\text{tr-u}}$ and $\mathcal{X}_{\text{te-u}}$. Since the test distributions are never biased, $\text{PN}_{\text{te}}$ is unaffected by shift. In contrast, as $\Pr[\text{Label}_{\text{tr}}=C_1|Y = +1]$ increases, there are fewer examples in $\mathcal{X}_{\text{tr-u}}$ with label $C_2$ causing a degradation in $\text{PN}_{\text{tr}}$'s performance.

PUc's and nnPU*'s performance begins to degrade at the same point where $\text{PN}_{\text{tr}}$'s and $\text{PN}_{\text{te}}$'s performance begins to diverge. For nnPU* in particular, this degradation is primarily attributable to fewer examples labeled $C_2$ in $\mathcal{X}_{\text{p}}$. PUc is more robust to bias than nnPU* (as shown by the slower rate of degradation) since it considers distributional shifts.

**Negative-Train Shift** These experiments follow the same basic concept as the positive-train class-conditional distribution shift described above except that the bias is instead applied to the negative-train class-conditional distribution, i.e., $p_{\text{tr-n}}(x)$. This bias means that $p_{\text{tr-n}}(x) \neq p_{\text{te-n}}(x)$. To reiterate, *these experiments deliberately violate Eq. (7)'s assumption* upon which our methods are predicated. The goal here is to understand our methods' robustness under intentionally deleterious conditions. It is more deleterious to bias the negative class in $\mathcal{X}_{\text{tr-u}}$ since both two-step methods and PURR use $\mathcal{X}_{\text{tr-u}}$'s negative risk in dependent calculations; any error propagates and compounds in these subsequent operations.

Let $C_1$ and $C_2$ now be the two labels that make up the negative class (e.g., $C_1 = $ Ship and $C_2 = $ Truck for CIFAR10). Now, $\Pr[\text{Label}_{\text{tr}} = C_1 | Y = -1]$ is swept along the x-axis from 0.5 to 1 (unbiased to maximally biased). The results for MNIST, 20 Newsgroups, and CIFAR10 are in Figures 7b, 7d, and 7f respectively.

With the exception of PU2wUU on MNIST, all of our methods showed moderate robustness to some negative class-conditional distribution bias. In particular, PU2aPNU was almost as robust as PUc in some cases. nnPU*'s robustness here is expected since anything not in $\mathcal{X}_{\text{p}}$ is assumed negative; even under bias, sufficient negative examples exist for each label in $\mathcal{X}_{\text{te-u}}$ to allow nnPU* to learn how to classify those examples.

(a) MNIST positive train bias

(b) MNIST negative train bias

(c) 20 Newsgroups positive train bias

(d) 20 Newsgroups negative train bias

(e) CIFAR10 positive train bias

(f) CIFAR10 negative train bias

Figure 7: Effect of positive ($p_{\text{tr-p}}(x)$) or negative ($p_{\text{tr-n}}(x)$) training class-conditional distribution shift on inductive misclassification rate (%) for the MNIST, 20 Newsgroups, and CIFAR10 datasets. The x-axis corresponds to $\Pr[\text{Label}_{\text{tr}} = C_1 | y = \hat{y}]$ where $\hat{y} \in \{\pm 1\}$. Each data point is the average of 100 trials.

### E.9  Effect of Prior Probability Misestimation

As explained in Section 4, this work treats the positive-class priors, $\pi_{\mathrm{tr}}$ and $\pi_{\mathrm{te}}$, as known. This set of experiments examines our methods' performance when the priors are misspecified.

**Experimental Setup**  These experiments reuse the partially disjoint positive-support experiment setups from Section 7.2's Table 1. Therefore, we are specifically considering the MNIST, 20 Newsgroups, and CIFAR10 datasets with Table 29 summarizing the experimental setups.

$\pi_{\mathrm{tr}}$ and $\pi_{\mathrm{te}}$ in Table 29 are the *actual* prior probabilities used to construct each training and test data set. We tested our methods' performance when each prior was specified correctly and when each prior was misspecified by $\pm20\%$ for a total of $9 = 3 \times 3$ conditions per learner. PUc estimates $\pi_{\mathrm{te}}$ as part of its density-ratio estimation. As such, we only report three bias conditions for PUc, all over training prior $\pi_{\mathrm{tr}}$. Like all previous experiments, performance was evaluated using the inductive (test) misclassification rate, and all methods saw identical datasets splits in each trial.

**Analysis**  Tables 30, 31, and 32 contain the results for MNIST, 20 Newsgroups, and CIFAR10 respectively. Each learner's results are presented in a $3 \times 3$ grid with $\pi_{\mathrm{tr}}$ changing row by row and $\pi_{\mathrm{te}}$ changing column by column. Each cell is shaded red, with a darker background denoting worse performance, i.e., a greater misclassification rate. In all but one setup, our methods outperformed PUc.

Similar to Section E.8's experiments, the MNIST results were most affected by bias. The 20 Newsgroups and CIFAR10 results were more immune due to the richer feature representations generated through transfer learning.

Of our three methods, PU2aPNU was the least affected by misspecified priors. For the two-step methods, the worst performing misestimation profile was dataset specific. In contrast, PURR's performance was always worst when the train and test priors were misspecified in opposite directions. To understand why this is, recall that PURR's definition in Eq. (14) includes prior ratio $\frac{1-\pi_{\mathrm{te}}}{1-\pi_{\mathrm{tr}}}$. This ratio compounds prior misestimations with opposite signs.

As an example, consider the MNIST experiment below with true priors $\pi_{\mathrm{tr}} = \pi_{\mathrm{te}} = 0.5$, making PURR's ideal prior ratio $\frac{1-0.5}{1-0.5} = 1$. This ratio remains 1 even if the priors are misspecified as $\pi_{\mathrm{tr}} = \pi_{\mathrm{te}} = 0.6$ or $\pi_{\mathrm{tr}} = \pi_{\mathrm{te}} = 0.4$. In contrast, if $\pi_{\mathrm{tr}} = 0.4$ and $\pi_{\mathrm{te}} = 0.6$, PURR's (erroneous) prior ratio is $\frac{1-0.6}{1-0.4} \approx 0.67$ – a 33% error. Furthermore, when the priors are misspecified as $\pi_{\mathrm{tr}} = 0.6$ and $\pi_{\mathrm{te}} = 0.4$, the prior ratio jumps to $\frac{1-0.4}{1-0.6} = 1.5$ – a 50% error. This is why over-estimation of the training prior and underestimation of the test prior is always PURR's worst performing configuration.

Table 29: Positive train ($P_{\mathrm{train}}$), positive test ($P_{\mathrm{test}}$), and negative (N) class definitions and actual prior probabilities for the experiments examining the effect of misspecified prior(s) on our algorithms' performance.

|  | N | $P_{\mathrm{train}}$ | $P_{\mathrm{test}}$ | $\pi_{\mathrm{tr}}$ | $\pi_{\mathrm{te}}$ |
|---|---|---|---|---|---|
| MNIST | 0, 2, 4, 6, 8 | 1, 3, 5, 7, 9 | 7, 9 | 0.5 | 0.5 |
| 20 News. | sci, soc, talk | alt, comp, misc, rec | misc, rec | 0.37 | 0.56 |
| CIFAR10 | Bird, Cat, Deer, Dog, Frog, Horse | Plane, Auto, Ship, Truck | Plane | 0.4 | 0.4 |

Table 30: Combined heat map and table showing the effect of incorrectly specified priors $\pi_{tr}$ and $\pi_{te}$ on MNIST's inductive misclassification rate (%). Each result is the average of 100 trials.

| | PURR | | | PU2aPNU | | | PU2wUU | | | PUc |
|---|---|---|---|---|---|---|---|---|---|---|
| | $0.8\pi_{te}$ | $\pi_{te}$ | $1.2\pi_{te}$ | $0.8\pi_{te}$ | $\pi_{te}$ | $1.2\pi_{te}$ | $0.8\pi_{te}$ | $\pi_{te}$ | $1.2\pi_{te}$ | |
| $0.8\pi_{tr}$ | 17.4 | 16.6 | 19.8 | 7.4 | 9.5 | 12.2 | 10.3 | 13.2 | 18.7 | 29.6 |
| $\pi_{tr}$ | 12.9 | 9.2 | 13.6 | 6.6 | 7.4 | 10.1 | 8.5 | 10.3 | 13.9 | 26.7 |
| $1.2\pi_{tr}$ | 25.3 | 15.8 | 12.7 | 18.0 | 7.7 | 7.5 | 16.9 | 8.9 | 10.3 | 26.3 |

Table 31: Combined heat map and table showing the effect of incorrectly specified priors $\pi_{tr}$ and $\pi_{te}$ on 20 Newsgroups's inductive misclassification rate (%). Each result is the average of 100 trials.

| | PURR | | | PU2aPNU | | | PU2wUU | | | PUc |
|---|---|---|---|---|---|---|---|---|---|---|
| | $0.8\pi_{te}$ | $\pi_{te}$ | $1.2\pi_{te}$ | $0.8\pi_{te}$ | $\pi_{te}$ | $1.2\pi_{te}$ | $0.8\pi_{te}$ | $\pi_{te}$ | $1.2\pi_{te}$ | |
| $0.8\pi_{tr}$ | 15.2 | 14.9 | 16.5 | 12.3 | 12.5 | 13.3 | 13.4 | 14.3 | 16.7 | 34.1 |
| $\pi_{tr}$ | 15.9 | 13.8 | 15.1 | 12.6 | 12.8 | 13.5 | 13.4 | 14.2 | 16.1 | 28.6 |
| $1.2\pi_{tr}$ | 18.7 | 13.3 | 14.0 | 13.4 | 14.2 | 15.0 | 14.4 | 15.6 | 17.2 | 24.9 |

Table 32: Combined heat map and table showing the effect of incorrectly specified priors $\pi_{tr}$ and $\pi_{te}$ on CIFAR10's inductive misclassification rate (%). Each result is the average of 100 trials.

| | PURR | | | PU2aPNU | | | PU2wUU | | | PUc |
|---|---|---|---|---|---|---|---|---|---|---|
| | $0.8\pi_{te}$ | $\pi_{te}$ | $1.2\pi_{te}$ | $0.8\pi_{te}$ | $\pi_{te}$ | $1.2\pi_{te}$ | $0.8\pi_{te}$ | $\pi_{te}$ | $1.2\pi_{te}$ | |
| $0.8\pi_{tr}$ | 16.4 | 15.6 | 18.0 | 13.9 | 15.1 | 17.5 | 18.2 | 19.3 | 21.1 | 23.4 |
| $\pi_{tr}$ | 16.0 | 13.7 | 15.3 | 13.3 | 14.4 | 16.6 | 14.2 | 14.9 | 16.9 | 20.1 |
| $1.2\pi_{tr}$ | 20.8 | 15.7 | 14.7 | 16.8 | 16.4 | 17.5 | 15.6 | 15.9 | 17.9 | 19.7 |

## Footnotes

[6]No restrictions are placed on $\mathcal{A}$ other than its existence and that selected hypothesis $\hat{\sigma}_{\mathcal{S}}$ satisfies Eq. (18).

[7] We used the latest version of the 20 Newsgroups dataset with duplicates and cross-posts removed.

[8]The raw TREC05 emails can be downloaded from `https://plg.uwaterloo.ca/~gvcormac/treccorpus/`.

[9]The raw TREC07 emails can be downloaded from `https://plg.uwaterloo.ca/~gvcormac/treccorpus07/`.

[10]Hyperparameter $\gamma$ only applies when using Kiryo et al. [8]'s non-negativity correction. $\gamma$ is not considered by our absolute-value correction.

[11]PU2wUU and PU2aPNU used top-k weighting (see Section E.7.1) for step #1.

[12]Kato et al.'s source code is publicly available at `https://github.com/MasaKat0/PUlearning`.

[13]The test conditions for MNIST, 20 Newsgroups, and CIFAR10 correspond to the unbiased test conditions (i.e., row 1 for each dataset where $P_\text{train} = P_\text{test}$) in Table 1/Table 13.