[Reviews · NeurIPS 2020]

Review 1

Summary and Contributions: The authors propose a method for learning with only positive and unlabled data when the positive class is allowed to have arbitrary shifts. The key assumption is that the negative class distribution stays constant across training and test sets. They provide a two-stage and a one-stage recursive algorithm for solving the problem. Experiments under different shift settings demonstrate the utility of their approach. The authors have addressed the questions raised by the reviewers well in their rebuttal, and I am keeping my score of acceptance.

Strengths: - The authors provide theoretical analysis to their risk estimators and provide more than one algorithm to solve the problem. - The setting of learning from positive data with arbitrary shifts is relevant and interesting and the authors provide several real world examples. - There are thorough empirical evaluations on multiple shift scenarios.

Weaknesses: - There are potentially many variations of this problem, e.g., by exchanging the role of positive and negative classes. One can assume the positive class distribution stays the same while allowing arbitrary shifts in the negative class. It would be clearer to the readers to explain the contributions in terms of these equivalent classes of problems.

Correctness: The theorems and empirical methodolgy are correct to the best of my knowledge.

Clarity: Yes, the paper is clear and well written.

Relation to Prior Work: Yes, the authors sufficiently cite prior works and point out the relations.

Reproducibility: Yes

Additional Feedback:


Review 2

Summary and Contributions: This paper works on the problem of positive-unlabeled binary classification, and the interest is when the positive distribution will change in test compared from train stage, but the negative distribution stays the same between train and test stage. The available data in training stage is positive and unlabeled data from training distribution, and unlabeled data from test distribution. The final goal is to learn a (inductive) classifier from these available resources. A similar problem setting was introduced in one of the previous works, but this paper has different assumptions and has a different algorithm to solve this problem. Experiments show that the proposed methods work better than previous methods for image and NLP datasets.

Strengths: - The paper explains strong potential applications for arbitrary-positive and unlabeled classification. - The writing is well organized and gradually improve one after another: starts with improving max operator with absolute value for original PU, then introduces a two-step method for aPU, and finally proposes a one-step method for aPU. - Experiments are extensive and are shown for both image and NLP domains. - It doesn't rely on the input-output consistent relationship which previous works relied on.

Weaknesses: If I understood Kiryo et al. paper correctly, Section 3 of the paper under review is already proposed in Kiryo et al.'s paper. Kiryo et al. proposed a method with max in Eq. 6, but the final proposal in Algorithm 1 in their paper doesn't use max, but instead flips the gradient when the loss goes below zero (or a specified hyper-parameter value). As far as I understand, mathematically, this is equivalent to using absolute value, because if the value inside abs is negative, it will flip the sign, and this will also flip the gradient's sign. Kiryo et al. has a step size hyper-parameter for adjusting the gradient (gamma), so if this hyper-parameter is set to 1, it seems that it becomes the same as using the absolute value function. The proposed method in Section 3 has the benefit of not having hyper-parameters gamma because it is using gamma=1 implicitly. The idea of PURR is interesting and creative. I think the motivation was to propose a one-step method so that errors in the first step do not propagate. The experiments seems like it is PURR is not necessarily beneficial over two-step methods. The intuition behind this was not explained. (On the other hand, the comparison between aPNU and wUU was clear and results were intuitive.) The experiments seem to use kernel models for PUc but neural network for proposed methods. Is there a reason for not aligning the underlying models? I think it would make the experiments much more meaningful if the underlying settings are aligned as much as possible.

Correctness: See my answer to the previous question about comments on Section 3.

Clarity: Yes.

Relation to Prior Work: Yes.

Reproducibility: Yes

Additional Feedback: Overall comment: Although I enjoyed reading the paper and it proposes novel ideas for PU learning research, I couldn't give a high score because: I feel it is hard to compare between methods in the experiments due to the usage of different models for proposed/baselines, some of the work in this paper (Sec. 3) is already proposed in a previous paper, and the benefit of one of the proposed methods (PURR) is not clear in experiments. Other comments: The output of logistic classifiers will be between 0 and 1, and theoretically it should be an estimate of p(y|x). Practically, the estimate of p(y|x) can become quite noisy, or may overfit and lead to peaky hat{p}(y|x) distributions, according to papers like "On Calibration of Modern Neural Networks" (ICML 2017). Assuming $\hat{\sigma}(x) = p_tr(y=-1|x)$ seems to be a strong assumption, but does this cause any issues in the experiments? A minor suggestion is to investigate confidence-calibration, and see how much sensitive the final PU classifier is for worse calibration. Can we tune $\rho$ in aPNU with validation data? It seems unrealistic to have knowledge of how much overlap there is between $p_{tr-p}(x)$ and $p_{te-p}(x)$. ________________________________________________ After rebuttal period: Thank you for answering my questions. Some of my initial concerns have been resolved and I have decided to raise my score. The response helped me understand the difference from Kiryo et al. 2017 paper. However, Table 16 (Appendix E.5) seem to show quite similar results and wasn't so sure of the significance of the proposal (although novel). For using different models for different baselines: I now think the comparisons are fair. It still puts some burden on the reader to get to that point, so if accepted, I would like to suggest using the same models as much as possible in the camera-ready version.


Review 3

Summary and Contributions: This paper focuses on an arbitrary-positive, unlabeled (aPU) learning setting where the labeled (positive) data may be arbitrarily different from the target distribution’s positive class. The paper uses absolute-value correction for PU risk estimation. It also proposes a two-step formulation method to solve aPU problem.

Strengths: +The arbitrary-positive, unlabeled (aPU) learning is an under-explored problem. The paper proposes a two-step formulation method to solve aPU problem., i.e., create a representative negative set and classify X_{te-u}. +The paper replaces non-negativity constraint (e.g., max term) by absolute-value correction.

Weaknesses: -Some notations are used before/without definition. This makes it hard to understand. For example, Line 70, p, n, and u are not defined until Line 103. -The paper directly uses some conclusions from other papers without any explanation, which makes it hard to understand. It would be nice to organize a brief introduction to make the paper self-contained.

Correctness: No totally understand.

Clarity: No.

Relation to Prior Work: Yes.

Reproducibility: No

Additional Feedback: I have read the response and the authors addressed my concerns. I raised my score to accept this paper.


Review 4

Summary and Contributions: The paper tackles the problem of PU learning where the labeled set of positives is biased arbitrarily. In addition to an unlabeled set that contains unbiased positives and negatives, they exploit another unlabeled set made of positives, biased similarly as the labeled set, and unbiased negatives. They introduce a modification of the non-negative risk estimator (for unbiased PU) by replacing the max operation by absolute value, which leads to a simpler algorithm. The absolute value based formulation of the risk is further modified in three different ways to provide a consistent risk estimators based on the biased positive set and the two unlabeled sets.

Strengths: 1) The problem formulation based on two unlabeled sets is novel and has some real world applications. 2) The method and the theoretical results, although not surprising, are novel. 3) Experiments on many datasets demonstrate the efficacy of the method.

Weaknesses: 1) The assumption that the positives in the training unlabeled set has the same bias as the labeled positives is restrictive and doesn’t solve the bias problem in PU learning in general. For example, in the medical domain it might be difficult to construct an unlabeled set where the distribution of diseased individuals is the same as the biased set of individuals known to have the disease and also have the healthy individuals in the set represent an unbiased distribution of healthy individuals. 2) The authors claim that the comparison to the bPU methods is infeasible. I understand that the bias assumptions of these methods are restrictive, but it should be possible to include them as baseline.

Correctness: yes

Clarity: The paper is very well written.

Relation to Prior Work: yes

Reproducibility: Yes

Additional Feedback: The epidemiology example might not be apt for this problem. If the population of the region is constant across the years and the distribution of diseased individuals changes, would it not imply that the distribution of the healthy individuals also changes? Since every individual in the region if not healthy is diseased and vice versa. =======================After rebuttal====================== I still think that the epidemiology example should be removed since the distribution of negative is not stationary from one year to another. If the authors want to keep it they should give some supporting evidence. The authors haven't responded to my comment on not including bPU for baseline, so I'm lovering my score to marginally above acceptance.

[Author Response · NeurIPS 2020]

We thank the reviewers for their very constructive feedback! We are encouraged that most reviewers (R1,R2,R4)[1] found

our paper to be well-written and that our experiments were thorough/extensive and reproducible. Space prohibits us

addressing all points raised by the reviewers so below we focus on the most pressing points from each reviewer.

**@R1,R2,R3,R4**: We will use the final submission's extra allowed page to clarify all points discussed below.

**@R2 Is our absolute-value correction the same as Kiryo et al.'s approach?**: No. Our approach is different in two

ways. First, during optimization, when $\widehat{R}_{u}^{-}(g) - \pi\widehat{R}_{p}^{-}(g) < 0$, Kiryo et al.'s update gradient is: $\gamma\nabla_{\theta}(-\widehat{R}_{u}^{-}(g) + \pi\widehat{R}_{p}^{-}(g))$.

In contrast, our method's gradient is: $\nabla_{\theta}(\pi\widehat{R}_{p}^{+}(g) - \widehat{R}_{u}^{-}(g) + \pi\widehat{R}_{p}^{-}(g))$. Put simply, Kiryo et al. stop optimizing $\widehat{R}_{p}^{+}(g)$

whenever $\widehat{R}_{u}^{-}(g) - \pi\widehat{R}_{p}^{-}(g) < 0$ whereas we always optimize $\widehat{R}_{p}^{+}(g)$. This means that we spend comparatively more

time optimizing the positive-labeled risk. Second, when estimating performance (i.e., risk) on validation data, our

approach penalizes implausible negative risk estimates while Kiryo et al.'s method does not.

Empirically, we find that this "soft-constraint" approach to implausible negative risk yields comparable or better models

(see supplemental Sec. E.5). We also show in the supplementals (e.g., Sec. C(.2) and Alg. 3) that all of our methods can

work with the Kiryo et al. approach, albeit with a (much) more complex implementation.

**@R4 Assumption of same bias between positives in positive-labeled set $\mathcal{X}_{p}$ and training unlabeled set $\mathcal{X}_{\text{tr-u}}$**: In

our problem setting, $\mathcal{X}_{p}$ is an unbiased sample of the positive examples in $\mathcal{X}_{\text{tr-u}}$, but $\mathcal{X}_{p}$ could be arbitrarily different

from the positive examples in test unlabeled set $\mathcal{X}_{\text{te-u}}$. However, if $\mathcal{X}_{p}$ is not representative of the positives in $\mathcal{X}_{\text{tr-u}}$, then

our two-step methods could still be adapted to handle this case by using any bPU algorithm in step 1 to create surrogate

negative set $\widetilde{\mathcal{X}}_{n}$, and our wUU estimator would be used in step 2 as normal. Thank you for this suggestion; we will note

this additional flexibility of our two-step methods in the paper.

**@R2 Why do our two-step methods sometimes outperform our joint approach (PURR)?**: Supplemental tables 13

and 15 detail many empirical setups where our PURR estimator outperforms both the baselines, PUc and nnPU*, and

our two-step approach. Nonetheless, the somewhat counterintuitive finding that a joint aPU learning method is not

consistently the top performer is itself interesting and supports PURR's inclusion in the paper.

As an intuition, note that PURR, with its three risk decompositions/corrections, is strictly harder to optimize than

wUU, aPNU, and nnPU, which each have one correction. This can lead to worse accuracy compared to the two-step

methods, especially on easier problems like MNIST, where each step can be solved accurately on its own. When the

dimensionality is low and dataset sizes are small (see Sec. E.4), PURR is generally the top performer since PURR is

able to extract more information by better combining the limited available data.

**@R2 How to empirically compare our methods to the baselines given the implementation differences**: For PUc,

we used the original authors' implementation, which relies on kernel methods for density-ratio estimation. For baseline

nnPU* as well as our methods (none of which require density estimation), we followed the standard of most recent

PU learning work (including Kiryo et al. in their nnPU paper), which uses neural networks.

However, our experiments show that PUc's biggest limitation is not its representation: On *unshifted* data (Table 1 row 1

for each dataset), PUc's performance is close to nnPU* and slightly outperforms our methods. On *shifted* data (Tab. 1

rows 2 & 3 for each dataset), PUc's & nnPU*'s performance degrades while our methods' performance improves. Thus,

data shift (and methods for handling it) are the biggest factor in performance. See also Sec. E.1 which shows simple

aPU learning tasks that our methods can successfully learn while PUc cannot – specifically with identical models.

We will add a "Discussion" subsection to the paper's "Experimental Results" (Sec. 7) to include the additional context

in our response to this question as well as to the previous question on our joint method PURR.

**@R4 Negative-Class Shift in Epidemiology Example**: We expect that negative-class shifts will be small relative to

shifts in the positive class, both in epidemiology and many other domains. Our experiments on TREC spam emails & in

Sec. E.6 show that our methods are still useful even when our assumption is violated and the negative class shifts some.

**@R1 Problem Variations**: Sakai & Shimizu discuss additional problem variations including PU learning with arbitrary-

negative shift, which they show is trivially equivalent to standard PU learning using positive set $\mathcal{X}_{p}$ and test unlabeled

set $\mathcal{X}_{\text{te-u}}$. If both classes shift, learning is impossible without additional datasets or assumptions (e.g., consistent

input/output for covariate shift). This expanded discussion of problem variations will be added to the paper.

**@R3 Not Reproducible**: Supplemental Sec. D enumerates our complete experimental setup, and our submission

included runnable source code to replicate all experiments. We respectfully believe our experiments are reproducible.

**@R3 Notation Used Before Defined?**: We will update the paper to clarify that line 70's notation is based on the

distribution notation listed at the end of the preceding paragraph (lines 65–66).

## Footnotes

[1]We denote each reviewer R$X$, where $X$ is the corresponding reviewer ID in CMT. Each reviewer identifier is also color coded.


[Meta-Review · NeurIPS 2020]

Following the author response and discussion, all reviewers had an overall positive impression of the paper, highlighting some salient features: + studies an interesting and under-explored problem setting, namely, PU learning where the positive samples are from a distribution unrelated to that of the target distribution + the proposed method is equipped with theoretical guarantees, and is demonstrated to perform well empirically Some areas for improvement include: - the lack of comparison against bPU. The argument of such techniques making an assumption that does not hold is fine, but how well do they perform on the tasks considered here? - lack of clarity about distinction to Kiryo et al., and reason for differing empirical setups (clarified in the response). - more qualification about the scope of the setting, and when it may not be appropriate. The authors are encouraged to incorporate these in a revised version.